# Neural Dueling Bandits: Preference-Based Optimization with Human Feedback

**Arun Verma**[1*], **Zhongxiang Dai**[2*], **Xiaoqiang Lin**[3],
**Patrick Jaillet**[4], **Bryan Kian Hsiang Low**[1,3]
[1]Singapore-MIT Alliance for Research and Technology, Republic of Singapore
[2]The Chinese University of Hong Kong, Shenzhen, China
[3]Department of Computer Science, National University of Singapore, Republic of Singapore
[4]LIDS and EECS, Massachusetts Institute of Technology, USA
arun.verma@smart.mit.edu, daizhongxiang@cuhk.edu.cn,
xiaoqiang.lin@u.nus.edu, jaillet@mit.edu, lowkh@comp.nus.edu.sg

## Abstract

Contextual dueling bandit is used to model the bandit problems, where a learner's goal is to find the best arm for a given context using observed noisy human preference feedback over the selected arms for the past contexts. However, existing algorithms assume the reward function is linear, which can be complex and non-linear in many real-life applications like online recommendations or ranking web search results. To overcome this challenge, we use a neural network to estimate the reward function using preference feedback for the previously selected arms. We propose upper confidence bound- and Thompson sampling-based algorithms with sub-linear regret guarantees that efficiently select arms in each round. We also extend our theoretical results to contextual bandit problems with binary feedback, which is in itself a non-trivial contribution. Experimental results on the problem instances derived from synthetic datasets corroborate our theoretical results.

## 1 Introduction

Contextual dueling bandits (or preference-based bandits) (Saha, 2021; Bengs et al., 2022; Li et al., 2024) is a sequential decision-making framework that is widely used to model the contextual bandit problems (Li et al., 2010; Chu et al., 2011; Krause & Ong, 2011; Zhou et al., 2020; Zhang et al., 2021) in which a learner's goal is to find an optimal arm by sequentially selecting a pair of arms (also refers as a *duel*) and then observing noisy human preference feedback (i.e., one arm is preferred over another) for the selected arms. Contextual dueling bandits has many real-life applications, such as online recommendation, ranking web search, fine-tuning large language models, and rating two restaurants or movies, especially in the applications where it is easier to observe human preference between two options (arms) than knowing the absolute reward for the selected option (arm). The preference feedback between two arms[1] is often assumed to follow the Bradley-Terry-Luce (BTL) model (Hunter, 2004; Luce, 2005; Saha, 2021; Bengs et al., 2022; Li et al., 2024) in which the probability of preferring an arm is proportional to the exponential of its reward.

Since the number of contexts (e.g., users of online platforms) and arms (e.g., movies/search results to recommend) can be very large (or infinite), the reward of an arm is assumed to be parameterized by an unknown function, e.g., a linear function (Saha, 2021; Bengs et al., 2022; Li et al., 2024). However, the reward function may not always be linear in practice. To overcome this challenge, this paper parameterizes the reward function via a non-linear function, which needs to be estimated using the available preference feedback for selected arms. To achieve this, we can estimate the non-linear function by using either a Gaussian processes (Williams & Rasmussen, 2006; Srinivas et al., 2010; Chowdhury & Gopalan, 2017) or a neural network (Zhou et al., 2020; Zhang et al., 2021). However, due to the limited expressive power of the Gaussian processes, it fails when optimizing

---

[*]Equal contribution and corresponding authors.

[1]For more than two arms, the preferences are assumed to follow the Plackett-Luce model (Saha, 2021; Soufiani et al., 2014).

highly complex functions. In contrast, neural networks (NNs) possess strong expressive power and can model highly complex functions (Dai et al., 2023; Lin et al., 2023).

In this paper, we first introduce the problem setting of neural dueling bandits, in which we use a neural network to model the unknown reward function in contextual dueling bandits. As compared to the existing work on neural contextual bandits (Zhou et al., 2020; Zhang et al., 2021), we have to use cross-entropy loss as an objective function for training the neural network to estimate the unknown non-linear reward function due to the preference feedback (i.e., 0/1). We next propose two neural dueling bandit algorithms based on, respectively, upper confidence bound (UCB) (Auer et al., 2002; Li et al., 2010; Chu et al., 2011; Abbasi-Yadkori et al., 2011; Li et al., 2017; Zhou et al., 2020) and Thompson sampling (TS) (Krause & Ong, 2011; Agrawal & Goyal, 2013a;b; Chowdhury & Gopalan, 2017; Zhang et al., 2021) (more details are in Section 3.2). Note that the existing contextual dueling bandit works (Saha, 2021; Bengs et al., 2022; Li et al., 2024) use different ways to select the pair of arms (more details are given in Appendix C.2), hence leading to different arm-selection strategies as compared to our work. Due to the differences in arm-selection strategy and reward function (which is non-linear and estimated using NNs), our regret analysis is also completely different.

Furthermore, existing neural contextual bandit works use root mean square error (RMSE) as an objective function for training the neural networks due to the assumption of real-valued reward or feedback. Therefore, their key results, especially the confidence ellipsoid, only hold for RMSE and can not be straightforwardly extended to our setting, which uses cross-entropy loss. Nevertheless, we derive an upper bound on the estimation error (i.e., represented as a confidence ellipsoid) of *the difference between the reward values of any pair of arms* (Theorem 1) predicted by the trained neural network, which is valid as long as the neural network is sufficiently wide. This result provides a theoretical assurance of the quality of our trained neural network using the preference feedback to minimize the cross-entropy loss. Based on the theoretical guarantee on the estimation error, we derive upper bounds on the cumulative regret of both of our algorithms (Theorem 2 and Theorem 3), which are sub-linear under some mild conditions. Our regret upper bounds lead to several interesting and novel insights (more details are given in Section 3.3).

As a special case, we extend our results to neural contextual bandit problems with binary feedback in Appendix D, which is itself of independent interest. Interestingly, our theoretical results also provide novel theoretical insights regarding the *reinforcement learning with human feedback* (RLHF) algorithm (Appendix E). Specifically, our Theorem 1 naturally provides *a theoretical guarantee on the quality of the learned reward model* in terms of its accuracy in estimating the reward differences between pairs of responses. Finally, we empirically validate the different performance aspects of our proposed algorithms in Section 5 using problem instances derived from synthetic datasets.

## 2 PROBLEM SETTING

**Contextual dueling bandits.** We consider a contextual dueling bandit problem in which a learner selects two arms (also refers as a *duel*) for a given context and observes preference feedback over selected arms. The learner's goal is to find the best arm for each context. Our problem differs from standard contextual bandits in which a learner selects a single arm and observes an absolute numerical reward for that arm. Let $\mathcal{C} \subset \mathbb{R}^{d_c}$ be the context set and $\mathcal{A} \subset \mathbb{R}^{d_a}$ be finite arm set, where $d_c \geq 1$ and $d_a \geq 1$. At the beginning of round $t$, the environment generates a context $c_t \in \mathcal{C}$ and the learner selects two arms (i.e., $a_{t,1}$, and $a_{t,2}$) from the finite arm set $\mathcal{A}$. After selecting two arms, the learner observes stochastic preference feedback $y_t$ for the selected arms, where $y_t = 1$ implies the arm $a_{t,1}$ is preferred over arm $a_{t,2}$ and $y_t = 0$ otherwise. We assume that the preference feedback depends on an unknown non-linear reward function $f : \mathcal{C} \times \mathcal{A} \to \mathbb{R}$. For brevity, we denote the set of all context-arm feature vectors in the round $t$ by $\mathcal{X}_t$. We also use $\mathcal{X}$ to denote the set of all feature vectors: $\mathcal{X}_t \subset \mathcal{X}, \forall t$ and $x_{t,a}$ to represent the context-arm feature vector for context $c_t$ and an arm $a$.

**Stochastic preference model.** We assume the preference has a Bernoulli distribution that follows the Bradley-Terry-Luce (BTL) model (Hunter, 2004; Luce, 2005), which is commonly used in the dueling bandits (Saha, 2021; Bengs et al., 2022; Li et al., 2024). Under the BTL preference model, the probability that the first selected arm $(x_{t,1})$ is preferred over the second selected arm $(x_{t,2})$ for the the given context $c_t$ and latent reward function $f$ is given by

$$\mathbb{P}\{x_{t,1} \succ x_{t,2}\} = \mathbb{P}\{y_t = 1 | x_{t,1}, x_{t,2}\} = \frac{\exp\left(f(x_{t,1})\right)}{\exp\left(f(x_{t,1})\right) + \exp\left(f(x_{t,2})\right)} = \mu\left(f(x_{t,1}) - f(x_{t,2})\right).$$

where $x_{t,1} \succ x_{t,2}$ denotes that $x_{t,1}$ is preferred over $x_{t,2}$, $\mu(x) = 1/(1+e^{-x})$ is the sigmoid function and $f(x_{t,i})$ is the latent reward of the $i$-th selected arm. Our results hold for other preference models like the Thurstone-Mosteller model and Exponential Noise as long as the stochastic transitivity holds (Bengs et al., 2022). To generalize our results across preference models, we make the following assumptions on function $\mu$ (also known as a *link function* (Li et al., 2017; Bengs et al., 2022)):

**Assumption 1.**  • $\kappa_\mu \doteq \inf_{x,x' \in \mathcal{X}} \dot{\mu}(f(x) - f(x')) > 0$ *for all pairs of context-arm.*

  • *The link function* $\mu : \mathbb{R} \to [0,1]$ *is continuously differentiable and Lipschitz with constant* $L_\mu$. *For sigmoid function,* $L_\mu \leq 1/4$.

**Performance measure.**  After selecting two arms, denoted by $x_{t,1}$ and $x_{t,2}$, in round $t$, the learner incurs an instantaneous regret. There are two common notions of instantaneous regret in the dueling bandits setting (Saha, 2021; Bengs et al., 2022; Li et al., 2024), i.e., average instantaneous regret: $r_t^a \doteq f(x_t^\star) - (f(x_{t,1}) + f(x_{t,2}))/2$, and weak instantaneous regret: $r_t^w \doteq f(x_t^\star) - \max\{f(x_{t,1}), f(x_{t,2})\}$, where $x_t^\star = \operatorname{argmax}_{x \in \mathcal{X}_t} f(x)$ denotes the best arm for a given context that maximizes the value of the underlying reward function. After observing preference feedback for $T$ pairs of arms, the *cumulative regret* (or regret, in short) of a sequential policy is given by $\mathfrak{R}_T^\tau = \sum_{t=1}^T r_t^\tau$, where $\tau \doteq \{a, w\}$. Note that $\mathfrak{R}_T^w \leq \mathfrak{R}_T^a$. Any good policy should have sub-linear regret, i.e., $\lim_{T \to \infty} \mathfrak{R}_T^\tau / T = 0$. A policy with a sub-linear regret implies that the policy will eventually find the best arm and recommend only the best arm in the duel for the given contexts.

## 3 NEURAL DUELING BANDITS

Having a good reward function estimator is the key for any contextual bandit algorithm to achieve good performance, i.e., smaller regret. As the underlying reward function is complex and non-linear, we use fully connected neural networks (Zhou et al., 2020; Zhang et al., 2021) to estimate the reward function only using the preference feedback. Using this estimated reward function, we propose two algorithms based on the UCB and TS with sub-linear regret guarantees.

### 3.1 REWARD FUNCTION ESTIMATION USING NEURAL NETWORK

To estimate the unknown reward function $f$, we use a fully connected neural network (NN) with depth $L \geq 2$, the width of hidden layer $m$, and ReLU activations as done in Zhou et al. (2020) and Zhang et al. (2021). Let $h(x; \theta)$ represent the output of a full-connected neural network with parameters $\theta$ for context-arm feature vector $x$, which is defined as follows:

$$h(x; \theta) = \boldsymbol{W}_L \text{ReLU}\left(\boldsymbol{W}_{L-1} \text{ReLU}\left(\cdots \text{ReLU}\left(\boldsymbol{W}_1 x\right)\right)\right),$$

where $\text{ReLU}(x) \doteq \max\{x, 0\}$, $\boldsymbol{W}_1 \in \mathbb{R}^{m \times d}$, $\boldsymbol{W}_l \in \mathbb{R}^{m \times m}$ for $2 \leq l < L$, $\boldsymbol{W}_L \in \mathbb{R}^{m \times 1}$. We denote the parameters of NN by $\theta = (\text{vec}(\boldsymbol{W}_1); \cdots \text{vec}(\boldsymbol{W}_L))$, where $\text{vec}(A)$ converts a $M \times N$ matrix $A$ into a $MN$-dimensional vector. We use $m$ to denote the width of every layer of the NN, use $p$ to represent the total number of NN parameters, i.e., $p = dm + m^2(L-1) + m$, and use $g(x; \theta)$ to denote the gradient of $h(x; \theta)$ with respect to $\theta$.

The arms selected by the learner for context received in round $s$ is denoted by $x_{s,1}, x_{s,2} \in \mathcal{X}_s$ and the observed stochastic preference feedback is denoted by $y_s = \mathbb{1}(x_{s,1} \succ x_{s,2})$, which is equal to $1$ if the arm $x_{s,1}$ is preferred over the arm $x_{s,2}$ and $0$ otherwise. At the beginning of round $t$, we use the current history of observations $\{(x_{s,1}, x_{s,2}, y_s)\}_{s=1}^{t-1}$ to train the neural network (NN) using gradient descent to minimize the following loss function:

$$\mathcal{L}_t(\theta) = -\frac{1}{m} \sum_{s=1}^{t-1} \left[\log \mu\left((-1)^{1-y_s}\left[h(x_{s,1}; \theta) - h(x_{s,2}; \theta)\right]\right)\right] + \frac{1}{2}\lambda \|\theta - \theta_0\|_2^2, \tag{1}$$

Here $\theta_0$ represents the initial parameters of the NN, and we initialize $\theta_0$ following the standard practice of neural bandits (Zhou et al., 2020; Zhang et al., 2021) (refer to Algorithm 1 in Zhang et al. (2021) for details). Here, minimizing the first term in the loss function (i.e., the term involving the summation from $t-1$ terms) corresponds to the maximum log likelihood estimate (MLE) of the parameters $\theta$. Next, we develop algorithms that use the trained NN with parameter $\theta_t$ to select the best arms (duel) for each context.

## 3.2 Neural Dueling Bandit Algorithms

With the trained NN as an estimate for the unknown reward function, the learner has to decide which two arms (or duel) must be selected for the subsequent contexts. We use UCB- and TS-based algorithms that handle the exploration-exploitation trade-off efficiently.

**UCB-based algorithm.** Using upper confidence bound for dealing with the exploration-exploitation trade-off is common in many sequential decision-making problems (Bengs et al., 2022; Zhou et al., 2020; Auer et al., 2002). We propose a UCB-based algorithm named **NDB-UCB**, which works as follows: At the beginning of the round $t$, the algorithm trains the NN using available observations. After receiving the context, it selects the first arm greedily (i.e., by maximizing the output of the trained NN with parameter $\theta_t$) as follows:

$$x_{t,1} = \arg\max_{x \in \mathcal{X}_t} h(x; \theta_t). \tag{2}$$

Next, the second arm $x_{t,2}$ is selected optimistically, i.e., by maximizing the UCB value:

---

**NDB-UCB** Algorithm for Neural Dueling Bandit based on Upper Confidence Bound

---

**Tuning parameters:** $\delta \in (0, 1)$, $\lambda > 0$, and $m > 0$
2: **for** $t = 1, \ldots, T$ **do**
3:     Train the NN using $\{(x_{s,1}, x_{s,2}, y_s)\}_{s=1}^{t-1}$ by minimizing the loss function defined in Eq. (1)
4:     Receive a context and $\mathcal{X}_t$ denotes the corresponding context-arm feature vectors
5:     Select $x_{t,1} = \arg\max_{x \in \mathcal{X}_t} h(x; \theta_t)$ as given in Eq. (2))
6:     Select $x_{t,2} = \arg\max_{x \in \mathcal{X}_t} [h(x; \theta_t) + \nu_T \sigma_{t-1}(x, x_{t,1})]$ (as given in Eq. (3))
7:     Observe preference feedback $y_t = \mathbb{1}_{\{x_{t,1} \succ x_{t,2}\}}$
8: **end for**

---

$$x_{t,2} = \arg\max_{x \in \mathcal{X}_t} \left[ h(x; \theta_t) + \nu_T \sigma_{t-1}(x, x_{t,1}) \right], \tag{3}$$

where $\nu_T \doteq \left( \beta_T + B\sqrt{\lambda/\kappa_\mu} + 1 \right) \sqrt{\kappa_\mu/\lambda}$ in which $\beta_T \doteq \frac{1}{\kappa_\mu} \sqrt{\widetilde{d} + 2\log(1/\delta)}$ and $\widetilde{d}$ is the *effective dimension*. We define the effective dimension in Section 3.3 (see Eq. (4)). We define

$$\sigma_{t-1}^2(x_1, x_2) \doteq \frac{\lambda}{\kappa_\mu} \left\| \frac{1}{\sqrt{m}} (\varphi(x_1) - \varphi(x_2)) \right\|_{V_{t-1}^{-1}}^2,$$

where $V_t \doteq \sum_{s=1}^t \varphi'(x_s) \varphi'(x_s)^\top \frac{1}{m} + \frac{\lambda}{\kappa_\mu} \mathbf{I}$. Here, $\varphi'(x_s) \doteq \varphi(x_{s,1}) - \varphi(x_{s,2}) = g(x_{s,1}; \theta_0) - g(x_{s,2}; \theta_0)$ and $g(x; \theta_0)/\sqrt{m}$ is used as the random features approximation for context-arm feature vector $x$. Intuitively, after the first arm $x_{t,1}$ is selected, *a larger $\sigma_{t-1}^2(x, x_{t,1})$ indicates that $x$ is very different from $x_{t,1}$ given the information of the previously selected pairs of arms.* Hence, the second term in Eq. (3) encourages the second selected arm to be different from the first arm.

**TS-based algorithm.** Thompson sampling (Li et al., 2024; Agrawal & Goyal, 2013b) selects an arm according to its probability of being the best. Many works (Li et al., 2024; Chowdhury & Gopalan, 2017; Agrawal & Goyal, 2013b; Chapelle & Li, 2011) have shown that TS is empirically superior than to its counterpart UCB-based bandit algorithms. Therefore, in addition, we also propose another algorithm based on TS named **NDB-TS**, which works similarly to **NDB-UCB** except that the second arm $x_{t,2}$ is selected differently. To select the second arm $x_{t,2}$, for every arm $x \in \mathcal{X}_t$, it firstly samples a reward $r_t(x) \sim \mathcal{N}\left( h(x; \theta_t) - h(x_{t,1}; \theta_t), \nu_T^2 \sigma_{t-1}^2(x, x_{t,1}) \right)$ and then selects the second arm as $x_{t,2} = \arg\max_{x \in \mathcal{X}_t} r_t(x)$.

## 3.3 Regret analysis

Let $K$ denote the finite number of available arms in each round, $\mathbf{H}$ denote the NTK matrix for all $T \times K$ context-arm feature vectors in the $T$ rounds, and $h = \left( f(x_1^1), \ldots, f(x_T^K) \right)$. The NTK matrix $\mathbf{H}$ definition is adapted to our setting from Definition 4.1 of Zhou et al. (2020). We denote the $j$-th element of the vector $x$ by $x_j$. We now introduce the assumptions needed for our regret analysis, all of which are standard assumptions in neural bandits Zhou et al. (2020); Zhang et al. (2021).

**Assumption 2.** *Without loss of generality, we assume that*

- *the reward function is bounded: $|f(x)| \le 1, \forall x \in \mathcal{X}_t, t \in [T]$,*
- *there exists $\lambda_0 > 0$ s.t. $\mathbf{H} \succeq \lambda_0 I$, and*
- *all context-arm feature vectors satisfy $\|x\|_2 = 1$ and $x_j = x_{j+d/2}$, $\forall x \in \mathcal{X}_t, \forall t \in [T]$.*

The last assumption in Assumption 2 above, together with the way we initialize $\theta_0$ (i.e., following standard practice in neural bandits Zhou et al. (2020); Zhang et al. (2021)), ensures that $h(x; \theta_0) = 0, \forall x \in \mathcal{X}_t, \forall t \in [T]$. The assumption of $x_j = x_{j+d/2}$ is a mild assumption and commonly used in the neural bandits literature Zhou et al. (2020); Zhang et al. (2021). This assumption is just for convenience in regret analysis: for any context $x$, $\|x\| = 1$, we can always construct a new context $x' = (x^\top, x^\top)^\top / \sqrt{2}$ that satisfies this assumption Zhou et al. (2020).

Let $\mathbf{H}' \doteq \sum_{s=1}^T \sum_{(i,j) \in C_2^K} z_j^i(s) z_j^i(s)^\top \frac{1}{m}$, in which $z_j^i(s) = \varphi(x_{s,i}) - \varphi(x_{s,j})$ and $C_2^K$ denotes all pairwise combinations of $K$ arms. We now define the *effective dimension* as follows:

$$\widetilde{d} = \log \det \left( \frac{\kappa_\mu}{\lambda} \mathbf{H}' + \mathbf{I} \right). \tag{4}$$

Compared to the previous works on neural bandits, our definition of $\widetilde{d}$ features extra dependencies on $\kappa_\mu$. Moreover, our $\mathbf{H}'$ contains $T \times K \times (K-1)$ contexts, which is more than the $T \times K$ contexts of Zhou et al. (2020) and Zhang et al. (2021).[2] Hence, our $\widetilde{d}$ is expected to be larger than their standard effective dimension. It follows from the Determinant-Trace Inequality (Lemma 10 of Abbasi-Yadkori et al. (2011)), the determinant of a covariance matrix in directly proportional to the number of feature vectors. In our setting, the effective dimension $\widetilde{d}$ is $\log \det \left( \frac{\kappa_\mu}{\lambda} \mathbf{H}' + \mathbf{I} \right)$ and $\mathbf{H}'$ contains $T \times K \times (K-1)$ contexts (i.e., feature vectors), which is more than the $T \times K$ contexts in the standard neural bandits. Therefore, the effective dimension $\widetilde{d}$ in the neural dueling bandits is larger than that of the standard neural bandits.

Note that placing an assumption on $\widetilde{d}$ above is analogous to the assumption on the eigenvalue of the matrix $M_t$ in the work on linear dueling bandits Bengs et al. (2022). For example, in order for our final regret bound to be sub-linear, we only need to assume that $\widetilde{d} = \widetilde{o}(\sqrt{T})$, which is analogous to the assumption from Bengs et al. (2022): $\sum_{t=\tau+1}^T \lambda_{\min}^{-1/2}(M_t) \le c\sqrt{T}$.

We use the above fact to prove our following result, which is equivalent to the confidence ellipsoid results used in the existing bandit algorithms (Li et al., 2017).

**Theorem 1.** *Let $\delta \in (0, 1)$, $\varepsilon'_{m,t} \doteq C_2 m^{-1/6} \sqrt{\log m} L^3 \left( \frac{t}{\lambda} \right)^{4/3}$ for some absolute constant $C_2 > 0$. As long as $m \ge poly(T, L, K, 1/\kappa_\mu, L_\mu, 1/\lambda_0, 1/\lambda, \log(1/\delta))$, then with probability of at least $1 - \delta$,*
$$| [f(x) - f(x')] - [h(x; \theta_t) - h(x'; \theta_t)] | \le \nu_T \sigma_{t-1}(x, x') + 2\varepsilon'_{m,t},$$
*for all $x, x' \in \mathcal{X}_t, t \in [T]$.*

The detailed proof of Theorem 1 and all other missing proofs are given in the Appendix. Note that as long as the width $m$ of the NN is large enough (i.e., if the conditions on $m$ in (8) are satisfied), we have that $\varepsilon'_{m,t} = \mathcal{O}(1/T)$. Theorem 1 ensures that when using our trained NN $h$ to estimate the latent reward function $f$, the estimation error of the reward difference between any pair of arms is upper-bounded. Of note, it is reasonable that our confidence ellipsoid in Theorem 1 is in terms of the *difference* between reward values, because the only observations we receive are pairwise comparisons. Now, we state the regret upper bounds of our proposed algorithms.

**Theorem 2** (**NDB-UCB**). *Let $\lambda > \kappa_\mu$, $B$ be a constant such that $\sqrt{2\mathbf{h}^\top \mathbf{H}^{-1} \mathbf{h}} \le B$, and $c_0 > 0$ be an absolute constant such that $\frac{1}{m} \|\varphi(x) - \varphi(x')\|_2^2 \le c_0, \forall x, x' \in \mathcal{X}_t, t \in [T]$. For $m \ge poly(T, L, K, 1/\kappa_\mu, L_\mu, 1/\lambda_0, 1/\lambda, \log(1/\delta))$, then with probability of at least $1 - \delta$, we have*

$$\mathfrak{R}_T \le \frac{3}{2} \left( \beta_T + B\sqrt{\frac{\lambda}{\kappa_\mu}} + 1 \right) \sqrt{T 2 c_0 \widetilde{d}} + 1 = \widetilde{O} \left( \left( \frac{\sqrt{\widetilde{d}}}{\kappa_\mu} + B\sqrt{\frac{\lambda}{\kappa_\mu}} \right) \sqrt{T \widetilde{d}} \right).$$

---

[2] The effective dimension in Zhou et al. (2020) and Zhang et al. (2021) is defined using $\mathbf{H}$: $\widetilde{d}' = \log \det (\mathbf{H}/\lambda + \mathbf{I}) / \log(1 + TK/\lambda)$. However, it is of the same order (up to log factors) as $\log \det \left( \widetilde{\mathbf{H}}/\lambda + \mathbf{I} \right)$, with $\widetilde{\mathbf{H}} \doteq \sum_{s=1}^T \sum_{i=1}^K g(x_{s,i}; \theta_0) g(x_{s,i}; \theta_0)^\top / m$ (see Lemma B.7 of Zhang et al. (2021)).

The detailed requirements on the width $m$ of the NN are given by Eq. (8) in Appendix A.

**Theorem 3 (NDB-TS).** *Let $c_1$ and $c_2$ be two constants. Under the same conditions as those in Theorem 2, then with probability of at least $1 - \delta$, we have*

$$\mathfrak{R}_T = \widetilde{O}\left(\left(\frac{\sqrt{\widetilde{d}}}{\kappa_\mu} + B\sqrt{\frac{\lambda}{\kappa_\mu}}\right)\sqrt{T\widetilde{d}}\right).$$

Note that in terms of asymptotic dependencies (ignoring the log factors), our UCB- and TS- algorithms have the same growth rates. The upper regret bounds also hold for weak cumulative regret as it is upper-bounded by average cumulative regret.

When we assume that the effective dimension $\widetilde{d} = \widetilde{o}(\sqrt{T})$ (which is analogous to the assumption on the minimum eigenvalue from Bengs et al. (2022)), then the regret upper bounds for both **NDB-UCB** and **NDB-TS** are sub-linear. The dependence of our regret bounds on $\frac{1}{\kappa_\mu}$ and $L_\mu$ (i.e., the parameters of the link function defined in Assumption 1) are consistent with the previous works on generalized linear bandits (Li et al., 2017) and linear dueling bandits Bengs et al. (2022).

Compared with the regret upper bounds of NeuralUCB Zhou et al. (2020) and NeuralTS Zhang et al. (2021), the effective dimension $\widetilde{d}$ in Theorem 2 and Theorem 3 are expected to be larger than the effective dimension $\widetilde{d}'$ from Zhou et al. (2020); Zhang et al. (2021) because our $\widetilde{d}$ results from the summation of a significantly larger number of contexts. Therefore, our regret upper bounds (Theorem 2 and Theorem 3) are expected to be worse than that of Zhou et al. (2020); Zhang et al. (2021): $\widetilde{O}(\widetilde{d}'\sqrt{T})$. This downside can be attributed to the difficulty of our neural dueling bandits setting, in which we can only access preference feedback.

### 3.4 Proof Sketch

In this section, we give a brief proof sketch of our regret analysis in Section 3.3.

#### 3.4.1 Sketch of Proof for Theorem 1

We start by presenting an outline for the proof of our Theorem 1. To begin with, by applying the theory of the NTK (Jacot et al., 2018) to our NN for reward estimation (Section 3.1), we show that as long as the NN is wide enough, its output can be approximated by a linear function (see Appendix A.1 for details). More specifically, for a pair of arms $x$ and $x'$, the difference between their predicted latent reward values $h(x; \theta_t) - h(x'; \theta_t)$ can be approximated by the difference between the linear approximations of the NN:

**Lemma 1.** *Let $\varepsilon'_{m,t} \triangleq C_2 m^{-1/6}\sqrt{\log m}L^3\left(\frac{t}{\lambda}\right)^{4/3}$ where $C_2 > 0$ is an absolute constant. Then*

$$|\langle \varphi(x) - \varphi(x'), \theta_t - \theta_0\rangle - (h(x; \theta_t) - h(x'; \theta_t))| \leq 2\varepsilon'_{m,t}, \ \ \forall t \in [T], x, x' \in \mathcal{X}_t.$$

Of note, the approximation error $\varepsilon'_{m,t}$ becomes smaller as the width $m$ of the NN is increased.

**Proof of the Confidence Ellipsoid.**    Next, we derive the confidence ellipsoid for our algorithms, which is formalized by Lemma 6 (see its detailed proof in Appendix A.2). Lemma 6 allows us to derive the following inequality, which shows that for a pair of arms $x$ and $x'$, the difference between their latent reward values $f(x) - f(x')$ can be approximated by the difference between their predicted values by the linearized NN:

$$|f(x) - f(x') - \langle\varphi(x) - \varphi(x'), \theta_t - \theta_0\rangle| \leq \left\|\frac{1}{\sqrt{m}}(\varphi(x) - \varphi(x'))\right\|_{V_{t-1}^{-1}}\left(\beta_T + B\sqrt{\frac{\lambda}{\kappa_\mu}} + 1\right), \ (5)$$

Importantly, Eq. (5) guarantees that our trained NN with parameters $\theta_t$ (after linearization) is an accurate approximation of the latent reward function $f$ in terms of pairwise differences. This crucial theoretical guarantee is achieved thanks to our carefully designed loss function Eq. (1). Specifically, a key step in the proof of our confidence ellipsoid (i.e., Lemma 6) is the following equality:

$$\frac{1}{m}\sum_{s=1}^{t-1}(\mu(h(x_{s,1}; \theta_t) - h(x_{s,2}; \theta_t)) - y_s)(g(x_{s,1}; \theta_t) - g(x_{s,2}; \theta_t)) + \lambda(\theta_t - \theta_0) = 0, \ \ (6)$$

which is achieved by obtaining the NN parameters $\theta_t$ by minimizing our loss function Eq. (1). This is, in fact, analogous to using maximum likelihood to estimate the unknown vector $\theta$ in the linear reward function in linear dueling bandits (Bengs et al., 2022) and linear bandits with binary feedback (Li et al., 2017). In addition, another important step in the proof of Lemma 6 is the recognition that the observation noise $\varepsilon_t = y_t - \mu(f(x_{t,1}) - f(x_{t,2}))$ is conditionally 1-sub-Gaussian (more details can be found in the proof of Lemma 8 in Appendix A.2). After the proof of the confidence ellipsoid as described above, we then combine Eq. (5) and Lemma 1 to derive an upper bound on the difference between $f(x) - f(x')$ and $h(x; \theta_t) - h(x'; \theta_t)$, which completes the proof of Theorem 1.

### 3.4.2   SKETCH OF PROOF FOR THEOREM 2 AND THEOREM 3

To obtain a regret upper bound for the **NDB-UCB** algorithm (i.e., to prove Theorem 2), we firstly use Theorem 1 to derive an upper bound on the average instantaneous regret $r_t^a = f(x_t^\star) - (f(x_{t,1}) + f(x_{t,2}))/2$ (see the detailed proof in Appendix A.3):

$$r_t^a \leq \frac{3}{2}\left(\beta_T + B\sqrt{\frac{\lambda}{\kappa_\mu}} + 1\right)\sqrt{\frac{\kappa_\mu}{\lambda}}\sigma_{t-1}(x_{t,1}, x_{t,2}) + 3\varepsilon'_{m,t}. \tag{7}$$

Our strategies to select the two arms $x_{t,1}$ and $x_{t,2}$ (i.e., Eq. (2) and Eq. (3)) are essential for deriving the upper bound in Eq. (7). Subsequently, we follow similar steps to the analysis of the kernelized bandit algorithms (e.g., GP-UCB (Srinivas et al., 2010)) to obtain an upper bound on $\sum_{t=1}^{T}\sigma_{t-1}(x_{t,1}, x_{t,2})$ in terms of $\widetilde{d}$. This then provides us with an upper bound on the average regret $\mathfrak{R}_T^a = \sum_{t=1}^{T} r_t^a$ (as well as the weak regret $\mathfrak{R}_T^w$ since $\mathfrak{R}_T^w \leq \mathfrak{R}_T^a$) and hence completes the proof of Theorem 2. The proof of Theorem 3 for the NDB-TS algorithm is also based on Theorem 1 and largely follows from the analysis of the GP-TS algorithm (Chowdhury & Gopalan, 2017). However, non-trivial modifications need to be made in order to incorporate our strategies to select the two arms $x_{t,1}$ and $x_{t,2}$. The detailed regret analysis for Theorem 2 and Theorem 3 can be found in Appendix A.3 and Appendix B, respectively.

## 4   FURTHER IMPLICATIONS OF OUR ALGORITHMS AND THEORY

Our results to learn the latent non-linear reward function using preference feedback is not only limited to dueling bandits, but it can be adapted to other settings as discussed in the following sections.

### 4.1   NEURAL CONTEXTUAL BANDITS WITH BINARY FEEDBACK

In many real-world applications, the learner can only observe binary feedback, e.g., whether a user clicks in an online recommendation system or whether a treatment is effective in clinical trials, and these problems are commonly modeled as contextual GLM bandits (Filippi et al., 2010; Li et al., 2017; Faury et al., 2020), which assume the probability of success (i.e., 1) is proportional to exponential of its reward that is a linear function of action-context features. However, in many real-life applications, the reward can be a non-linear function of action-context features, which can modeled using a suitable neural network. We refer to this problem as neural contextual bandits with binary feedback. Therefore, our results can be extended to the neural contextual bandit problem, where the learner only receives binary feedback for the selected arms, which depends on the non-linear reward function. This adaption is not straightforward as the learner selects only one arm and observes binary feedback (success or failure) for the selected action. This difference leads to different arm selection strategies as only one needs to be selected in neural contextual bandits, and different loss functions must be optimized to train the neural network to estimate the unknown reward functions. The main challenge is to derive the confidence ellipsoid result, which is a crucial component for proving the regret bounds. While some ideas are common, the derivations differ due to the underlying differences in the problem settings. We have given more details of the neural contextual bandit with binary feedback problem and our regret bounds of algorithms proposed for this setting in Appendix D.

### 4.2   REINFORCEMENT LEARNING FROM HUMAN FEEDBACK

Our algorithms and theoretical results can also provide insights on the celebrated reinforcement learning with human feedback (RLHF) algorithm (Chaudhari et al., 2024), which has been the most

widely used method for aligning large language models (LLMs). Firstly, our Theorem 1 provides a theoretical guarantee on the quality of the learned reward model during RLHF. This is because Theorem 1 serves as an upper bound on the estimation error of the estimated reward differences between any pair of prompts. Secondly, our proposed algorithms can be used to improve the quality of the human preference dataset for RLHF. That is when collecting the dataset for RLHF, we can let the LLM generate a large number of responses, from which we can use our algorithms for neural dueling bandits to select two responses (corresponding to two arms) to be shown to the user for preference feedback. More details can be found in Appendix E.

## 5 EXPERIMENTS

To corroborate our theoretical results, we empirically demonstrate the performance of our proposed algorithms on different synthetic reward functions. We adopt the following two synthetic functions from earlier work on neural bandits (Zhou et al., 2020; Zhang et al., 2021; Dai et al., 2023): $f(x) = 10(x^\top\theta)^2$ (Square) and $f(x) = \cos(3x^\top\theta)$ (Cosine). We repeat all our experiments 20 times and show the average and weak cumulative regret with a 95% confidence interval (represented by the vertical line on each curve) Due to space constraints, additional results are given in the Appendix C.

The details of our experiments are as follows: We use a $d$-dimensional space to generate the sample features of each context-arm pair. We denote the context-arm feature vector for context $c_t$ and arm $a$ by $x_t^a$, where $x_t^a = \left(x_{t,a}^1, \ldots, x_{t,a}^d\right)$, $\forall\, t \geq 1$. The value of $i$-the element of $x_t^a$ vector is sampled uniformly at random from $(-1, 1)$. Note that the number of arms remains the same across the rounds, i.e., $K$. We then select a $d$-dimensional vector $\theta$ by sampling uniformly at random from $(-1, 1)^d$. In all our experiments, the binary preference feedback about $x_1$ being preferred over $x_2$ (or binary feedback in Appendix D) is sampled from a Bernoulli distribution with parameter $p = \mu\left(f(x_1) - f(x_2)\right)$. In all our experiments, we use a NN with 2 hidden layers with width 50, $\lambda = 1.0$, $\delta = 0.05$, $d = 5$, $K = 5$, and fixed value of $\nu_T = \nu = 1.0$. For having a fair comparison, we choose the value of $\nu$ after doing a hyperparameter search over set $\{10.0, 5.0, 1.0, 0.1, 0.01, 0.001, 0.0\}$ for linear baselines, i.e., LinDB-UCB and LinDB-TS. As shown in Fig. 1, the average cumulative regret is minimum for $\nu = 1.0$. Note that we did not perform any hyperparameter search for **NDB-UCB** and **NDB-TS**, whose performance can be further improved by doing the hyperparameter search.

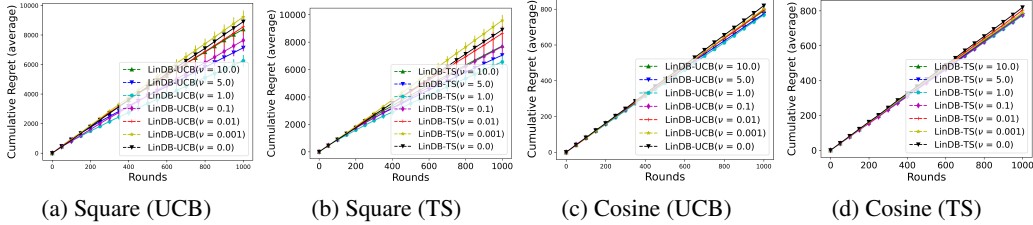

(a) Square (UCB)  (b) Square (TS)  (c) Cosine (UCB)  (d) Cosine (TS)

Figure 1: Average cumulative regret of LinDB-UCB and LinDB-TS vs. different values of $\nu$ for Square reward function ($i.e., 10(x^\top\theta)^2$).

As supported by the neural tangent kernel (NTK) theory, we can substitute the initial gradient $g(x; \theta_0)$ for the original feature vector $x$ as $g(x; \theta_0)$ represents the random Fourier features for the NTK Jacot et al. (2018). In this paper, we use the feature vectors $g(x; \theta_t)$ instead of $g(x; \theta_0)$ and recompute all $g(x; \theta_t)$ in each round for all past context-arm pairs. Additionally, compared to NTK theory, we have designed our algorithm to be more practical by adhering to the common practices in neural bandits Zhou et al. (2020); Zhang et al. (2021). Specifically, in the loss function for training our NN (as defined in Eq. (1)), we replaced the theoretical regularization parameter $\frac{1}{2}m\lambda\|\theta - \theta_0\|_2^2$ (where $m$ is the width of the NN) with the simpler $\lambda\|\theta\|_2^2$. Similarly, for the random features of the NTK, we replaced the theoretical $\frac{1}{\sqrt{m}}g(x; \theta_t)$ with $g(x; \theta_t)$. We retrain the NN after every 20 rounds and set the number of gradient steps to 50 in all our experiments.

**Regret comparison with baselines.** We compare regret (average/weak of our proposed algorithms with three baselines: LinDB-UCB (adapted from (Saha, 2021)), LinDB-TS, and CoLSTIM (Bengs et al., 2022). LinDB-UCB and LinDB-TS can be treated as variants of **NDB-UCB** and **NDB-TS**,

respectively, in which a linear function approximates the reward function. As expected, **NDB-UCB** and **NDB-TS** outperform all linear baselines as these algorithms cannot estimate the non-linear reward function and hence incur linear regret. For a fair comparison, we used the best hyperparameters of LinDB-UCB and LinDB-TS (see Fig. 1) for **NDB-UCB** and **NDB-TS**. We observe the same trend for different non-linear reward functions (see Fig. 5 and Fig. 6 in Appendix C) and also for neural contextual bandit with binary feedback (see Fig. 7 in Appendix D).

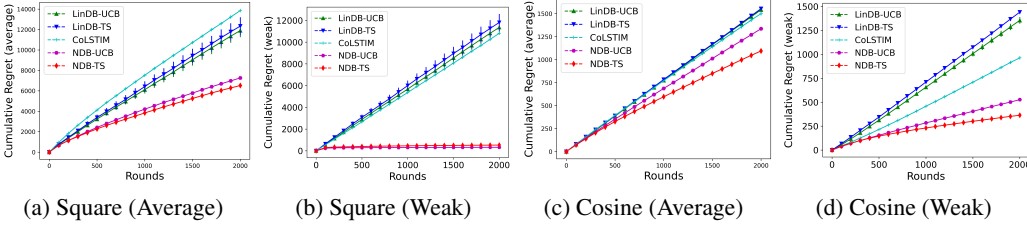

(a) Square (Average)    (b) Square (Weak)    (c) Cosine (Average)    (d) Cosine (Weak)

Figure 2: Comparisons of cumulative regret (average and weak) of different dueling bandits algorithms for non-linear reward functions: Square $(10(x^\top\theta)^2)$ and Cosine $(cos(3x^\top\theta))$.

**Varying dimension and arms vs. regret** Increasing the number of arms ($K$) and the dimension of the context-arm feature vectors ($d$) makes the problem more difficult. To see how increasing $K$ and $d$ affects the regret of our proposed algorithms, we vary the $K = \{5, 10, 15, 20, 25\}$ and $d = \{5, 10, 15, 20, 25\}$ while keeping the other problem parameters constant. As expected, the regret of **NDB-UCB** increases with increase in $K$ and $d$ as shown in Fig. 3. We also observe the same behavior for **NDB-TS** as shown Fig. 4. All missing figures from this section are in the Appendix C.

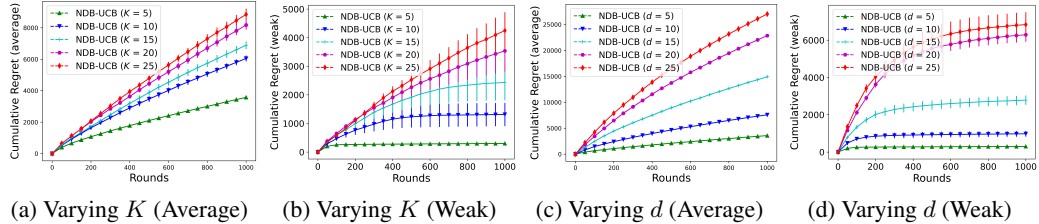

(a) Varying $K$ (Average)    (b) Varying $K$ (Weak)    (c) Varying $d$ (Average)    (d) Varying $d$ (Weak)

Figure 3: Cumulative regret (average and weak) of **NDB-UCB** vs. different number of arms ($K$) and dimension of the context-arm feature vector ($d$) for Square reward function ($i.e., 10(x^\top\theta)^2$).

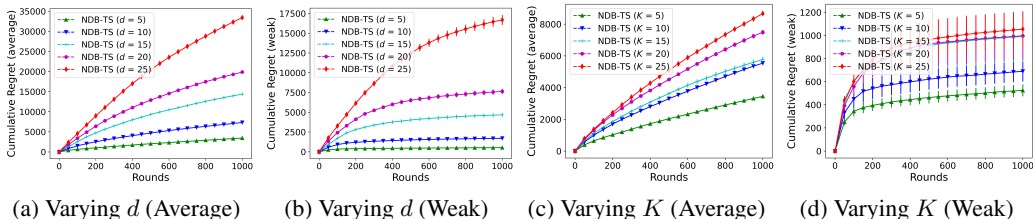

(a) Varying $d$ (Average)    (b) Varying $d$ (Weak)    (c) Varying $K$ (Average)    (d) Varying $K$ (Weak)

Figure 4: Cumulative regret (average and weak) of **NDB-TS** vs. different number of arms ($K$) and dimension of the context-arm feature vector ($d$) for Square reward function ($i.e., 10(x^\top\theta)^2$).

**Practical challenges.** Applying algorithms in scenarios requiring fast online interactions can be challenging for two reasons: updating the neural network (NN) and the arm-selection strategy. To address the first challenge, we can use a batched version of our algorithm, where the NN is updated only after a fixed interval. To address the second challenge, we can use an $\varepsilon_t$-greedy method for selecting arms when computing optimistic values using UCB or TS is not computationally feasible.

**Computational resources used for all experiments.** All the experiments are run on a server with AMD EPYC 7543 32-Core Processor, 256GB RAM, and 8 GeForce RTX 3080.

## 6 RELATED WORK

We now review the relevant work, especially in dueling bandits and contextual bandits, to our problem.

**Finite-Armed Dueling Bandits.** Learning from pairwise or $K$-wise comparisons has been thoroughly explored in the literature. In the context of finite-armed dueling bandits, the learner only observes a pairwise preference between two selected arms, and the goal is to find the best arm (Yue & Joachims, 2009; 2011; Yue et al., 2012). In dueling bandit literature, different criteria (e.g., Borda winner, Condorcet winner, Copeland winner, or von Neumann winner) are used to find the best arm while focusing on minimizing the regret only using preference feedback (Zoghi et al., 2014b; Ailon et al., 2014; Zoghi et al., 2014a; Komiyama et al., 2015; Gajane et al., 2015; Saha & Gopalan, 2018; 2019a;b; Zhu et al., 2023). We refer the readers to (Bengs et al., 2021) for a detailed survey on various works covering different settings of dueling bandits.

**Contextual Bandits.** Many real-life applications in online recommendation, advertising, web search, and e-commerce can be modeled as contextual bandits (Slivkins, 2019; Lattimore & Szepesvári, 2020). In the contextual bandit setting, a learner receives a context (information before selecting an arm), selects an action for that context, and then receives a reward for the selected action. To deal with the large (or infinite) number of context-action pairs, the mean reward of each action is assumed to be parameterized by an unknown function of action-context features, e.g., linear (Li et al., 2010; Chu et al., 2011; Abbasi-Yadkori et al., 2011; Agrawal & Goyal, 2013b), GLM (Filippi et al., 2010; Li et al., 2017; Jun et al., 2017), and non-linear (Valko et al., 2013; Chowdhury & Gopalan, 2017; Zhou et al., 2020; Zhang et al., 2021). We adopt the simplest neural contextual bandit algorithms for dealing with non-linear rewards, i.e., NeuralUCB (Zhou et al., 2020) and NeuralTS (Zhang et al., 2021) to our setting, as also done in earlier works (Dai et al., 2023; Lin et al., 2023). Since NeuralUCB or NeuralTS primarily influences the arm selection strategy, we can incorporate any variants of these algorithms by making appropriate modifications to Assumption 2 and Theorem 1. These adoptions can be challenging because of our setting solely relies on pairwise comparisons.

**Contextual Dueling Bandits.** The closest work to ours is contextual dueling bandits (Saha, 2021; Saha & Krishnamurthy, 2022; Bengs et al., 2022; Di et al., 2023; Li et al., 2024). Specifically, Saha (2021); Saha & Krishnamurthy (2022) proposed algorithms for contextual linear dueling bandits with pairwise and subset-wise preference feedback and established lower bounds for contextual preference bandits using a logistic link function. Whereas Bengs et al. (2022) generalized the setting to the contextual linear stochastic transitivity model, Di et al. (2023) considered the variance-aware algorithm, and Li et al. (2024) proposed an algorithm based on Feel-Good Thompson Sampling (Zhang, 2022). However, there are two key differences: non-linear reward function and arm-selection strategy. Existing work only considers the linear reward function, which may not be practical in many real-life applications. Our work fills this gap in the literature and generalizes the existing setting by considering the non-linear reward function in contextual dueling bandits. Furthermore, the existing works use different ways to select the pair of arms, leading to different arm-selection strategies than ours. Due to the differences in arm-selection strategy and non-linear reward function (which is estimated using an NN), our regret analysis is completely different than existing works.

## 7 CONCLUSION

Due to their prevalence in many real-life applications, from online recommendations to ranking web search results, we consider contextual dueling bandit problems that can have a complex and non-linear reward function. We used a neural network to estimate this reward function using human preference feedback observed for the previously selected arms. We proposed upper confidence bound- and Thompson sampling-based algorithms with sub-linear regret guarantees for contextual dueling bandits. Experimental results using synthetic functions corroborate our theoretical results. Our algorithms and theoretical results can also provide insights into the celebrated reinforcement learning with human feedback (RLHF) algorithm, such as a theoretical guarantee on the quality of the learned reward model. We also extend our results to contextual bandit problems with binary feedback, which is in itself a non-trivial contribution. A limitation of our work is that we currently do not account for problems where multiple arms are selected simultaneously (multi-way preference), which is an interesting future direction. Another future topic is to apply our algorithms to important real-world problems involving preference or binary feedback, e.g., LLM alignment using human feedback.

## ETHICS STATEMENT

The contributions of our work are primarily theoretical. Therefore, we do not foresee any immediate negative societal impact in the short term. Regarding our longer-term impacts, as discussed in Appendix E, our algorithms can be potentially adopted to improve online RLHF. On the positive side, our work can lead to better and more efficient alignment of LLMs through improved online RLHF, which could benefit society. On the other hand, the potential negative societal impacts arising from RLHF may also apply to our work. On the other hand, potential mitigation measures to prevent the misuse of RLHF would also help safeguard the potential misuse of our algorithms.

## ACKNOWLEDGEMENTS

This research is supported by the National Research Foundation (NRF), Prime Minister's Office, Singapore under its Campus for Research Excellence and Technological Enterprise (CREATE) programme. The Mens, Manus, and Machina (M3S) is an interdisciplinary research group (IRG) of the Singapore MIT Alliance for Research and Technology (SMART) centre. DesCartes: this research is supported by the National Research Foundation, Prime Minister's Office, Singapore under its Campus for Research Excellence and Technological Enterprise (CREATE) programme.

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

## A  Theoretical Analysis for NDB-UCB

Let $\{x_{(n)}\}_{n=1}^{TK}$ be a set of all possible context-arm feature vectors: $\{x_{t,a}\}_{1 \le t \le T, 1 \le a \le K}$, where $n = K(t-1) + a$. Define

$$\widetilde{\mathbf{H}}_{p,q}^{(1)} = \mathbf{\Sigma}_{p,q}^{(1)} = \langle x_{(p)}, x_{(q)} \rangle, \mathbf{A}_{p,q}^{(l)} = \begin{pmatrix} \mathbf{\Sigma}_{p,q}^{(l)} & \mathbf{\Sigma}_{p,q}^{(l)} \\ \mathbf{\Sigma}_{p,q}^{(l)} & \mathbf{\Sigma}_{q,q}^{(l)} \end{pmatrix},$$

$$\mathbf{\Sigma}_{p,q}^{(l+1)} = 2\mathbb{E}_{(u,v) \sim \mathcal{N}(0, \mathbf{A}_{p,q}^{(l)})}[\max\{u, 0\} \max\{v, 0\}],$$

$$\widetilde{\mathbf{H}}_{p,q}^{(l+1)} = 2\widetilde{\mathbf{H}}_{p,q}^{(l)} \mathbb{E}_{(u,v) \sim \mathcal{N}(0, \mathbf{A}_{p,q}^{(l)})}[\mathbb{K}(u \ge 0)\mathbb{K}(v \ge 0)] + \mathbf{\Sigma}_{p,q}^{(l+1)}.$$

Then, $\mathbf{H} = (\widetilde{\mathbf{H}}^{(L)} + \mathbf{\Sigma}^{(L)})/2$ is called the neural tangent kernel (NTK) matrix on the set of context-arm feature vectors $\{x_{(n)}\}_{n=1}^{TK}$.

Next, we first list the specific conditions we need for the width $m$ of the NN:

$$
\begin{aligned}
m &\ge CT^4 K^4 L^6 \log(T^2 K^2 L/\delta)/\lambda_0^4, \\
m(\log m)^{-3} &\ge C\kappa_\mu^{-3} T^8 L^{21} \lambda^{-5}, \\
m(\log m)^{-3} &\ge C\kappa_\mu^{-3} T^{14} L^{21} \lambda^{-11} L_\mu^6, \\
m(\log m)^{-3} &\ge CT^{14} L^{18} \lambda^{-8},
\end{aligned}
\tag{8}
$$

for some absolute constant $C > 0$. To ease exposition, we express these conditions above as $m \ge \text{poly}(T, L, K, 1/\kappa_\mu, L_\mu, 1/\lambda_0, 1/\lambda, \log(1/\delta))$.

To simplify exposition, we use an error probability of $\delta$ for all probabilistic statements. Our final results hold naturally by taking a union bound over all required $\delta$'s. The lemma below shows that the ground-truth utility function $f$ can be expressed as a linear function.

**Lemma 2** (Lemma B.3 of Zhang et al. (2021)). *As long as the width $m$ of the NN is wide enough:*

$$m \ge C_0 T^4 K^4 L^6 \log(T^2 K^2 L/\delta)/\lambda_0^4,$$

*then with probability of at least $1 - \delta$, there exits a $\theta_f$ such that*

$$f(x) = \langle g(x; \theta_0), \theta_f - \theta_0 \rangle, \sqrt{m} \|\theta_f - \theta_0\|_2 \le \sqrt{2\mathbf{h}^\top \mathbf{H}^{-1} \mathbf{h}} \le B.$$

*for all $x \in \mathcal{X}_t, \forall t \in [T]$.*

Let $y_s = \mathbb{1}(x_{s,1} \succ x_{s,2})$, then we can write $\mathbb{P}(y_s = 1) = \mu(h(x_{s,1}; \theta) - h(x_{s,2}; \theta))$ and $\mathbb{P}(y_s = 0) = 1 - \mu(h(x_{s,1}; \theta) - h(x_{s,2}; \theta)) = \mu(h(x_{s,2}; \theta) - h(x_{s,1}; \theta))$.

### A.1  Theoretical Guarantee about the Neural Network

The following lemma gives an upper bound on the distance between $\theta_t$ and $\theta_0$:

**Lemma 3.** *We have that $\|\theta_t - \theta_0\|_2 \le 2\sqrt{\frac{t}{m\lambda}}, \forall t \in [T]$.*

*Proof.* As $\mu(\cdot) \in [0, 1]$, then using Eq. (1) gives us

$$
\begin{aligned}
\frac{1}{2}\lambda \|\theta_t - \theta_0\|_2^2 &\le \mathcal{L}_t(\theta_t) \le \mathcal{L}_t(\theta_0) \\
&= -\frac{1}{m} \sum_{s=1}^{t-1} \Big[ \mathbb{1}_{x_{t,1} \succ x_{t,2}} \log \mu(h(x_{s,1}; \theta_0) - h(x_{s,2}; \theta_0)) + \\
&\quad (1 - \mathbb{1}_{x_{t,1} \succ x_{t,2}}) \log \mu(h(x_{s,2}; \theta_0) - h(x_{s,1}; \theta_0)) \Big] + \frac{1}{2}\lambda \|\theta_0 - \theta_0\|_2^2
\end{aligned}
$$

$$\stackrel{(a)}{=} -\frac{1}{m} \sum_{s=1}^{t-1} \left[ \mathbb{1}_{x_{t,1} \succ x_{t,2}} \log \mu(0) + (1 - \mathbb{1}_{x_{t,1} \succ x_{t,2}}) \log \mu(0) \right]$$

$$= -\frac{1}{m} \sum_{s=1}^{t-1} \log 0.5$$

$$\leq \frac{1}{m} t(-\log 0.5)$$

$$\stackrel{(b)}{\leq} \frac{t}{m}.$$

Step $(a)$ follow because $h(x; \theta_0) = 0, \forall x \in \mathcal{X}, t \in [T]$ which is ensured by Assumption 2, step $(b)$ follows because $-\log 0.5 < 1$. Therefore, we have that $\|\theta_t - \theta_0\|_2 \leq \sqrt{2\frac{t}{m\lambda}} \leq 2\sqrt{\frac{t}{m\lambda}}$. $\qquad\square$

Now, Lemma 3 allows us to obtain the following lemmas regarding the gradients of the NN.

**Lemma 4.** *Let* $\tau = 2\sqrt{\frac{t}{m\lambda}}$. *Then for absolute constants* $C_3, C_1 > 0$, *with probability of at least* $1 - \delta$,

$$\|g(x; \theta_t)\|_2 \leq C_3 \sqrt{mL},$$

$$\|g(x; \theta_0) - g(x; \theta_t)\|_2 \leq C_1 \sqrt{m \log m} \tau^{1/3} L^{7/2} = C_1 m^{1/3} \sqrt{\log m} \left(\frac{t}{\lambda}\right)^{1/3} L^{7/2},$$

*for all* $x \in \mathcal{X}_t, t \in [T]$.

*Proof.* It can be easily verified that our $\tau = 2\sqrt{\frac{t}{m\lambda}}$ satisfies the requirement on $\tau$ specified in Lemmas B.5 and B.6 from Zhang et al. (2021). Therefore, the results from Lemmas B.5 and B.6 from Zhang et al. (2021) are applicable for $\theta_t$ because our Lemma 3 guarantees that $\|\theta_t - \theta_0\|_2 \leq \tau$. $\qquad\square$

In addition, Lemmas B.4 from Zhou et al. (2020) allows us to obtain the following lemma, which shows that the output of the NN can be approximated by its linearization.

**Lemma 5.** *Let* $\tau \triangleq 2\sqrt{\frac{t}{m\lambda}}$. *Let* $\varepsilon'_{m,t} \triangleq C_2 m^{-1/6} \sqrt{\log m} L^3 \left(\frac{t}{\lambda}\right)^{4/3}$. *Then for some absolute constant* $C_2 > 0$, *with probability of at least* $1 - \delta$,

$$|h(x; \theta_t) - \langle \theta_t - \theta_0, g(x; \theta_0) \rangle| \leq C_2 \tau^{4/3} L^3 \sqrt{m \log m} = C_2 m^{-1/6} \sqrt{\log m} L^3 \left(\frac{t}{\lambda}\right)^{4/3} = \varepsilon'_{m,t},$$

*for all* $x \in \mathcal{X}_t, t \in [T]$.

An immediate consequence of Lemma 5 is the following lemma.

**Lemma 1.** *Let* $\varepsilon'_{m,t} \triangleq C_2 m^{-1/6} \sqrt{\log m} L^3 \left(\frac{t}{\lambda}\right)^{4/3}$ *where* $C_2 > 0$ *is an absolute constant. Then*

$$|\langle \varphi(x) - \varphi(x'), \theta_t - \theta_0 \rangle - (h(x; \theta_t) - h(x'; \theta_t))| \leq 2\varepsilon'_{m,t}, \quad \forall t \in [T], x, x' \in \mathcal{X}_t.$$

*Proof.* By re-arranging the left-hand side and then using Lemma 5, we get

$$|\langle \varphi(x) - \varphi(x'), \theta_t - \theta_0 \rangle - (h(x; \theta_t) - h(x'; \theta_t))|$$
$$= |\langle \varphi(x), \theta_t - \theta_0 \rangle - h(x; \theta_t) + h(x'; \theta_t) - \langle \varphi(x'), \theta_t - \theta_0 \rangle|$$
$$\leq |\langle \varphi(x), \theta_t - \theta_0 \rangle - h(x; \theta_t)| + |h(x'; \theta_t) - \langle \varphi(x'), \theta_t - \theta_0 \rangle|$$
$$\leq 2C_2 m^{-1/6} \sqrt{\log m} L^3 \left(\frac{t}{\lambda}\right)^{4/3}$$
$$= 2\varepsilon'_{m,t}. \qquad\square$$

## A.2 Proof of Confidence Ellipsoid

In our next proofs, we denote $\varphi'_s \triangleq g(x_{s,1}; \theta_0) - g(x_{s,2}; \theta_0)$, $\widetilde{\varphi}'_s \triangleq g(x_{s,1}; \theta_t) - g(x_{s,2}; \theta_t)$, and $\widetilde{h}_{s,t} \triangleq h(x_{s,1}; \theta_t) - h(x_{s,2}; \theta_t)$. Recall that $p$ is the total number of parameters of the NN. We next prove the confidence ellipsoid for our algorithm, including Lemma 6 and Theorem 1 below.

**Lemma 6.** *Let $\beta_T \triangleq \frac{1}{\kappa_\mu} \sqrt{\widetilde{d} + 2\log(1/\delta)}$. Assuming that the conditions on $m$ from Eq. (8) are satisfied. With probability of at least $1 - \delta$, we have that*

$$\sqrt{m} \left\| \theta_f - \theta_t \right\|_{V_{t-1}} \leq \beta_T + B \sqrt{\frac{\lambda}{\kappa_\mu}} + 1, \qquad \forall t \in [T].$$

### A.2.1 Proof of Lemma 6

For any $\theta_{f'} \in \mathbb{R}^p$, define

$$G_t(\theta_{f'}) \triangleq \frac{1}{m} \sum_{s=1}^{t-1} \Big[ \mu \left( \langle \theta_{f'} - \theta_0, \varphi'_s \rangle \right) - \mu \left( \langle \theta_f - \theta_0, \varphi'_s \rangle \right) \Big] \varphi'_s + \lambda (\theta_{f'} - \theta_0). \tag{9}$$

We start by decomposing $\left\| \theta_f - \theta_t \right\|_{V_{t-1}}$ in terms of $G_t$ in the following lemma.

**Lemma 7.** *Choose $\lambda > 0$ such that $\lambda/\kappa_\mu > 1$. Define $V_{t-1} \triangleq \sum_{s=1}^{t-1} \varphi'_s \varphi'^\top_s \frac{1}{m} + \frac{\lambda}{\kappa_\mu} \mathbf{I}$.*

$$\left\| \theta_f - \theta_t \right\|_{V_{t-1}} \leq \frac{1}{\kappa_\mu} \left\| G_t(\theta_t) \right\|_{V_{t-1}^{-1}} + \sqrt{\frac{\lambda}{\kappa_\mu}} \frac{B}{\sqrt{m}}.$$

*Proof.* Let $\lambda' \in (0,1)$. For any $\theta_{f'_1}, \theta_{f'_2} \in \mathbb{R}^p$, setting $\theta_{\bar{f}} = \lambda' \theta_{f'_1} + (1-\lambda') \theta_{f'_2}$ and using mean-value theorem, we get:

$$G_t(\theta_{f'_1}) - G_t(\theta_{f'_2}) = \left[ \sum_{s=1}^{t-1} \frac{1}{m} \dot{\mu}(\langle \theta_{\bar{f}} - \theta_0, \varphi'_s \rangle) \varphi'_s \varphi'^\top_s + \lambda \mathbf{I}_p \right] (\theta_{f'_1} - \theta_{f'_2})$$

$$\geq \kappa_\mu \left[ \sum_{s=1}^{t-1} \varphi'_s \varphi'^\top_s \frac{1}{m} + \frac{\lambda}{\kappa_\mu} \mathbf{I}_p \right] (\theta_{f'_1} - \theta_{f'_2})$$

$$= \kappa_\mu V_{t-1} (\theta_{f'_1} - \theta_{f'_2}).$$

Note that $G_t(\theta_f) = \lambda (\theta_f - \theta_0)$. Let $f_t$ be the estimate of $f$ at the beginning of the iteration $t$ and $f_{t,s} = \langle \theta_t - \theta_0, \varphi'_s \rangle$. Now using the equation above,

$$\left\| G_t(\theta_t) - \lambda (\theta_f - \theta_0) \right\|^2_{V_{t-1}^{-1}} = \left\| G_t(\theta_f) - G_t(\theta_t) \right\|^2_{V_{t-1}^{-1}}$$

$$\geq (\kappa_\mu V_{t-1} (\theta_f - \theta_t))^\top V_{t-1}^{-1} \kappa_\mu V_{t-1} (\theta_f - \theta_t)$$

$$= \kappa_\mu^2 (\theta_f - \theta_t)^\top V_{t-1} V_{t-1}^{-1} V_{t-1} (\theta_f - \theta_t)$$

$$= \kappa_\mu^2 \left\| \theta_f - \theta_t \right\|^2_{V_{t-1}}.$$

This allows us to show that

$$\left\| \theta_f - \theta_t \right\|_{V_{t-1}} \leq \frac{1}{\kappa_\mu} \left\| G_t(\theta_t) - \lambda(\theta_f - \theta_0) \right\|_{V_{t-1}^{-1}} \leq \frac{1}{\kappa_\mu} \left\| G_t(\theta_t) \right\|_{V_{t-1}^{-1}} + \frac{1}{\kappa_\mu} \left\| \lambda (\theta_f - \theta_0) \right\|_{V_{t-1}^{-1}},$$

$$\tag{10}$$

in which we have made use of the triangle inequality.

Note that we choose $\lambda$ such that $\frac{\lambda}{\kappa_\mu} > 1$. This allows us to show that $V_{t-1} \succeq \frac{\lambda}{\kappa_\mu} I$ and hence $V_{t-1}^{-1} \preceq \frac{\kappa_\mu}{\lambda} I$. Recall that Lemma 2 tells us that $\|\theta_f - \theta_0\|_2 \leq \frac{B}{\sqrt{m}}$, which tells us that

$$
\begin{aligned}
\frac{1}{\kappa_\mu} \|\lambda(\theta_f - \theta_0)\|_{V_{t-1}^{-1}} &= \frac{\lambda}{\kappa_\mu} \sqrt{(\theta_f - \theta_0)^\top V_{t-1}^{-1} (\theta_f - \theta_0)} \\
&\leq \frac{\lambda}{\kappa_\mu} \sqrt{(\theta_f - \theta_0)^\top \frac{\kappa_\mu}{\lambda} (\theta_f - \theta_0)} \\
&\leq \sqrt{\frac{\lambda}{\kappa_\mu}} \|\theta_f - \theta_0\|_2 \\
&\leq \sqrt{\frac{\lambda}{\kappa_\mu}} \frac{B}{\sqrt{m}}.
\end{aligned}
\tag{11}
$$

Plugging Eq. (11) into Eq. (10) completes the proof. $\qquad\square$

Recall that we denote $y_s = \mu(f(x_{s,1}) - f(x_{s,2})) + \varepsilon_s$, in which $y_s$ is a binary observation and $\varepsilon_s$ can be seen as the observation noise. Next, we derive an upper bound on the first term from Lemma 7:

$$
\begin{aligned}
\frac{1}{\kappa_\mu} \|G_t(\theta_t)\|_{V_{t-1}^{-1}} &= \frac{1}{\kappa_\mu} \left\| \frac{1}{m} \sum_{s=1}^{t-1} \left[ \mu(\langle \theta_t - \theta_0, \varphi_s' \rangle) - \mu(\langle \theta_f - \theta_0, \varphi_s' \rangle) \right] \varphi_s' + \lambda(\theta_t - \theta_0) \right\|_{V_{t-1}^{-1}} \\
&= \frac{1}{\kappa_\mu} \left\| \frac{1}{m} \sum_{s=1}^{t-1} (\mu(f_{t,s}) - \mu(f(x_{s,1}) - f(x_{s,2}))) \varphi_s' + \lambda(\theta_t - \theta_0) \right\|_{V_{t-1}^{-1}} \\
&= \frac{1}{\kappa_\mu} \left\| \frac{1}{m} \sum_{s=1}^{t-1} (\mu(f_{t,s}) - (y_s - \varepsilon_s)) \varphi_s' + \lambda(\theta_t - \theta_0) \right\|_{V_{t-1}^{-1}} \\
&= \frac{1}{\kappa_\mu} \left\| \frac{1}{m} \sum_{s=1}^{t-1} (\mu(f_{t,s}) - y_s) \varphi_s' + \frac{1}{m} \sum_{s=1}^{t-1} \varepsilon_s \varphi_s' + \lambda(\theta_t - \theta_0) \right\|_{V_{t-1}^{-1}} \\
&\leq \frac{1}{\kappa_\mu} \left\| \frac{1}{m} \sum_{s=1}^{t-1} (\mu(f_{t,s}) - y_s) \varphi_s' + \lambda(\theta_t - \theta_0) \right\|_{V_{t-1}^{-1}} + \frac{1}{\kappa_\mu} \left\| \frac{1}{m} \sum_{s=1}^{t-1} \varepsilon_s \varphi_s' \right\|_{V_{t-1}^{-1}}.
\end{aligned}
\tag{12}
$$

Next, we derive an upper bound on the first term in Eq. (12). To simplify exposition, we define

$$
A_1 \triangleq \frac{1}{m} \sum_{s=1}^{t-1} \left( \mu(f_{t,s}) - y_s \right) \left( \varphi_s' - \widetilde{\varphi}_s' \right), \qquad A_2 \triangleq \frac{1}{m} \sum_{s=1}^{t-1} \left( \mu(f_{t,s}) - \mu(\widetilde{h}_{s,t}) \right) \widetilde{\varphi}_s'.
\tag{13}
$$

Now the first term in Eq. (12) can be decomposed as:

$$
\left\| \frac{1}{m} \sum_{s=1}^{t-1} \Big( \mu(f_{t,s}) - y_s \Big) \varphi_s' + \lambda(\theta_t - \theta_0) \right\|_{V_{t-1}^{-1}}
$$

$$
= \left\| \frac{1}{m} \sum_{s=1}^{t-1} \Big( \mu(f_{t,s}) - y_s \Big) \Big( \varphi_s' + \widetilde{\varphi}_s' - \widetilde{\varphi}_s' \Big) + \lambda(\theta_t - \theta_0) \right\|_{V_{t-1}^{-1}}
$$

$$
= \left\| \frac{1}{m} \sum_{s=1}^{t-1} \Big( \mu(f_{t,s}) - y_s \Big) \widetilde{\varphi}_s' + \lambda(\theta_t - \theta_0) + A_1 \right\|_{V_{t-1}^{-1}}
$$

$$
= \left\| \frac{1}{m} \sum_{s=1}^{t-1} \Big( \mu(f_{t,s}) + \mu(\widetilde{h}_{s,t}) - \mu(\widetilde{h}_{s,t}) - y_s \Big) \widetilde{\varphi}_s' + \lambda(\theta_t - \theta_0) + A_1 \right\|_{V_{t-1}^{-1}} \qquad (14)
$$

$$
= \left\| \frac{1}{m} \sum_{s=1}^{t-1} \Big( \mu(\widetilde{h}_{s,t}) - y_s \Big) \widetilde{\varphi}_s' + \lambda(\theta_t - \theta_0) + A_2 + A_1 \right\|_{V_{t-1}^{-1}}
$$

$$
\overset{(a)}{=} \left\| A_2 + A_1 \right\|_{V_{t-1}^{-1}}
$$

$$
\leq \| A_2 \|_{V_{t-1}^{-1}} + \| A_1 \|_{V_{t-1}^{-1}}
$$

$$
\leq \sqrt{\frac{\kappa_\mu}{\lambda}} \| A_2 \|_2 + \sqrt{\frac{\kappa_\mu}{\lambda}} \| A_1 \|_2 .
$$

Note that step $(a)$ above follows because

$$
\frac{1}{m} \sum_{s=1}^{t-1} \Big( \mu(\widetilde{h}_{s,t}) - y_s \Big) \widetilde{\varphi}_s' + \lambda(\theta_t - \theta_0)
$$

$$
= \frac{1}{m} \sum_{s=1}^{t-1} \Big( \mu(h(x_{s,1}; \theta_t) - h(x_{s,2}; \theta_t)) - y_s \Big) (g(x_{s,1}; \theta_t) - g(x_{s,2}; \theta_t)) + \lambda(\theta_t - \theta_0)
$$

$$
= 0,
$$

$$(15)$$

which is ensured by the way in which we train our NN (see Eq. (6)). Next, we derive an upper bound on the norm of $A_1$. To begin with, we have that

$$
\| \varphi_s' - \widetilde{\varphi}_s' \|_2 = \| g(x_{s,1}; \theta_0) - g(x_{s,2}; \theta_0) - g(x_{s,1}; \theta_t) + g(x_{s,2}; \theta_t) \|_2
$$

$$
\leq \| g(x_{s,1}; \theta_0) - g(x_{s,1}; \theta_t) \|_2 + \| g(x_{s,2}; \theta_0) - g(x_{s,2}; \theta_t) \|_2
$$

$$
\leq 2 C_1 m^{1/3} \sqrt{\log m} \left( \frac{Ct}{\lambda} \right)^{1/3} L^{7/2},
$$

in which the last inequality follows from Lemma 4. Now the norm of $A_1$ can be bounded as:

$$\|A_1\|_2 = \left\|\frac{1}{m}\sum_{s=1}^{t-1}\Big(\mu(f_{t,s}) - y_s\Big)\Big(\varphi_s' - \widetilde{\varphi}_s'\Big)\right\|_2$$

$$\leq \frac{1}{m}\sum_{s=1}^{t-1}\left\|\Big(\mu(f_{t,s}) - y_s\Big)\Big(\varphi_s' - \widetilde{\varphi}_s'\Big)\right\|_2$$

$$= \frac{1}{m}\sum_{s=1}^{t-1}|\mu(f_{t,s}) - y_s|\,\|\varphi_s' - \widetilde{\varphi}_s'\|_2 \tag{16}$$

$$\leq \frac{1}{m}\sum_{s=1}^{t-1}\|\varphi_s' - \widetilde{\varphi}_s'\|_2$$

$$\leq \frac{1}{m}\sum_{s=1}^{t-1}2C_1 m^{1/3}\sqrt{\log m}\left(\frac{t}{\lambda}\right)^{1/3}L^{7/2}$$

$$= m^{-2/3}\sqrt{\log m}\,t^{4/3}2C_1\lambda^{-1/3}L^{7/2}.$$

Next, we proceed to bound the norm of $A_2$. Let $\lambda' \in (0,1)$, and let $a_{t,s} = \lambda' f_{t,s} + (1-\lambda')\widetilde{h}_{s,t}$. Following the mean-value theorem, we have for some $\lambda'$ that

$$\mu(f_{t,s}) - \mu(\widetilde{h}_{s,t}) = (f_{t,s} - \widetilde{h}_{s,t})\dot{\mu}(a_{t,s}).$$

Note that $\dot{\mu}(a_{t,s}) \leq L_\mu$ which follows from our Assumption 1. This allows us to show that

$$|\mu(f_{t,s}) - \mu(\widetilde{h}_{s,t})| = |(f_{t,s} - \widetilde{h}_{s,t})\dot{\mu}(a_{t,s})|$$

$$= |f_{t,s} - \widetilde{h}_{s,t}||\dot{\mu}(a_{t,s})|$$

$$\leq L_\mu|f_{t,s} - \widetilde{h}_{s,t}|$$

$$= L_\mu|\langle\theta_t - \theta_0, g(x_{s,1};\theta_0)\rangle - \langle\theta_t - \theta_0, g(x_{s,2};\theta_0)\rangle - (h(x_{s,1};\theta_t) - h(x_{s,2};\theta_t))|$$

$$\leq L_\mu\left(\big|\langle\theta_t - \theta_0, g(x_{s,1};\theta_0)\rangle - h(x_{s,1};\theta_t)\big| + \big|h(x_{s,2};\theta_t) - \langle\theta_t - \theta_0, g(x_{s,2};\theta_0)\rangle\big|\right)$$

$$\leq L_\mu \times 2 \times C_2 m^{-1/6}\sqrt{\log m}L^3\left(\frac{t}{\lambda}\right)^{4/3}$$

$$= 2L_\mu C_2 m^{-1/6}\sqrt{\log m}L^3\left(\frac{t}{\lambda}\right)^{4/3}$$

in which we have used Lemma 5 in the last inequality. Also, Lemma 4 allows us to show that
$\|\widetilde{\varphi}_s'\|_2 = \|g(x_{s,1};\theta_t) - g(x_{s,2};\theta_t)\|_2 \leq \|g(x_{s,1};\theta_t)\|_2 + \|g(x_{s,2};\theta_t)\|_2 \leq 2C_3\sqrt{mL}$.

Now we can derive an upper bound on the norm of $A_2$:

$$\|A_2\|_2 = \left\|\frac{1}{m}\sum_{s=1}^{t-1}\Big(\mu(f_{t,s}) - \mu(\widetilde{h}_{s,t})\Big)\widetilde{\varphi}_s'\right\|_2$$

$$\leq \frac{1}{m}\sum_{s=1}^{t-1}\left\|(\mu(f_{t,s}) - \mu(\widetilde{h}_{s,t}))\widetilde{\varphi}_s'\right\|_2$$

$$= \frac{1}{m}\sum_{s=1}^{t-1}|\mu(f_{t,s}) - \mu(\widetilde{h}_{s,t})|\,\|\widetilde{\varphi}_s'\|_2 \tag{17}$$

$$\leq \frac{1}{m}\sum_{s=1}^{t-1}2L_\mu C_2 m^{-1/6}\sqrt{\log m}L^3\left(\frac{t}{\lambda}\right)^{4/3} \times 2C_3\sqrt{mL}$$

$$\leq 4L_\mu C_2 C_3 m^{-2/3}\sqrt{\log m}\,t^{7/3}L^{7/2}\lambda^{-4/3}.$$

Lastly, plugging Eq. (16) and Eq. (17) into Eq. (14), we can derive an upper bound on the first term in Eq. (12):

$$
\frac{1}{\kappa_\mu} \left\| \frac{1}{m} \sum_{s=1}^{t-1} (\mu(f_{t,s}) - y_s)\varphi'_s + \lambda(\theta_t - \theta_0) \right\|_{V_{t-1}^{-1}}
$$
$$
\leq \frac{1}{\sqrt{\kappa_\mu \lambda}} m^{-2/3} \sqrt{\log m} t^{4/3} 2C_1 \lambda^{-1/3} L^{7/2} + \tag{18}
$$
$$
\frac{1}{\sqrt{\kappa_\mu \lambda}} 4L_\mu C_2 C_3 m^{-2/3} \sqrt{\log m} t^{7/3} L^{7/2} \lambda^{-4/3}.
$$

Next, plugging equation Eq. (18) into equation Eq. (12), and plugging the results into Lemma 7, we have that

$$
\|\theta_f - \theta_t\|_{V_{t-1}}
$$
$$
\leq \frac{1}{\kappa_\mu \sqrt{m}} \left\| \sum_{s=1}^{t-1} \varepsilon_s \varphi'_s \frac{1}{\sqrt{m}} \right\|_{V_{t-1}^{-1}} + \sqrt{\frac{\lambda}{\kappa_\mu}} \frac{B}{\sqrt{m}} + \frac{1}{\sqrt{\kappa_\mu \lambda}} m^{-2/3} \sqrt{\log m} t^{4/3} 2C_1 \lambda^{-1/3} L^{7/2} +
$$
$$
\frac{1}{\sqrt{\kappa_\mu \lambda}} 4L_\mu C_2 C_3 m^{-2/3} \sqrt{\log m} t^{7/3} L^{7/2} \lambda^{-4/3}.
$$

Here we define

$$
\varepsilon_{m,t} \triangleq B\sqrt{\frac{\lambda}{\kappa_\mu}} + \frac{1}{\sqrt{\kappa_\mu \lambda}} m^{-1/6} \sqrt{\log m} t^{4/3} 2C_1 \lambda^{-1/3} L^{7/2} +
$$
$$
\frac{1}{\sqrt{\kappa_\mu \lambda}} 4L_\mu C_2 C_3 m^{-1/6} \sqrt{\log m} t^{7/3} L^{7/2} \lambda^{-4/3}. \tag{19}
$$

It is easy to verify that as long as the conditions on $m$ from Eq. (8) are satisfied (i.e., the width $m$ of the NN is large enough), we have that $\varepsilon_{m,t} \leq B\sqrt{\frac{\lambda}{\kappa_\mu}} + 1$.

This allows us to show that

$$
\sqrt{m} \|\theta_f - \theta_t\|_{V_{t-1}} \leq \frac{1}{\kappa_\mu} \left\| \sum_{s=1}^{t-1} \varepsilon_s \varphi'_s \frac{1}{\sqrt{m}} \right\|_{V_{t-1}^{-1}} + \varepsilon_{m,t}
$$
$$
\leq \frac{1}{\kappa_\mu} \left\| \sum_{s=1}^{t-1} \varepsilon_s \varphi'_s \frac{1}{\sqrt{m}} \right\|_{V_{t-1}^{-1}} + B\sqrt{\frac{\lambda}{\kappa_\mu}} + 1. \tag{20}
$$

Finally, in the next lemma, we derive an upper bound on the first term in Eq. (20).

**Lemma 8.** *Let $\beta_T \triangleq \frac{1}{\kappa_\mu} \sqrt{\tilde{d} + 2\log(1/\delta)}$. With probability of at least $1 - \delta$, we have that*

$$
\frac{1}{\kappa_\mu} \left\| \sum_{s=1}^{t-1} \varepsilon_s \varphi'_s \frac{1}{\sqrt{m}} \right\|_{V_{t-1}^{-1}} \leq \beta_T.
$$

*Proof.* To begin with, we derive an upper bound on the log determinant of the matrix $V_t \triangleq \sum_{s=1}^{t} \varphi'_s \varphi'^{\top}_s \frac{1}{m} + \frac{\lambda}{\kappa_\mu} \mathbf{I}$. Here we use $C_2^K$ to denote all possible pairwise combinations of the indices of $K$ arms. Here we denote $z_j^i(s) \triangleq \varphi(x_{s,i}) - \varphi(x_{s,j})$. Also recall we have defined in the main text

that $\mathbf{H}' \triangleq \sum_{s=1}^{T} \sum_{(i,j)\in C_2^K} z_j^i(s) z_j^i(s)^\top \frac{1}{m}$. Now the determinant of $V_t$ can be upper-bounded as

$$
\begin{aligned}
\det(V_t) &= \det\left( \sum_{s=1}^{t} (\varphi(x_{s,1}) - \varphi(x_{s,2})) (\varphi(x_{s,1}) - \varphi(x_{s,2}))^\top \frac{1}{m} + \frac{\lambda}{\kappa_\mu}\mathbf{I} \right) \\
&\leq \det\left( \sum_{s=1}^{T} (\varphi(x_{s,1}) - \varphi(x_{s,2})) (\varphi(x_{s,1}) - \varphi(x_{s,2}))^\top \frac{1}{m} + \frac{\lambda}{\kappa_\mu}\mathbf{I} \right) \\
&\leq \det\left( \sum_{s=1}^{T} \sum_{(i,j)\in C_2^K} z_j^i(s) z_j^i(s)^\top \frac{1}{m} + \frac{\lambda}{\kappa_\mu}\mathbf{I} \right) \\
&= \det\left( \mathbf{H}' + \frac{\lambda}{\kappa_\mu}\mathbf{I} \right).
\end{aligned}
\tag{21}
$$

Recall that in our algorithm, we have set $V_0 = \frac{\lambda}{\kappa_\mu}\mathbf{I}_p$. This leads to

$$
\begin{aligned}
\log \frac{\det V_t}{\det V_0} &\leq \log \frac{\det\left( \mathbf{H}' + \frac{\lambda}{\kappa_\mu}\mathbf{I} \right)}{\det V_0} \\
&= \log \frac{(\lambda/\kappa_\mu)^p \det\left( \frac{\kappa_\mu}{\lambda}\mathbf{H}' + \mathbf{I} \right)}{(\lambda/\kappa_\mu)^p} \\
&= \log \det\left( \frac{\kappa_\mu}{\lambda}\mathbf{H}' + \mathbf{I} \right).
\end{aligned}
\tag{22}
$$

We use $\varepsilon_s$ to denote the observation noise in iteration $s \in [T]$: $y_s = \mu(f(x_{s,1}) - f(x_{s,2})) + \varepsilon_s$. Let $\mathcal{F}_{t-1}$ denote the sigma algebra generated by history $\{(x_{s,1}, x_{s,2}, \varepsilon_s)_{s\in[t-1]}, x_{t,1}, x_{t,2}\}$. Here we justify that the sequence of noise $\{\varepsilon_s\}_{s=1,\dots,T}$ is conditionally 1-sub-Gaussian conditioned on $\mathcal{F}_{t-1}$.

Note that the observation $y_t$ is equal to 1 if $x_{t,1}$ is preferred over $x_{t,2}$ and 0 otherwise. Therefore, the noise $\varepsilon_t$ can be expressed as

$$
\varepsilon_t = \begin{cases} 1 - \mu(f(x_{t,1}) - f(x_{t,2})), & \text{w.p. } \mu(f(x_{t,1}) - f(x_{t,2})) \\ -\mu(f(x_{t,1}) - f(x_{t,2})), & \text{w.p. } 1 - \mu(f(x_{t,1}) - f(x_{t,2})), \end{cases}
$$

It can be easily seen that $\varepsilon_s$ is $\mathcal{F}_t$-measurable. Next, if can be easily verified that that conditioned on $\mathcal{F}_{t-1}$ (i.e., given $x_{t,1}$ and $x_{t,2}$), we have that $\mathbb{E}[\varepsilon_t|\mathcal{F}_{t-1}] = 0$. Also note that the absolute value of $\varepsilon_t$ is bounded: $|\varepsilon_t| \leq 1$. Therefore, we can infer that $\varepsilon_t$ is conditionally 1-sub-Gaussian, i.e.,

$$
\mathbb{E}[\exp(\lambda \varepsilon_t)|\mathcal{F}_t] \leq \exp\left( \frac{\lambda^2 \sigma^2}{2} \right), \quad \forall \lambda \in \mathbb{R}.
$$

with $\sigma = 1$.

Next, making use of the 1-sub-sub-Gaussianity of the sequence of noise $\{\varepsilon_s\}$ and Theorem 1 from Abbasi-Yadkori et al. (2011), we can show that with probability of at least $1 - \delta$,

$$
\begin{aligned}
\left\| \sum_{s=1}^{t-1} \varepsilon_s \varphi_s' \frac{1}{\sqrt{m}} \right\|_{V_{t-1}^{-1}} &\leq \sqrt{\log\left( \frac{\det V_{t-1}}{\det V_0} \right) + 2\log(1/\delta)} \\
&\leq \sqrt{\log \det\left( \frac{\kappa_\mu}{\lambda}\mathbf{H}' + \mathbf{I} \right) + 2\log(1/\delta)} \\
&\leq \sqrt{\widetilde{d} + 2\log(1/\delta)},
\end{aligned}
$$

in which we have made use of the definition of the effective dimension $\widetilde{d} = \log \det\left( \frac{\kappa_\mu}{\lambda}\mathbf{H}' + \mathbf{I} \right)$. This completes the proof. □

Finally, we plug Lemma 8 into equation Eq. (20) to complete the proof of Lemma 6:

$$
\sqrt{m} \|\theta_f - \theta_t\|_{V_{t-1}} \leq \beta_T + B\sqrt{\frac{\lambda}{\kappa_\mu}} + 1, \quad \forall t \in [T].
$$

### A.2.2 PROOF OF THEOREM 1

**Theorem 1.** *Let $\delta \in (0,1)$, $\varepsilon'_{m,t} \doteq C_2 m^{-1/6} \sqrt{\log m} L^3 \left(\frac{t}{\lambda}\right)^{4/3}$ for some absolute constant $C_2 > 0$. As long as $m \geq poly(T, L, K, 1/\kappa_\mu, L_\mu, 1/\lambda_0, 1/\lambda, \log(1/\delta))$, then with probability of at least $1 - \delta$,*

$$|\,[f(x) - f(x')] - [h(x;\theta_t) - h(x';\theta_t)]\,| \leq \nu_T \sigma_{t-1}(x, x') + 2\varepsilon'_{m,t},$$

*for all $x, x' \in \mathcal{X}_t, t \in [T]$.*

*Proof.* Denote $\varphi(x) = g(x;\theta_0)$. Recall that Lemma 2 tells us that $f(x) = \langle g(x;\theta_0), \theta_f - \theta_0 \rangle = \langle \varphi(x), \theta_f - \theta_0 \rangle$ for all $x \in \mathcal{X}_t, t \in [T]$. To begin with, for all $x, x' \in \mathcal{X}_t, t \in [T]$ we have that

$$
\begin{aligned}
&|f(x) - f(x') - \langle \varphi(x) - \varphi(x'), \theta_t - \theta_0 \rangle| \\
&= |\langle \varphi(x) - \varphi(x'), \theta_f - \theta_0 \rangle - \langle \varphi(x) - \varphi(x'), \theta_t - \theta_0 \rangle| \\
&= |\langle \varphi(x) - \varphi(x'), \theta_f - \theta_t \rangle| \\
&= |\langle \frac{1}{\sqrt{m}} \varphi(x) - \varphi(x'), \sqrt{m}\,(\theta_f - \theta_t) \rangle| \\
&\leq \left\| \frac{1}{\sqrt{m}} (\varphi(x) - \varphi(x')) \right\|_{V_{t-1}^{-1}} \sqrt{m} \|\theta_f - \theta_t\|_{V_{t-1}} \\
&\leq \left\| \frac{1}{\sqrt{m}} (\varphi(x) - \varphi(x')) \right\|_{V_{t-1}^{-1}} \left( \beta_T + B\sqrt{\frac{\lambda}{\kappa_\mu}} + 1 \right),
\end{aligned}
\tag{23}
$$

in which we have used Lemma 6 in the last inequality. Now making use of the equation above and Lemma 1, we have that

$$
\begin{aligned}
&|f(x) - f(x') - (h(x;\theta_t) - h(x';\theta_t))| \\
&= |f(x) - f(x') - \langle \varphi(x) - \varphi(x'), \theta_t - \theta_0 \rangle \\
&\qquad\qquad + \langle \varphi(x) - \varphi(x'), \theta_t - \theta_0 \rangle - (h(x;\theta_t) - h(x';\theta_t))| \\
&\leq |(f(x) - f(x')) - \langle \varphi(x) - \varphi(x'), \theta_t - \theta_0 \rangle| \\
&\qquad\qquad + |\langle \varphi(x) - \varphi(x'), \theta_t - \theta_0 \rangle - (h(x;\theta_t) - h(x';\theta_t))| \\
&\leq \left\| \frac{1}{\sqrt{m}} (\varphi(x) - \varphi(x')) \right\|_{V_{t-1}^{-1}} \left( \beta_T + B\sqrt{\frac{\lambda}{\kappa_\mu}} + 1 \right) + 2\varepsilon'_{m,t}.
\end{aligned}
$$

This completes the proof of Theorem 1. □

### A.3 REGRET ANALYSIS

Now we can analyze the instantaneous regret. To begin with, we have

$$
\begin{aligned}
2r_t &= f(x_t^*) - f(x_{t,1}) + f(x_t^*) - f(x_{t,2}) \\
&\overset{(a)}{\leq} \langle \varphi(x_t^*) - \varphi(x_{t,1}), \theta_t - \theta_0 \rangle + \left\| \frac{1}{\sqrt{m}} (\varphi(x_t^*) - \varphi(x_{t,1})) \right\|_{V_{t-1}^{-1}} \left( \beta_T + B\sqrt{\lambda/\kappa_\mu} + 1 \right) \\
&\quad + \langle \varphi(x_t^*) - \varphi(x_{t,2}), \theta_t - \theta_0 \rangle + \left\| \frac{1}{\sqrt{m}} (\varphi(x_t^*) - \varphi(x_{t,2})) \right\|_{V_{t-1}^{-1}} \left( \beta_T + B\sqrt{\lambda/\kappa_\mu} + 1 \right) \\
&= \langle \varphi(x_t^*) - \varphi(x_{t,1}), \theta_t - \theta_0 \rangle + \left\| \frac{1}{\sqrt{m}} (\varphi(x_t^*) - \varphi(x_{t,1})) \right\|_{V_{t-1}^{-1}} \left( \beta_T + B\sqrt{\lambda/\kappa_\mu} + 1 \right) \\
&\quad + \langle \varphi(x_t^*) - \varphi(x_{t,1}), \theta_t - \theta_0 \rangle + \langle \varphi(x_{t,1}) - \varphi(x_{t,2}), \theta_t - \theta_0 \rangle \\
&\quad + \left\| \frac{1}{\sqrt{m}} (\varphi(x_t^*) - \varphi(x_{t,1}) + \varphi(x_{t,1}) - \varphi(x_{t,2})) \right\|_{V_{t-1}^{-1}} \left( \beta_T + B\sqrt{\lambda/\kappa_\mu} + 1 \right)
\end{aligned}
$$

$$\leq 2\langle \varphi(x_t^*) - \varphi(x_{t,1}), \theta_t - \theta_0 \rangle + 2\left\| \frac{1}{\sqrt{m}} \left( \varphi(x_t^*) - \varphi(x_{t,1}) \right) \right\|_{V_{t-1}^{-1}} \left( \beta_T + B\sqrt{\lambda/\kappa_\mu} + 1 \right)$$

$$+ \langle \varphi(x_{t,1}) - \varphi(x_{t,2}), \theta_t - \theta_0 \rangle + \left\| \frac{1}{\sqrt{m}} \left( \varphi(x_{t,1}) - \varphi(x_{t,2}) \right) \right\|_{V_{t-1}^{-1}} \left( \beta_T + B\sqrt{\lambda/\kappa_\mu} + 1 \right)$$

$$\overset{(b)}{\leq} 2h(x_t^*; \theta_t) - 2h(x_{t,1}; \theta_t) + 4\varepsilon'_{m,t} + 2\left\| \frac{1}{\sqrt{m}} \left( \varphi(x_t^*) - \varphi(x_{t,1}) \right) \right\|_{V_{t-1}^{-1}} \left( \beta_T + B\sqrt{\lambda/\kappa_\mu} + 1 \right)$$

$$+ h(x_{t,1}; \theta_t) - h(x_{t,2}; \theta_t) + 2\varepsilon'_{m,t} + \left\| \frac{1}{\sqrt{m}} \left( \varphi(x_{t,1}) - \varphi(x_{t,2}) \right) \right\|_{V_{t-1}^{-1}} \left( \beta_T + B\sqrt{\lambda/\kappa_\mu} + 1 \right)$$

$$\overset{(c)}{\leq} 2h(x_{t,2}; \theta_t) - 2h(x_{t,1}; \theta_t) + 2\left\| \frac{1}{\sqrt{m}} \left( \varphi(x_{t,2}) - \varphi(x_{t,1}) \right) \right\|_{V_{t-1}^{-1}} \left( \beta_T + B\sqrt{\lambda/\kappa_\mu} + 1 \right)$$

$$+ h(x_{t,1}; \theta_t) - h(x_{t,2}; \theta_t) + 6\varepsilon'_{m,t} + \left\| \frac{1}{\sqrt{m}} \left( \varphi(x_{t,1}) - \varphi(x_{t,2}) \right) \right\|_{V_{t-1}^{-1}} \left( \beta_T + B\sqrt{\lambda/\kappa_\mu} + 1 \right)$$

$$= h(x_{t,2}; \theta_t) - h(x_{t,1}; \theta_t) + 3\left\| \frac{1}{\sqrt{m}} \left( \varphi(x_{t,1}) - \varphi(x_{t,2}) \right) \right\|_{V_{t-1}^{-1}} \left( \beta_T + B\sqrt{\lambda/\kappa_\mu} + 1 \right) + 6\varepsilon'_{m,t}$$

$$\overset{(d)}{\leq} 3\left( \beta_T + B\sqrt{\lambda/\kappa_\mu} + 1 \right) \left\| \frac{1}{\sqrt{m}} \left( \varphi(x_{t,1}) - \varphi(x_{t,2}) \right) \right\|_{V_{t-1}^{-1}} + 6\varepsilon'_{m,t}.$$

Step $(a)$ follows from Eq. (23), step $(b)$ results from Lemma 1, step $(c)$ follows from the way in which $x_{t,2}$ is selected: $x_{t,2} = \arg\max_{x \in \mathcal{X}_t} \left( h(x; \theta_t) + \left\| \frac{1}{\sqrt{m}} \left( \varphi(x) - \varphi(x_{t,1}) \right) \right\|_{V_{t-1}^{-1}} \left( \beta_T + B\sqrt{\frac{\lambda}{\kappa_\mu}} + 1 \right) \right)$, and step $(d)$ follows from the way in which $x_{t,1}$ is selected: $x_{t,1} = \arg\max_{x \in \mathcal{X}_t} h(x; \theta_t)$.

Now denote $\sigma_{t-1}^2(x_{t,1}, x_{t,2}) \doteq \frac{\lambda}{\kappa_\mu} \left\| \frac{1}{\sqrt{m}}(\varphi(x_{t,1}) - \varphi(x_{t,2})) \right\|_{V_{t-1}^{-1}}^2$. Of note, $\sigma_{t-1}^2(x_{t,1}, x_{t,2})$ can be interpreted as the Gaussian process posterior variance with the kernel defined as $k\big((x_1, x_2), (x_1', x_2')\big) = \langle \frac{1}{\sqrt{m}} \left( \varphi(x_1) - \varphi(x_2) \right), \left( \frac{1}{\sqrt{m}}(\varphi(x_1') - \varphi(x_2')) \right) \rangle$, and with a noise variance of $\frac{\lambda}{\kappa_\mu}$. It is easy to see that the kernel is positive semi-definite and is hence a valid kernel. Following the derivations of the Gaussian process posterior variance, it is easy to verify that

$$\sigma_{t-1}^2(x_{t,1}, x_{t,2}) \leq (\varphi(x_{t,1}) - \varphi(x_{t,2}))^\top (\varphi(x_{t,1}) - \varphi(x_{t,2})) \frac{1}{m}$$

$$= \left\| (\varphi(x_{t,1}) - \varphi(x_{t,2})) \frac{1}{\sqrt{m}} \right\|_2^2$$

$$= \frac{1}{m} \|\varphi(x_{t,1}) - \varphi(x_{t,2})\|_2^2 \leq c_0,$$

in which we have denoted $c_0 > 0$ as an absolute constant such that $\frac{1}{m} \|\varphi(x) - \varphi(x')\|_2^2 \leq c_0, \forall x, x' \in \mathcal{X}_t, t \in [T]$. Note that this is similar to the standard assumption in the literature that the value of the NTK is upper-bounded by a constant (Kassraie & Krause, 2022). Therefore, this implies that $\sigma_{t-1}^2(x_{t,1}, x_{t,2})/c_0 \leq 1$ for some constant $c_0 \geq 1$. Recall that we choose $\lambda$ such that $\lambda/\kappa_\mu \geq 1$. Note that for any $\alpha \in [0, 1]$, we have that $\alpha/2 \leq \log(1 + \alpha)$. With these, we have that

$$\frac{1}{2} \left( \frac{\lambda}{\kappa_\mu} \right)^{-1} \frac{\sigma_{t-1}^2(x_{t,1}, x_{t,2})}{c_0} \leq \log \left( 1 + \left( \frac{\lambda}{\kappa_\mu} \right)^{-1} \frac{\sigma_{t-1}^2(x_{t,1}, x_{t,2})}{c_0} \right)$$

$$\leq \log \left( 1 + \left( \frac{\lambda}{\kappa_\mu} \right)^{-1} \sigma_{t-1}^2(x_{t,1}, x_{t,2}) \right),$$

which leads to

$$\sigma_{t-1}^2(x_{t,1}, x_{t,2}) \leq 2c_0 \frac{\lambda}{\kappa_\mu} \log \left( 1 + \frac{\kappa_\mu}{\lambda} \sigma_{t-1}^2(x_{t,1}, x_{t,2}) \right). \tag{24}$$

Following the analysis of Chowdhury & Gopalan (2017) and using the chain rule of conditional information gain, we can show that

$$\sum_{s=1}^{t} \log\left(1 + \frac{\kappa_\mu}{\lambda}\sigma_{s-1}^2(x_{s,1}, x_{s,2})\right) = \log\det\left(\mathbf{I} + \frac{\kappa_\mu}{\lambda}\mathbf{K}_t\right),$$

in which $\mathbf{K}_t$ is a $t \times t$ matrix in which every element is $\mathbf{K}_t[i,j] = \frac{1}{m}(\varphi(x_{i,1}) - \varphi(x_{i,2}))^\top(\varphi(x_{j,1}) - \varphi(x_{j,2}))$. Define the $p \times t$ matrix $\mathbf{J}_t = [\frac{1}{\sqrt{m}}(\varphi(x_{i,1}) - \varphi(x_{i,2}))]_{i=1,\dots,t}$. Then we have that $\mathbf{K}_t = \mathbf{J}_t^\top\mathbf{J}_t$. This allows us to show that

$$
\begin{aligned}
\sum_{s=1}^{t} \log\left(1 + \frac{\kappa_\mu}{\lambda}\sigma_{s-1}^2(x_{s,1}, x_{s,2})\right) &= \log\det\left(\mathbf{I} + \frac{\kappa_\mu}{\lambda}\mathbf{K}_t\right) \\
&= \log\det\left(\mathbf{I} + \frac{\kappa_\mu}{\lambda}\mathbf{J}_t^\top\mathbf{J}_t\right) \\
&= \log\det\left(\mathbf{I} + \frac{\kappa_\mu}{\lambda}\mathbf{J}_t\mathbf{J}_t^\top\right) \\
&= \log\det\left(\mathbf{I} + \frac{\kappa_\mu}{\lambda}\sum_{s=1}^{t}(\varphi(x_{s,1}) - \varphi(x_{s,2}))(\varphi(x_{s,1}) - \varphi(x_{s,2}))^\top\frac{1}{m}\right) \\
&\leq \log\det\left(\frac{\kappa_\mu}{\lambda}\mathbf{H}' + \mathbf{I}\right) \\
&= \widetilde{d}
\end{aligned}
\tag{25}
$$

in which we have followed the same line of analysis as Eq. (21) and Eq. (22) in the second last inequality.

Combining the results from Eq. (24) and Eq. (25), we have that

$$\sum_{t=1}^{T}\sigma_{t-1}^2(x_{t,1}, x_{t,2}) \leq 2c_0\frac{\lambda}{\kappa_\mu}\sum_{t=1}^{T}\log\left(1 + \frac{\kappa_\mu}{\lambda}\sigma_{t-1}^2(x_{t,1}, x_{t,2})\right)$$

$$\leq 2c_0\frac{\lambda}{\kappa_\mu}\widetilde{d}.$$

Finally, we can derive an upper bound on the cumulative regret:

$$
\begin{aligned}
\mathfrak{R}_T = \sum_{t=1}^{T}r_t &\leq \sum_{t=1}^{T}\frac{1}{2}\left(3\left(\beta_T + B\sqrt{\frac{\lambda}{\kappa_\mu}} + 1\right)\sqrt{\frac{\kappa_\mu}{\lambda}}\sigma_{t-1}(x_{t,1}, x_{t,2}) + 6\varepsilon'_{m,t}\right) \\
&\leq \frac{3}{2}\left(\beta_T + B\sqrt{\frac{\lambda}{\kappa_\mu}} + 1\right)\sqrt{\frac{\kappa_\mu}{\lambda}}\sum_{t=1}^{T}\sigma_{t-1}(x_{t,1}, x_{t,2}) + 6T\varepsilon'_{m,T} \\
&\leq \frac{3}{2}\left(\beta_T + B\sqrt{\frac{\lambda}{\kappa_\mu}} + 1\right)\sqrt{\frac{\kappa_\mu}{\lambda}}\sqrt{T\sum_{t=1}^{T}\sigma_{t-1}^2(x_{t,1}, x_{t,2})} + 6T\varepsilon'_{m,T} \\
&\leq \frac{3}{2}\left(\beta_T + B\sqrt{\frac{\lambda}{\kappa_\mu}} + 1\right)\sqrt{\frac{\kappa_\mu}{\lambda}}\sqrt{T2c_0\frac{\lambda}{\kappa_\mu}\widetilde{d}} + 6T\varepsilon'_{m,T}. \\
&\leq \frac{3}{2}\left(\beta_T + B\sqrt{\frac{\lambda}{\kappa_\mu}} + 1\right)\sqrt{T2c_0\widetilde{d}} + 6T\varepsilon'_{m,T}.
\end{aligned}
$$

Recall that $\varepsilon'_{m,t} = C_2 m^{-1/6}\sqrt{\log m}L^3\left(\frac{t}{\lambda}\right)^{4/3}$. It can be easily verified that as long as the conditions on $m$ specified in Eq. (8) are satisfied (i.e., as long as the NN is wide enough), we have that $6T\varepsilon'_{m,T} \leq 1$. Recall that $\beta_T = \widetilde{\mathcal{O}}(\frac{1}{\kappa_\mu}\sqrt{\widetilde{d}})$. This allows us to simplify the regret upper bound to be

$$\mathfrak{R}_T \leq \frac{3}{2}\left(\beta_T + B\sqrt{\frac{\lambda}{\kappa_\mu}} + 1\right)\sqrt{T2c_0\widetilde{d}} + 1 = \widetilde{\mathcal{O}}\left(\left(\frac{\sqrt{\widetilde{d}}}{\kappa_\mu} + B\sqrt{\frac{\lambda}{\kappa_\mu}}\right)\sqrt{\widetilde{d}T}\right).$$

## B    THEORETICAL ANALYSIS FOR **NDB-TS**

Denote $\nu_T \triangleq \left(\beta_T + B\sqrt{\lambda/\kappa_\mu} + 1\right)\sqrt{\kappa_\mu/\lambda}$, $c_t \triangleq \nu_T(1 + \sqrt{2\log(Kt^2)})$, and $\sigma_{t-1}^2(x_1, x_2) \triangleq \frac{\lambda}{\kappa_\mu}\left\|\frac{1}{\sqrt{m}}(\varphi(x_1) - \varphi(x_2))\right\|_{V_{t-1}^{-1}}^2$. Here we use $\mathcal{F}_{t-1}$ to denote the filtration containing the history of selected inputs and observations up to iteration $t - 1$. To use Thompson sampling (TS) to select the second arm $x_{t,2}$, firstly, for each arm $x \in \mathcal{X}_t$, we sample a reward value $\widetilde{r}_t(x)$ from the normal distribution $\mathcal{N}\left(h(x;\theta_t) - h(x_{t,1};\theta_t), \nu_T^2\sigma_{t-1}^2(x, x_{t,1})\right)$. Then, we choose the second arm as $x_{t,2} = \arg\max_{x \in \mathcal{X}_t}\widetilde{r}_t(x)$.

**Lemma 9.** *Let $\delta \in (0, 1)$. Define $E^f(t)$ as the following event:*

$$|\left[f(x) - f(x_{t,1})\right] - \left[h(x;\theta_t) - h(x_{t,1};\theta_t)\right]| \leq \nu_T\sigma_{t-1}(x, x_{t,1}) + 2\varepsilon'_{m,t}.$$

*According to Theorem 1, we have that the event $E^f(t)$ holds with probability of at least $1 - \delta$.*

**Lemma 10.** *Define $E^{f_t}(t)$ as the following event*

$$|\widetilde{r}_t(x) - \left[h(x;\theta_t) - h(x_{t,1};\theta_t)\right]| \leq \nu_T\sqrt{2\log(Kt^2)}\sigma_{t-1}(x, x_{t,1}).$$

*We have that $\mathbb{P}\left[E^{f_t}(t)|\mathcal{F}_{t-1}\right] \geq 1 - 1/t^2$ for any possible filtration $\mathcal{F}_{t-1}$.*

**Definition 1.** *In iteration $t$, define the set of saturated points as*

$$S_t = \{x \in \mathcal{X}_t : \Delta(x) > c_t\sigma_{t-1}(x, x_{t,1}) + 4\varepsilon'_{m,t}\},$$

*where $\Delta(x) = f(x_t^*) - f(x)$ and $x_t^* \in \arg\max_{x \in \mathcal{X}_t}f(x)$.*

Note that according to this definition, $x_t^*$ is always unsaturated.

**Lemma 11.** *For any filtration $\mathcal{F}_{t-1}$, conditioned on the event $E^f(t)$, we have that $\forall x \in \mathcal{Q}$,*

$$\mathbb{P}\left(\widetilde{r}_t(x) + 2\varepsilon'_{m,t} > f(x) - f(x_{t,1})|\mathcal{F}_{t-1}\right) \geq p,$$

*where $p = \frac{1}{4e\sqrt{\pi}}$.*

*Proof.* Adding and subtracting $\frac{\mu_{t-1}(x)}{\nu_t\sigma_{t-1}(x)}$ both sides of $\mathbb{P}\left(f_t(x) > \rho_m f(x)|\mathcal{F}_{t-1}\right)$, we get

$$\mathbb{P}\left\{\widetilde{r}_t(x) + 2\varepsilon'_{m,t} > f(x) - f(x_{t,1})|\mathcal{F}_{t-1}\right\}$$

$$= \mathbb{P}\left\{\frac{\widetilde{r}_t(x) + 2\varepsilon'_{m,t} - \left[h(x;\theta_t) - h(x_{t,1};\theta_t)\right]}{\nu_T\sigma_{t-1}(x, x_{t,1})} > \frac{f(x) - f(x_{t,1}) - \left[h(x;\theta_t) - h(x_{t,1};\theta_t)\right]}{\nu_T\sigma_{t-1}(x, x_{t,1})}\Big|\mathcal{F}_{t-1}\right\}$$

$$\geq \mathbb{P}\left\{\frac{\widetilde{r}_t(x) + 2\varepsilon'_{m,t} - \left[h(x;\theta_t) - h(x_{t,1};\theta_t)\right]}{\nu_T\sigma_{t-1}(x, x_{t,1})} > \frac{|f(x) - f(x_{t,1}) - \left[h(x;\theta_t) - h(x_{t,1};\theta_t)\right]|}{\nu_T\sigma_{t-1}(x, x_{t,1})}\Big|\mathcal{F}_{t-1}\right\}$$

$$\geq \mathbb{P}\left\{\frac{\widetilde{r}_t(x) - \left[h(x;\theta_t) - h(x_{t,1};\theta_t)\right]}{\nu_T\sigma_{t-1}(x, x_{t,1})} > \frac{|f(x) - f(x_{t,1}) - \left[h(x;\theta_t) - h(x_{t,1};\theta_t)\right]| - 2\varepsilon'_{m,t}}{\nu_T\sigma_{t-1}(x, x_{t,1})}\Big|\mathcal{F}_{t-1}\right\}$$

$$\geq \mathbb{P}\left\{\frac{\widetilde{r}_t(x) - \left[h(x;\theta_t) - h(x_{t,1};\theta_t)\right]}{\nu_T\sigma_{t-1}(x, x_{t,1})} > 1|\mathcal{F}_{t-1}\right\}$$

$$\geq \frac{1}{4e\sqrt{\pi}},$$

in which the third inequality makes use of Lemma 9 (note that we have conditioned on the event $E^f(t)$ here), and the last inequality follows from the Gaussian anti-concentration inequality: $\mathbb{P}(z > a) \geq \exp(-a^2)/(4\sqrt{\pi}a)$ where $z \sim \mathcal{N}(0, 1)$. $\qquad\square$

The next lemma proves a lower bound on the probability that the selected input $x_{t,2}$ is unsaturated.

**Lemma 12.** *For any filtration $\mathcal{F}_{t-1}$, conditioned on the event $E^f(t)$, we have that,*

$$\mathbb{P}\left(x_{t,2} \in \mathcal{X}_t \setminus S_t|\mathcal{F}_{t-1}\right) \geq p - 1/t^2.$$

*Proof.* To begin with, we have that

$$\mathbb{P}\left(x_{t,2} \in \mathcal{X}_t \setminus S_t | \mathcal{F}_{t-1}\right) \geq \mathbb{P}\left(\widetilde{r}_t(x_t^*) > \widetilde{r}_t(x), \forall x \in S_t | \mathcal{F}_{t-1}\right). \tag{26}$$

This inequality can be justified because the event on the right hand side implies the event on the left hand side. Specifically, according to Definition 1, $x_t^*$ is always unsaturated. Therefore, because $x_{t,2}$ is selected by $x_{t,2} = \arg\max_{x \in \mathcal{X}_t} \widetilde{r}_t(x)$, we have that if $\widetilde{r}_t(x_t^*) > \widetilde{r}_t(x), \forall x \in S_t$, then the selected $x_{t,2}$ is guaranteed to be unsaturated. Now conditioning on both events $E^f(t)$ and $E^{f_t}(t)$, for all $x \in S_t$, we have that

$$
\begin{aligned}
\widetilde{r}_t(x) &\leq f(x) - f(x_{t,1}) + c_t \sigma_{t-1}(x, x_{t,1}) + 2\varepsilon'_{m,t} \\
&= f(x) - f(x_{t,1}) + c_t \sigma_{t-1}(x, x_{t,1}) + 4\varepsilon'_{m,t} - 2\varepsilon'_{m,t} \\
&\leq f(x) - f(x_{t,1}) + \Delta(x) - 2\varepsilon'_{m,t} \\
&= f(x) - f(x_{t,1}) + f(x_t^*) - f(x) - 2\varepsilon'_{m,t} \\
&= f(x_t^*) - f(x_{t,1}) - 2\varepsilon'_{m,t}
\end{aligned}
\tag{27}
$$

in which the first inequality follows from Lemma 9 and Lemma 10 and the second inequality makes use of Definition 1. Next, separately considering the cases where the event $E^{f_t}(t)$ holds or not and making use of Eq. (26) and Eq. (27), we have that

$$
\begin{aligned}
\mathbb{P}\left(x_{t,2} \in \mathcal{X}_t \setminus S_t | \mathcal{F}_{t-1}\right) &\geq \mathbb{P}\left(\widetilde{r}_t(x_t^*) > \widetilde{r}_t(x), \forall x \in S_t | \mathcal{F}_{t-1}\right) \\
&\geq \mathbb{P}\left(\widetilde{r}_t(x_t^*) > f(x_t^*) - f(x_{t,1}) - 2\varepsilon'_{m,t} | \mathcal{F}_{t-1}\right) - \mathbb{P}\left(\overline{E^{f_t}(t)} | \mathcal{F}_{t-1}\right) \\
&\geq p - 1/t^2,
\end{aligned}
$$

in which the last inequality has made use of Lemma 10 and Lemma 11. $\qquad\square$

Next, we use the following lemma to derive an upper bound on the expected instantaneous regret.

**Lemma 13.** *For any filtration $\mathcal{F}_{t-1}$, conditioned on the event $E^f(t)$, we have that,*

$$\mathbb{E}[2r_t | \mathcal{F}_{t-1}] \leq \frac{23c_t}{p} \mathbb{E}\left[\sigma_{t-1}(x_{t,2}, x_{t,1}) | \mathcal{F}_{t-1}\right] + 18\varepsilon'_{m,t} + \frac{4}{t^2}.$$

*Proof.* To begin with, define $\overline{x}_t$ as the unsaturated input with the smallest $\sigma_{t-1}(x, x_{t,1})$:

$$\overline{x}_t = \arg\min_{x \in \mathcal{X}_t \setminus S_t} \sigma_{t-1}(x, x_{t,1}).$$

This definition gives us:

$$
\begin{aligned}
\mathbb{E}\left[\sigma_{t-1}(x_{t,2}, x_{t,1}) | \mathcal{F}_{t-1}\right] &\geq \mathbb{E}\left[\sigma_{t-1}(x_{t,2}, x_{t,1}) | \mathcal{F}_{t-1}, x_t \in \mathcal{X}_t \setminus S_t\right] \mathbb{P}\left(x_{t,2} \in \mathcal{X}_t \setminus S_t | \mathcal{F}_{t-1}\right) \\
&\geq \sigma_{t-1}(\overline{x}_t, x_{t,1})(p - 1/t^2),
\end{aligned}
\tag{28}
$$

in which the second inequality makes use of Lemma 12, as well as the definition of $\overline{x}_t$.

Next, conditioned on both events $E^f(t)$ and $E^{f_t}(t)$, we can decompose the instantaneous regret as

$$
\begin{aligned}
2r_t &= f(x_t^*) - f(x_{t,1}) + f(x_t^*) - f(x_{t,2}) \\
&= f(x_t^*) - f(x_{t,2}) + f(x_{t,2}) - f(x_{t,1}) + f(x_t^*) - f(x_{t,2}) \\
&= 2\left[f(x_t^*) - f(x_{t,2})\right] + f(x_{t,2}) - f(x_{t,1}).
\end{aligned}
\tag{29}
$$

Next, we separately analyze the two terms above. Firstly, we have that

$$
\begin{aligned}
f(x_t^*) - f(x_{t,2}) &= f(x_t^*) - f(\overline{x}_t) + f(\overline{x}_t) - f(x_{t,2}) \\
&= \Delta(\overline{x}_t) + \left[f(\overline{x}_t) - f(x_{t,1})\right] - \left[f(x_{t,2}) - f(x_{t,1})\right] \\
&\leq \Delta(\overline{x}_t) + \widetilde{r}_t(\overline{x}_t) + c_t \sigma_{t-1}(\overline{x}_t, x_{t,1}) + 2\varepsilon'_{m,t} - \widetilde{r}_t(x_{t,2}) + c_t \sigma_{t-1}(x_{t,2}, x_{t,1}) + 2\varepsilon'_{m,t} \\
&\leq \Delta(\overline{x}_t) + c_t \sigma_{t-1}(\overline{x}_t, x_{t,1}) + c_t \sigma_{t-1}(x_{t,2}, x_{t,1}) + 4\varepsilon'_{m,t} \\
&\leq c_t \sigma_{t-1}(\overline{x}_t, x_{t,1}) + 4\varepsilon'_{m,t} + c_t \sigma_{t-1}(\overline{x}_t, x_{t,1}) + c_t \sigma_{t-1}(x_{t,2}, x_{t,1}) + 4\varepsilon'_{m,t} \\
&\leq 2c_t \sigma_{t-1}(\overline{x}_t, x_{t,1}) + c_t \sigma_{t-1}(x_{t,2}, x_{t,1}) + 8\varepsilon'_{m,t},
\end{aligned}
$$

$$\tag{30}$$

in which the first inequality follows from Lemma 9 and Lemma 10, the second inequality follows from the way in which $x_{t,2}$ is selected: $x_{t,2} = \arg\max_{x \in \mathcal{X}_t} \widetilde{r}_t(x)$ which guarantees that $\widetilde{r}_t(\overline{x}_t) \leq \widetilde{r}_t(x_{t,2})$. The third inequality follows because $\overline{x}_t$ is unsaturated. Next, we analyze the second term from Eq. (29).

$$
\begin{aligned}
f(x_{t,2}) - f(x_{t,1}) &\leq h(x_{t,2}; \theta_t) - h(x_{t,1}; \theta_t) + \nu_T \sigma_{t-1}(x_{t,2}, x_{t,1}) + 2\varepsilon'_{m,t} \\
&\leq \nu_T \sigma_{t-1}(x_{t,2}, x_{t,1}) + 2\varepsilon'_{m,t} \\
&\leq c_t \sigma_{t-1}(x_{t,2}, x_{t,1}) + 2\varepsilon'_{m,t},
\end{aligned}
\tag{31}
$$

in which the first inequality follows from Lemma 9, the second inequality results from the way in which $x_{t,1}$ is selected: $x_{t,1} = \arg\max_{x \in \mathcal{X}_t} h(x; \theta_t)$, and the third inequality follows because $\nu_T \leq c_t$ by definition. Now we can plug Eq. (30) and Eq. (31) into Eq. (29):

$$
\begin{aligned}
2r_t &\leq 2\left(2c_t \sigma_{t-1}(\overline{x}_t, x_{t,1}) + c_t \sigma_{t-1}(x_{t,2}, x_{t,1}) + 8\varepsilon'_{m,t}\right) + c_t \sigma_{t-1}(x_{t,2}, x_{t,1}) + 2\varepsilon'_{m,t} \\
&\leq 4c_t \sigma_{t-1}(\overline{x}_t, x_{t,1}) + 3c_t \sigma_{t-1}(x_{t,2}, x_{t,1}) + 18\varepsilon'_{m,t}.
\end{aligned}
\tag{32}
$$

Next, by separately considering the cases where the event $E^{f_t}(t)$ holds and otherwise, we are ready to upper-bound the expected instantaneous regret:

$$
\begin{aligned}
\mathbb{E}[2r_t | \mathcal{F}_{t-1}] &\leq \mathbb{E}[4c_t \sigma_{t-1}(\overline{x}_t, x_{t,1}) + 3c_t \sigma_{t-1}(x_{t,2}, x_{t,1}) + 18\varepsilon'_{m,t} | \mathcal{F}_{t-1}] + \frac{4}{t^2} \\
&\leq \mathbb{E}\left[4c_t \sigma_{t-1}(x_{t,2}, x_{t,1})\frac{1}{p - 1/t^2} + 3c_t \sigma_{t-1}(x_{t,2}, x_{t,1}) + 18\varepsilon'_{m,t} | \mathcal{F}_{t-1}\right] + \frac{4}{t^2} \\
&= c_t\left(\frac{4}{p - 1/t^2} + 3\right)\mathbb{E}\left[\sigma_{t-1}(x_{t,2}, x_{t,1}) | \mathcal{F}_{t-1}\right] + 18\varepsilon'_{m,t} + \frac{4}{t^2} \\
&\leq c_t\frac{23}{p}\mathbb{E}\left[\sigma_{t-1}(x_{t,2}, x_{t,1}) | \mathcal{F}_{t-1}\right] + 18\varepsilon'_{m,t} + \frac{4}{t^2}
\end{aligned}
$$

in which the first inequality have made use of Eq. (32), the second inequality results from Eq. (28), and the last inequality follows because $\frac{1}{p-1/t^2} \leq 5/p$ and $1 \leq 1/p$. □

Next, we define the following stochastic process $(Y_t : t = 0, \ldots, T)$, which we prove is a super-martingale in the subsequent lemma by making use of Lemma 13.

**Definition 2.** *Define $Y_0 = 0$, and for all $t = 1, \ldots, T$,*

$$
\overline{r}_t = r_t \mathbb{I}\{E^f(t)\}, \quad X_t = \overline{r}_t - \frac{23c_t}{2p}\sigma_{t-1}(x_{t,2}, x_{t,1}) - 9\varepsilon'_{m,t} - \frac{2}{t^2}, \text{ and } \quad Y_t = \sum_{s=1}^{t} X_s.
$$

**Lemma 14.** $(Y_t : t = 0, \ldots, T)$ *is a super-martingale with respect to the filtration $\mathcal{F}_t$.*

*Proof.* As $X_t = Y_t - Y_{t-1}$, we have

$$
\begin{aligned}
\mathbb{E}[Y_t - Y_{t-1} | \mathcal{F}_{t-1}] &= \mathbb{E}[X_t | \mathcal{F}_{t-1}] \\
&= \mathbb{E}\left[\overline{r}_t - \frac{23c_t}{2p}\sigma_{t-1}(x_{t,2}, x_{t,1}) - 9\varepsilon'_{m,t} - \frac{2}{t^2} | \mathcal{F}_{t-1}\right] \\
&= \mathbb{E}[\overline{r}_t | \mathcal{F}_{t-1}] - \left[\frac{23c_t}{2p}\mathbb{E}[\sigma_{t-1}(x_{t,2}, x_{t,1}) | \mathcal{F}_{t-1}] + 9\varepsilon'_{m,t} + \frac{2}{t^2}\right] \\
&\leq 0.
\end{aligned}
$$

When the event $E^f(t)$ holds, the last inequality follows from Lemma 13; when $E^f(t)$ is false, $\overline{r}_t = 0$ and hence the inequality trivially holds. □

Lastly, we are ready to prove the upper bound on the cumulative regret of ou algorithm by applying the Azuma-Hoeffding Inequality to the stochastic process defined above.

*Proof.* To begin with, we derive an upper bound on $|Y_t - Y_{t-1}|$:

$$|Y_t - Y_{t-1}| = |X_t| \le |\overline{r}_t| + \frac{23c_t}{2p}\sigma_{t-1}(x_{t,2}, x_{t,1}) + 9\varepsilon'_{m,t} + \frac{2}{t^2}$$

$$\le 2 + \frac{23c_t}{2p}c_0 + 9\varepsilon'_{m,t} + 2$$

$$\le \frac{1}{p}\left(4 + 12c_t c_0 + 9\varepsilon'_{m,t}\right),$$

where the second inequality follows because $\sigma_{t-1}(x_{t,2}, x_{t,1}) \le c_0$, and the last inequality follows since $\frac{1}{p} \ge 1$. Now we are ready to apply the Azuma-Hoeffding Inequality to $(Y_t : t = 0, \dots, T)$ with an error probability of $\delta$:

$$\sum_{t=1}^{T}\overline{r}_t \le \sum_{t=1}^{T}\frac{23c_t}{2p}\sigma_{t-1}(x_{t,2}, x_{t,1}) + \sum_{t=1}^{T}9\varepsilon'_{m,t} + \sum_{t=1}^{T}\frac{2}{t^2}$$

$$+ \sqrt{2\log(1/\delta)\sum_{t=1}^{T}\left(\frac{1}{p}\left(4 + 12c_t c_0 + 9\varepsilon'_{m,t}\right)\right)^2}$$

$$\le 12c_T\sum_{t=1}^{T}\sigma_{t-1}(x_{t,2}, x_{t,1}) + 9T\varepsilon'_{m,T} + 2\sum_{t=1}^{T}1/t^2$$

$$+ \left(\frac{1}{p}\left(4 + 12c_T c_0 + 9\varepsilon'_{m,T}\right)\right)\sqrt{2T\log(1/\delta)}$$

$$\le 12c_T\sqrt{T2c_0\frac{\lambda}{\kappa_\mu}\widetilde{d}} + 9T\varepsilon'_{m,T} + \frac{\pi^2}{3} + \frac{4 + 12c_T c_0 + 9\varepsilon'_{m,T}}{p}\sqrt{2T\log(1/\delta)}.$$

The second inequality makes use of the fact that $c_t$ and $\varepsilon'_{m,t}$ are both monotonically increasing in $t$. The last inequality follows because $\sum_{t=1}^{T}\sigma_{t-1}(x_{t,1}, x_{t,2}) \le \sqrt{T2c_0\frac{\lambda}{\kappa_\mu}\widetilde{d}}$ which we have shown in the proof of the UCB algorithm, and $\sum_{t=1}^{T}1/t^2 \le \pi^2/6$. Note that Appendix B holds with probability $\ge 1 - \delta$. Also note that $\overline{r}_t = r_t$ with probability of $\ge 1 - \delta$ because the event $E^f(t)$ holds with probability of $\ge 1 - \delta$ (Lemma 9). Therefore, replacing $\delta$ by $\delta/2$, the upper bound from Appendix B is an upper bound on $\mathfrak{R}_T = \sum_{t=1}^{T}r_t$ with probability of $1 - \delta$.

Lastly, recall we have defined that $\nu_T \doteq \left(\beta_T + B\sqrt{\lambda/\kappa_\mu} + 1\right)\sqrt{\kappa_\mu/\lambda}$, $c_t \triangleq \nu_T(1 + \sqrt{2\log(Kt^2)})$, and $\beta_T = \widetilde{\mathcal{O}}(\frac{1}{\kappa_\mu}\sqrt{\widetilde{d}})$. This implies that $c_T = \widetilde{O}\left(\left(\frac{1}{\kappa_\mu}\sqrt{\widetilde{d}} + B\sqrt{\lambda/\kappa_\mu}\right)\sqrt{\kappa_\mu/\lambda}\right) = \widetilde{O}\left(\sqrt{\frac{\widetilde{d}}{\kappa_\mu\lambda}} + B\right)$. Also recall that as long as the conditions on $m$ specified in Eq. (8) are satisfied (i.e., as long as the NN is wide enough), we can ensure that $9T\varepsilon'_{m,T} \le 1$. Therefore, the final regret upper bound can be expressed as:

$$\mathfrak{R}_T = \widetilde{O}\left(\left(\sqrt{\frac{\widetilde{d}}{\kappa_\mu\lambda}} + B\right)\sqrt{T\frac{\lambda}{\kappa_\mu}\widetilde{d}} + \left(\sqrt{\frac{\widetilde{d}}{\kappa_\mu\lambda}} + B\right)\sqrt{T}\right)$$

$$= \widetilde{O}\left(\left(\sqrt{\frac{\widetilde{d}}{\kappa_\mu\lambda}} + B\right)\sqrt{T\frac{\lambda}{\kappa_\mu}\widetilde{d}}\right) = \widetilde{O}\left(\left(\frac{\sqrt{\widetilde{d}}}{\kappa_\mu} + B\sqrt{\frac{\lambda}{\kappa_\mu}}\right)\sqrt{T\widetilde{d}}\right).$$

This completes the proof. As we can see, our TS algorithm enjoys the same asymptotic regret upper bound as our UCB algorithm, ignoring the log factors. $\qquad\square$

# C    ADDITIONAL EXPERIMENTS FOR SECTION 5

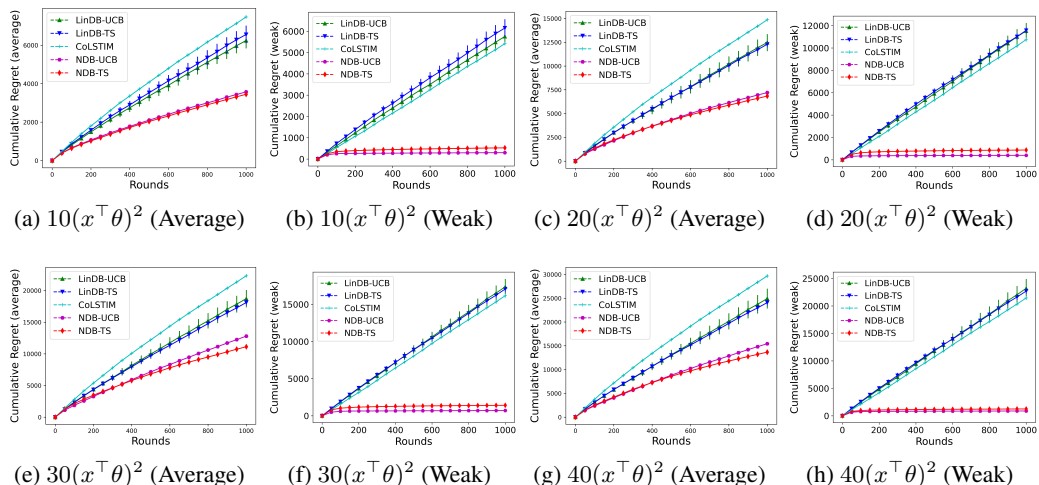

(a) $10(x^\top\theta)^2$ (Average) (b) $10(x^\top\theta)^2$ (Weak) (c) $20(x^\top\theta)^2$ (Average) (d) $20(x^\top\theta)^2$ (Weak)

(e) $30(x^\top\theta)^2$ (Average) (f) $30(x^\top\theta)^2$ (Weak) (g) $40(x^\top\theta)^2$ (Average) (h) $40(x^\top\theta)^2$ (Weak)

Figure 5: Comparisons of cumulative regret (average and weak) of different dueling bandits algorithms for different form of Square function, i.e., $a(x^\top\theta)^2$, where $a = \{10, 20, 30, 40\}$.

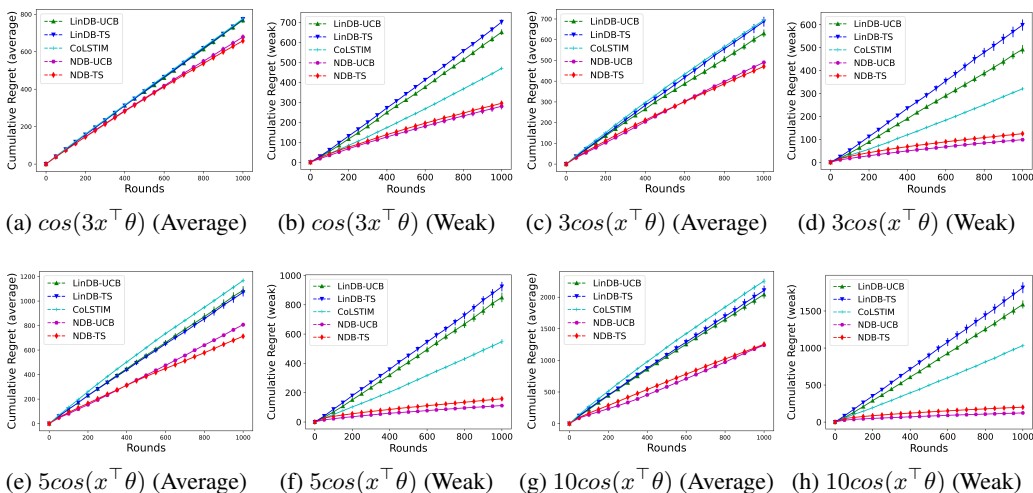

(a) $cos(3x^\top\theta)$ (Average) (b) $cos(3x^\top\theta)$ (Weak) (c) $3cos(x^\top\theta)$ (Average) (d) $3cos(x^\top\theta)$ (Weak)

(e) $5cos(x^\top\theta)$ (Average) (f) $5cos(x^\top\theta)$ (Weak) (g) $10cos(x^\top\theta)$ (Average) (h) $10cos(x^\top\theta)$ (Weak)

Figure 6: Comparisons of cumulative regret (average and weak) of different dueling bandits algorithms for different form of Cosine function, i.e., $cos(3x^\top\theta)$ and $bcos(x^\top\theta)$, where $b = \{3, 5, 10\}$.

## C.1    COMPUTATIONAL EFFICIENCY AND SCALABILITY

To discuss the computational efficiency and scalability of our proposed algorithms, we consider the following two key aspects:

**Size of the neural network (NN):** The primary computational complexity of our proposed algorithms from the NN used to estimate the latent reward function. Let $d$ be the dimension of the context-arm feature vector (input dimension for the NN). Assume the NN has $L$ hidden layers, each with $m$ neurons. Then, the inference cost from the input layer to the first hidden layer is $O(dm)$, the inference cost for hidden layers is $O(Lm^2)$, and the inference cost for the final layer is $O(m)$. Therefore, the overall inference cost for each context-arm pair is $O(dm + Lm^2 + m)$. Note that $p = dm + Lm^2 + m$ is the total number of NN parameters. The training time for the NN is $O(ECLm^2)$, where $E$ is

the number of training epochs and $C$ is the number of context-arm pairs observations. Therefore, choosing the appropriate size of NN for the given problem is very important, as having smaller NNs may not accurately approximate the latent reward function, while larger NNs can incur significant training and inference costs.

**Number of contexts and arms:** Let $K$ be the number of arms and $p$ be the total number of NN parameters. Since we use the gradients with respect to the current estimate of NN as context-arm features, the gradient computation cost for each context is $O(Kdp)$, where $d$ is the dimension of the context-arm feature vector. The computational costs for computing reward estimate and confidence terms are $O(Kp)$ and $O(Kp^2)$, respectively. For selection arms, the computational cost for the first arm is $O(Kp + K)$, which includes estimating the reward for each context-arm pair $O(Kp)$ and then selecting the arm with the highest estimated reward $O(K)$; and the computational cost for the second arm is $O(Kp + (K-1)p^2)$, i.e., computing estimated reward $O(Kp)$ and confidence terms $O((K-1)p^2)$ for all arm with respect to the first selected arm Thus, the overall computational cost of our proposed algorithms for selecting a pair of arms for each context is $O(Kdp + Kp^2)$. Since each context-arm pair is independent of others, each arm's gradients, reward estimate, and optimistic term can be computed in parallel. Consequently, the computational cost for selecting a pair of arms for each context can reduced to $O(dp + p^2)$.

## C.2 COMPARISION WITH EXISTING CONTEXTUAL DUELING BANDIT ALGORITHMS

Saha (2021) proposed a contextual linear dueling bandits algorithm using a logistic link function with pairwise and subset-wise preference feedback. Whereas Bengs et al. (2022) generalized the setting to the contextual linear stochastic transitivity model, and Li et al. (2024) proposed an algorithm based on Feel-Good Thompson Sampling (Zhang, 2022). These existing works only consider the linear reward function, which may not be practical in many real-life applications. Our work fills this gap in the literature and generalizes the existing setting by considering the non-linear reward function in contextual dueling bandits. Furthermore, the arm-selection strategies are different, as Saha (2021) and Li et al. (2024) select the most informative pair of arms, and Bengs et al. (2022) select the first arm optimistically and select the second arm that beats the first arm optimistically. In contrast, we select the first arm greedily to ensure the best-performing arm is always selected and then the second arm optimistically to focus on exploration, making the arm-selection process computationally efficient as our algorithms only need to compare $K - 1$ pairs instead of all possible pairs (i.e., $O(K^2)$). Since we use an NN-based reward estimator, our optimistic term calculations for the context-arm pairs differ from the existing work. Therefore, our regret analysis uses techniques different from existing algorithms for contextual dueling bandits.

## C.3 WHAT IF FIRST AND SECOND ARM ARE SAME

Since our proposed algorithms have a sub-linear regret guarantee, they will eventually select the best arm and recommend it in duels. This implies that for a given context, both the first and second arms chosen may ultimately be the same, representing the best-performing arm. Therefore, the first and second arms can be the same in our proposed algorithms. However, observations in which both arms are the same do not provide any useful preference information for estimating the latent reward function, as no comparison is being made. Including such observations for estimating latent reward function would only contribute to noise and result in a constant loss. Thus, we have to exclude these observations from the reward estimation.

# D  NEURAL CONTEXTUAL BANDITS WITH BINARY FEEDBACK

We extend our results to the neural contextual bandit problem in which a learner only observes binary feedback for the selected arms (note that the learner only selects one arm in every iteration). Observing binary feedback is very common in many real-life applications, e.g., click or not in online recommendation and treatment working or not in clinical trials (Li et al., 2017; Faury et al., 2020).

**Contextual bandits with binary feedback.** We consider a contextual bandit problem with binary feedback. In this setting, we assume that the action set is denoted by $\mathcal{A}$. Let $\mathcal{X}_t \subset \mathbb{R}^d$ denote the set of all context-arm feature vectors in the round $t$ and $x_{t,a}$ represent the context-arm feature

vector for context $c_t$ and an arm $a \in \mathcal{A}$. At the beginning of round $t$, the environment generates context-arm feature vectors $\{x_{t,a}\}_{a \in \mathcal{A}}$ and the learner selects an arm $a_t$, whose corresponding context-arm feature vector is given by $x_{t,a}$. After selecting the arm, the learner observes a stochastic binary feedback $y_t \in \{0, 1\}$ for the selected arm. We assume the binary feedback follows a Bernoulli distribution, where the probability of $y_t = 1$ for context-arm feature vector $x_{t,a}$ is given by $\mathbb{P}\{y_t = 1 | x_{t,a}\} = \mu(f(x_{t,a}))$, where $\mu : \mathbb{R} \to [0, 1]$ is a continuously differentiable and Lipschitz with constant $L_\mu$, e.g., logistic function, i.e., $\mu(x) = 1/(1 + e^{-x})$. The link function must also satisfy $\kappa_\mu \doteq \inf_{x \in \mathcal{X}} \dot{\mu}(f(x)) > 0$ for all arms.

**Performance measure.** The learner's goal is to select the best arm for each context, denoted by $x_t^\star = \arg\max_{x \in \mathcal{X}_t} f(x)$. Since the reward function $f$ is unknown, the learner uses available observations $\{x_{s,a}, y_s\}_{s=1}^{t-1}$ to estimate the function $f$ and then use the estimated function to select the arm $a_t$ for context $x_t$. After selecting the arm $a_t$, the learner incurs an instantaneous regret, $r_t = \mu(f(x_t^\star)) - \mu(f(x_{t,a}))$. For $T$ contexts, the (cumulative) regret of a policy that selects action $a_t$ for a context observed in round $t$ is given by $\mathfrak{R}_T = \sum_{t=1}^{T} r_t = \sum_{t=1}^{T} [\mu(f(x_t^\star)) - \mu(f(x_{t,a}))]$. Any good policy should have sub-linear regret, i.e., $\lim_{T \to \infty} \mathfrak{R}_T / T = 0$. Having sub-linear regret implies that the policy will eventually select the best arm for the given contexts.

## D.1 REWARD FUNCTION ESTIMATION USING NEURAL NETWORK AND OUR ALGORITHMS

To estimate the unknown reward function $f$, we use a fully connected neural network (NN) with parameters $\theta$ as used in the Section 3. The context-arm feature vector selected by the learner in round $s$ is denoted by $x_{s,a} \in \mathcal{X}_s$, and the observed stochastic binary feedback is denoted by $y_s$. At the beginning of round $t$, we use the current history of observations $\{(x_{s,a}, y_s)\}_{s=1}^{t-1}$ and use it to train the neural network (NN) by minimizing the following loss function (using gradient descent):

$$\mathcal{L}_t(\theta) = -\frac{1}{m} \sum_{s=1}^{t-1} \left[ y_s \log \mu(h(x_{s,a}; \theta)) + (1 - y_s) \log(1 - \mu(h(x_{s,a}; \theta))) \right] + \frac{\lambda \|\theta - \theta_0\|_2^2}{2}, \quad (33)$$

where $\theta_0$ represents the initial parameters of the NN. With the trained NN, we use UCB- and TS-based algorithms that handle the exploration-exploitation trade-off efficiently.

**UCB-based algorithm.** We propose a UCB-based algorithm named **NCBF-UCB**, which works as follows: At the beginning of the round $t$, it trains the NN using available observations. After receiving a context, the algorithm selects the arm optimistically as follows:

$$x_{t,a} = \arg\max_{x \in \mathcal{X}_t} [h(x; \theta_t) + \nu_T \sigma_{t-1}(x)], \quad (34)$$

where $\sigma_{t-1}^2(x) \doteq \frac{\lambda}{\kappa_\mu} \left\| \frac{g(x; \theta_0)}{\sqrt{m}} \right\|_{V_{t-1}^{-1}}^2$, in which $V_t \doteq \sum_{s=1}^{t} g(x; \theta_0) g(x; \theta_0)^\top \frac{1}{m} + \frac{\lambda}{\kappa_\mu} \mathbf{I}$, $\nu_T \doteq (\beta_T + B\sqrt{\lambda/\kappa_\mu} + 1)\sqrt{\kappa_\mu/\lambda}$ in which $\beta_T \doteq \frac{1}{\kappa_\mu} \sqrt{\widetilde{d}_b + 2\log(1/\delta)}$ and $\widetilde{d}_b$ is the *effective dimension*. We define the effective dimension later in this section (see Eq. (35)), which is different from Eq. (4).

---

**NCBF-UCB** Algorithm for Neural Contextual Bandits with Binary Feedback based on UCB

---

**Tuning parameters:** $\delta \in (0, 1)$ and $\lambda > 0$
2: **for** $t = 1, \ldots, T$ **do**
3:  Train the NN using $\{(x_{s,a}, y_s)\}_{s=1}^{t-1}$ by minimizing the loss function defined in Eq. (33)
4:  Receive a context and $\mathcal{X}_t$ denotes the corresponding context-arm feature vectors
5:  Select $x_{t,a} = \arg\max_{x \in \mathcal{X}_t} [h(x; \theta_t) + \nu_T \sigma_{t-1}(x)]$ (as given in Eq. (34))
6:  Observe preference feedback binary $y_t$
7: **end for**

---

**TS-based algorithm.** We also propose TS-based algorithm named **NCBF-TS**, which is similar to **NCBF-UCB** except to select the arm $x_{t,a}$, it firstly samples a reward $r_t(x) \sim \mathcal{N}(h(x; \theta_t), \nu_T^2 \sigma_{t-1}^2(x))$ for every arm $x \in \mathcal{X}_t$ and then selects the arm $x_{t,a} = \arg\max_{x \in \mathcal{X}_t} r_t(x)$.

**Regret analysis.** Let $K$ denote the finite number of available arms. Our analysis here makes use of the same assumptions as the analysis in Section 3 (i.e., Assumption 1 and Assumption 2). Let $\mathbf{H}_b \doteq \sum_{s=1}^{T} \sum_{i=1}^{K} g(x_{s,i}; \theta_0) g(x_{s,i}; \theta_0)^\top \frac{1}{m}$. We now define the *effective dimension* as follows:

$$\widetilde{d}_b = \log \det \left( \frac{\kappa_\mu}{\lambda} \mathbf{H}_b + \mathbf{I} \right). \tag{35}$$

Compared to $\mathbf{H}'$ defined in Section 3.3, $\mathbf{H}_b$ contains only $T \times K$ contexts, which is less than the $T \times K \times (K-1)$ contexts in $\mathbf{H}'$. Therefore, our $\widetilde{d}_b$ is expected to be generally smaller than in the neural dueling bandit feedback, as binary reward is more informative than preference feedback. A key step in our proof is that minimizing the loss function Eq. (1) allows us to achieve the following:

$$\frac{1}{m} \sum_{s=1}^{t-1} [\mu(h(x_{s,a}; \theta_t)) - y_s] g(x_{s,a}; \theta_t) + \lambda(\theta_t - \theta_0) = 0. \tag{36}$$

We use the above fact to prove the following confidence ellipsoid result as done in linear reward function (Li et al., 2017; Jun et al., 2017; Faury et al., 2020).

**Theorem 4.** *Let $\delta \in (0,1)$, $\varepsilon'_{m,t} \doteq C_2 m^{-1/6} \sqrt{\log m} L^3 \left( \frac{t}{\lambda} \right)^{4/3}$ for some absolute constant $C_2 > 0$. As long as $m \geq poly(T, L, K, 1/\kappa_\mu, L_\mu, 1/\lambda_0, 1/\lambda, \log(1/\delta))$, then with probability of at least $1 - \delta$, we have*

$$|f(x) - h(x; \theta_t)| \leq \nu_T \sigma_{t-1}(x) + \varepsilon'_{m,t},$$

*for all $x \in \mathcal{X}_t, t \in [T]$.*

Similar to Theorem 1, as long as the NN is wide enough (i.e., if the conditions on $m$ in Eq. (8) are satisfied. More details are in Appendix A), we have that $\varepsilon'_{m,t} = \mathcal{O}(1/T)$. Also note that in contrast to Theorem 1 whose confidence ellipsoid is in terms of reward differences, our confidence ellipsoid in Theorem 4 is in terms of the value of the reward function. This is because in contrast to neural dueling bandits (Section 3), here we get to collect an observation for every selected arm.

In the following results, we state the regret upper bounds of our proposed algorithms for neural contextual bandits with binary feedback.

**Theorem 5 (NCBF-UCB).** *Let $\lambda > \kappa_\mu$, $B$ be a constant such that $\sqrt{2\mathbf{h}^\top \mathbf{H}^{-1} \mathbf{h}} \leq B$, and $c_0 > 0$ be an absolute constant such that $\frac{1}{m} \|g(x_{s,i}; \theta_0)\|_2^2 \leq c_0, \forall x \in \mathcal{X}_t, t \in [T]$. For $m \geq poly(T, L, K, 1/\kappa_\mu, L_\mu, 1/\lambda_0, 1/\lambda, \log(1/\delta))$, then with probability of at least $1 - \delta$, we have*

$$\mathfrak{R}_T = \widetilde{O} \left( \left( \frac{\sqrt{\widetilde{d}_b}}{\kappa_\mu} + B \sqrt{\frac{\lambda}{\kappa_\mu}} \right) \sqrt{T \widetilde{d}_b} \right)$$

**Theorem 6 (NCBF-TS).** *Under the conditions as those in Theorem 5 holds, then with probability of at least $1 - \delta$, we have*

$$\mathfrak{R}_T = \widetilde{O} \left( \left( \frac{\sqrt{\widetilde{d}_b}}{\kappa_\mu} + B \sqrt{\frac{\lambda}{\kappa_\mu}} \right) \sqrt{T \widetilde{d}_b} \right).$$

Note that in terms of asymptotic dependencies (ignoring the log factors), our UCB- and TS- algorithms have similar growth rates. All missing proofs and additional details are in Appendix D.2.

**Comparison with Neural Bandits.** The regret upper bounds of our **NCBF-UCB** and **NCBF-TS** algorithms are worse than the regret of NeuralUCB Zhou et al. (2020) and NeuralTS Zhang et al. (2021): $\widetilde{O}(\widetilde{d}_b \sqrt{T})$ (with $\kappa_\mu = 1$) because of our extra dependency on $\kappa_\mu$ and $L_\mu$.[3] Specifically, note that $\kappa_\mu < 1$, therefore, the regret bounds in Theorem 5 and Theorem 6 are increased as a result of the dependency on $\kappa_\mu$. In addition, the dependency on $L_\mu$ also places an extra requirement on the width

---

[3]Note that our effective dimension $\widetilde{d}_b$ is defined using $\mathbf{H}_b$ Eq. (35), while the effective dimension $\widetilde{d}'$ in Zhou et al. (2020) and Zhang et al. (2021) are defined using $\mathbf{H}$. However, as we have discussed in Footnote 2, $\widetilde{d}'$ has the same order of growth as $\log \det (\mathbf{H}_b/\lambda + \mathbf{I})$. So, our regret upper bounds are comparable with those from Zhou et al. (2020) and Zhang et al. (2021).

$m$ of the NN. Therefore, our regret bounds are worse than that of standard neural bandit algorithms that do not depend on $\kappa_\mu$ and $L_\mu$. This can be attributed to the additional difficulty of our problem setting, i.e., we only have access to binary feedback, whereas standard neural bandits Zhou et al. (2020); Zhang et al. (2021) can use continuous observations.

Also note that our regret upper bounds here (Theorem 5 and Theorem 6) are expected to be smaller than those of neural dueling bandits (Theorem 2 and Theorem 3), because $\widetilde{d}_b$ here is likely to be smaller than $\widetilde{d}$ from Theorem 2 and Theorem 3. This may be attributed to the extra difficulty in the feedback in neural dueling bandits, i.e., only pairwise comparisons are available.

## D.2 THEORETICAL ANALYSIS

In this section, we show the proof of Theorem 5 and Theorem 6 for neural contextual bandits with binary feedback (Appendix D). We can largely reuse the proof from Appendix A, and here we will only highlight the changes we need to make to the proof in Appendix A.

To begin with, in our analysis here, we adopt the same requirement on the width of the NN specified in Eq. (8). First of all, Lemma 2 still holds in this case, which allows us to approximate the unknown utility function $f$ using a linear function. It is also easy to verify that Lemma 3 still holds. As a consequence, Lemma 4 and Lemma 5 both hold naturally.

### D.2.1 PROOF OF CONFIDENCE ELLIPSOID

Similar to the our proof in Appendix A.2, in iteration $s$, we denote $\varphi'_s \triangleq g(x_s; \theta_0)$, $\widetilde{\varphi}'_s \triangleq g(x_s; \theta_t)$, and $\widetilde{h}_{s,t} \triangleq h(x_s; \theta_t)$. Here we show how the proof of Lemma 6 should be modified. For any $\theta_{f'} \in \mathbb{R}^p$, define

$$G_t(\theta_{f'}) \triangleq \frac{1}{m} \sum_{s=1}^{t-1} \Big[ \mu\left(\langle \theta_{f'} - \theta_0, \varphi'_s \rangle\right) - \mu\left(\langle \theta_f - \theta_0, \varphi'_s \rangle\right) \Big] \varphi'_s + \lambda(\theta_{f'} - \theta_0). \quad (37)$$

Note that the definition of $G_t$ in Eq. (37) is exactly the same as that in Eq. (9), except that here we use a modified definition of $\varphi'_s$. Note that here $V_t$ is defined as $V_t \doteq \sum_{s=1}^{t} g(x_s; \theta_0) g(x_s; \theta_0)^\top \frac{1}{m} + \frac{\lambda}{\kappa_\mu} \mathbf{I} = \sum_{s=1}^{t} \varphi'_s \varphi'^\top_s \frac{1}{m} + \frac{\lambda}{\kappa_\mu} \mathbf{I}$. In addition, the definition of $f_{t,s}$ remains: $f_{t,s} = \langle \theta_t - \theta_0, \varphi'_s \rangle$. With the modified definitions of $V_{t-1}$, we can easily show that Lemma 7 remains valid. Note that here the binary observation can be expressed as $y_s = \mu(f(x_s)) + \varepsilon_s$, in which $\varepsilon_s$ is the observation noise. It is easy to verify that the decomposition in Eq. (12) remains valid.

Next, defining $A_1$ and $A_2$ in the same way as Eq. (13), it is easy to verify that Eq. (14) is still valid. Note that during the proof of Eq. (14), we have made use of Eq. (36), which allows us to ensure the validity of $\frac{1}{m} \sum_{s=1}^{t-1} \left( \mu(\widetilde{h}_{s,t}) - y_s \right) \widetilde{\varphi}'_s + \lambda(\theta_t - \theta_0) = 0$ in Eq. (15). This is ensured by the way we train our neural network with the binary observations. Next, we derive an upper bound on the norm of $A_1$. To begin with, we have that

$$\|\varphi'_s - \widetilde{\varphi}'_s\|_2 = \|g(x_s; \theta_0) - g(x_s; \theta_t)\|_2$$
$$\leq C_1 m^{1/3} \sqrt{\log m} \left( \frac{Ct}{\lambda} \right)^{1/3} L^{7/2},$$

in which the inequality follows from Lemma 4. Then, the proof in Eq. (16) can be reused to show that

$$\|A_1\|_2 = \left\| \frac{1}{m} \sum_{s=1}^{t-1} \left( \mu(f_{t,s}) - y_s \right) \left( \varphi'_s - \widetilde{\varphi}'_s \right) \right\|_2 \leq m^{-2/3} \sqrt{\log m} t^{4/3} C_1 \widetilde{C}^{1/3} \lambda^{-1/3} L^{7/2}.$$

Note that the upper bound above is smaller than that from Eq. (16) by a factor of 2. Similarly, we can follow the proof of Eq. (17) to derive an upper bound on the norm of $A_2$:

$$\|A_2\|_2 = \left\| \frac{1}{m} \sum_{s=1}^{t-1} \left( \mu(f_{t,s}) - \mu(\widetilde{h}_{s,t}) \right) \widetilde{\varphi}'_s \right\|_2 \leq 2 L_\mu C_2 C_3 \widetilde{C}^{4/3} m^{-2/3} \sqrt{\log m} t^{7/3} L^{7/2} \lambda^{-4/3},$$

in which the upper bound is also smaller than that from Eq. (17) by a factor of 2. As a result, defining $\varepsilon_{m,t}$ in the same way as Eq. (19) (except that the second and third terms in $\varepsilon_{m,t}$ are reduced by a factor of 2), we can show that Eq. (20) is still valid:

$$\sqrt{m} \left\| \theta_f - \theta_t \right\|_{V_{t-1}} \leq \frac{1}{\kappa_\mu} \left\| \sum_{s=1}^{t-1} \varepsilon_s \varphi'_s \frac{1}{\sqrt{m}} \right\|_{V_{t-1}^{-1}} + B \sqrt{\frac{\lambda}{\kappa_\mu}} + 1. \tag{38}$$

Now we derive an upper bound on the first term in Eq. (38) in the next lemma, which is proved by modifying the proof of Lemma 8.

**Lemma 15.** *Let* $\beta_T \doteq \frac{1}{\kappa_\mu} \sqrt{\widetilde{d}_b + 2 \log(1/\delta)}$. *With probability of at least* $1 - \delta$, *we have that*

$$\frac{1}{\kappa_\mu} \left\| \sum_{s=1}^{t-1} \varepsilon_s \varphi'_s \frac{1}{\sqrt{m}} \right\|_{V_{t-1}^{-1}} \leq \beta_T.$$

*Proof.* Note that in the main text, we have the following modified definitions: $\mathbf{H}_b \doteq \sum_{s=1}^{T} \sum_{i \in K} g(x_{s,i}; \theta_0) g(x_{s,i}; \theta_0)^\top \frac{1}{m}$, and $\widetilde{d}_b = \log \det \left( \frac{\kappa_\mu}{\lambda} \mathbf{H}_b + \mathbf{I} \right)$.

To begin with, we derive an upper bound on the log determinant of the matrix $V_t \doteq \sum_{s=1}^{t} g(x_s; \theta_0) g(x_s; \theta_0)^\top \frac{1}{m} + \frac{\lambda}{\kappa_\mu} \mathbf{I}$. Now the determinant of $V_t$ can be upper-bounded as

$$\det(V_t) = \det \left( \sum_{s=1}^{t} g(x_s; \theta_0) g(x_s; \theta_0)^\top \frac{1}{m} + \frac{\lambda}{\kappa_\mu} \mathbf{I} \right)$$

$$\leq \det \left( \sum_{s=1}^{T} \sum_{i \in K} g(x_{s,i}; \theta_0) g(x_{s,i}; \theta_0)^\top \frac{1}{m} + \frac{\lambda}{\kappa_\mu} \mathbf{I} \right)$$

$$= \det \left( \mathbf{H}_b + \frac{\lambda}{\kappa_\mu} \mathbf{I} \right).$$

Recall that in our algorithm, we have set $V_0 = \frac{\lambda}{\kappa_\mu} \mathbf{I}_p$. This leads to

$$\log \frac{\det V_t}{\det V_0} \leq \log \frac{\det \left( \mathbf{H}_b + \frac{\lambda}{\kappa_\mu} \mathbf{I} \right)}{\det V_0}$$

$$= \log \frac{(\lambda/\kappa_\mu)^p \det \left( \frac{\kappa_\mu}{\lambda} \mathbf{H}_b + \mathbf{I} \right)}{(\lambda/\kappa_\mu)^p}$$

$$= \log \det \left( \frac{\kappa_\mu}{\lambda} \mathbf{H}_b + \mathbf{I} \right).$$

Next, following the same line of argument in the proof of Lemma 8 about the observation noise $\varepsilon$, we can easily show that in this case of neural contextual bandits with binary observation, the sequence of noise $\{\varepsilon_s\}$ is also conditionally 1-sub-Gaussian.

Next, making use of the 1-sub-sub-Gaussianity of the sequence of noise $\{\varepsilon_s\}$ and Theorem 1 from Abbasi-Yadkori et al. (2011), we can show that with probability of at least $1 - \delta$,

$$\left\| \sum_{s=1}^{t-1} \varepsilon_s \varphi'_s \frac{1}{\sqrt{m}} \right\|_{V_{t-1}^{-1}} \leq \sqrt{\log \left( \frac{\det V_{t-1}}{\det V_0} \right) + 2 \log(1/\delta)}$$

$$\leq \sqrt{\log \det \left( \frac{\kappa_\mu}{\lambda} \mathbf{H}_b + \mathbf{I} \right) + 2 \log(1/\delta)}$$

$$\leq \sqrt{\widetilde{d}_b + 2 \log(1/\delta)},$$

in which we have made use of the definition of the effective dimension $\widetilde{d}_b = \log \det \left( \frac{\kappa_\mu}{\lambda} \mathbf{H}_b + \mathbf{I} \right)$. This completes the proof. □

Finally, plugging Lemma 15 into Eq. (38) allows us to show that Lemma 6 remains valid:

$$\sqrt{m} \left\| \theta_f - \theta_t \right\|_{V_{t-1}} \le \beta_T + B \sqrt{\frac{\lambda}{\kappa_\mu}} + 1, \qquad \forall t \in [T]. \tag{39}$$

Now we can prove the confidence ellipsoid in Theorem 4:

**Theorem 4.** *Let $\delta \in (0, 1)$, $\varepsilon'_{m,t} \doteq C_2 m^{-1/6} \sqrt{\log m} L^3 \left( \frac{t}{\lambda} \right)^{4/3}$ for some absolute constant $C_2 > 0$. As long as $m \ge poly(T, L, K, 1/\kappa_\mu, L_\mu, 1/\lambda_0, 1/\lambda, \log(1/\delta))$, then with probability of at least $1 - \delta$, we have*

$$|f(x) - h(x; \theta_t)| \le \nu_T \sigma_{t-1}(x) + \varepsilon'_{m,t},$$

*for all $x \in \mathcal{X}_t, t \in [T]$.*

*Proof.* Denote $\varphi(x) = g(x; \theta_0)$. Recall that Lemma 2 tells us that $f(x) = \langle g(x; \theta_0), \theta_f - \theta_0 \rangle = \langle \varphi(x), \theta_f - \theta_0 \rangle$ for all $x \in \mathcal{X}_t, t \in [T]$. To begin with, for all $x \in \mathcal{X}_t, t \in [T]$ we have that

$$
\begin{aligned}
|f(x) - \langle \varphi(x), \theta_t - \theta_0 \rangle| &= |\langle \varphi(x), \theta_f - \theta_0 \rangle - \langle \varphi(x), \theta_t - \theta_0 \rangle| \\
&= |\langle \varphi(x), \theta_f - \theta_t \rangle| \\
&= |\langle \frac{1}{\sqrt{m}} \varphi(x), \sqrt{m} (\theta_f - \theta_t) \rangle| \\
&\le \left\| \frac{1}{\sqrt{m}} \varphi(x) \right\|_{V_{t-1}^{-1}} \sqrt{m} \left\| \theta_f - \theta_t \right\|_{V_{t-1}} \\
&\le \left\| \frac{1}{\sqrt{m}} \varphi(x) \right\|_{V_{t-1}^{-1}} \left( \beta_T + B \sqrt{\frac{\lambda}{\kappa_\mu}} + 1 \right),
\end{aligned}
\tag{40}
$$

in which we have used Lemma 6 (reproduced in Eq. (39)) in the last inequality. Now making use of the equation above and Lemma 5, we have that

$$
\begin{aligned}
&|f(x) - h(x; \theta_t)| \\
&= |f(x) - \langle \varphi(x), \theta_t - \theta_0 \rangle + \langle \varphi(x), \theta_t - \theta_0 \rangle - h(x; \theta_t)| \\
&\le |f(x) - \langle \varphi(x), \theta_t - \theta_0 \rangle| + |\langle \varphi(x), \theta_t - \theta_0 \rangle - h(x; \theta_t)| \\
&\le \left\| \frac{1}{\sqrt{m}} \varphi(x) \right\|_{V_{t-1}^{-1}} \left( \beta_T + B \sqrt{\frac{\lambda}{\kappa_\mu}} + 1 \right) + \varepsilon'_{m,t},
\end{aligned}
$$

in which the last inequality follows from Eq. (40) and Lemma 5.

Recall that we have defined in the paper $\sigma_{t-1}^2(x) \doteq \frac{\lambda}{\kappa_\mu} \left\| \frac{g(x; \theta_0)}{\sqrt{m}} \right\|_{V_{t-1}^{-1}}^2$, and $\nu_T \doteq (\beta_T + B\sqrt{\lambda/\kappa_\mu} + 1)\sqrt{\kappa_\mu/\lambda}$ in which $\beta_T \doteq \frac{1}{\kappa_\mu} \sqrt{\widetilde{d}_b + 2\log(1/\delta)}$. This completes the proof of Theorem 4. $\square$

### D.2.2 REGRET ANALYSIS

Now we can analyze the instantaneous regret:

$$
\begin{aligned}
r_t &= f(x_t^*) - f(x_t) \\
&\le h(x_t^*; \theta_t) + \nu_T \sigma_{t-1}(x_t^*) + \varepsilon'_{m,t} - h(x_t; \theta_t) + \nu_T \sigma_{t-1}(x_t) + \varepsilon'_{m,t} \\
&\le h(x_t; \theta_t) + \nu_T \sigma_{t-1}(x_t) - h(x_t; \theta_t) + \nu_T \sigma_{t-1}(x_t) + 2\varepsilon'_{m,t} \\
&= 2\nu_T \sigma_{t-1}(x_t) + 2\varepsilon'_{m,t}.
\end{aligned}
$$

Next, the subsequent analysis in Appendix A.3 follows by replacing $\sigma_{t-1}(x_{t,1}, x_{t,2})$ by $\sigma_{t-1}(x_t)$, which allows us to show that

$$\sum_{t=1}^{T} \sigma_{t-1}^2(x_t) \le 2c_0 \frac{\lambda}{\kappa_\mu} \widetilde{d}_b.$$

Finally, we can derive an upper bound on the cumulative regret:

$$
\Re_T = \sum_{t=1}^{T} r_t \le \sum_{t=1}^{T} \left( 2\nu_T \sigma_{t-1}(x_t) + 2\varepsilon'_{m,t} \right)
$$

$$
\le 2 \sum_{t=1}^{T} \nu_T \sigma_{t-1}(x_t) + 2 \sum_{t=1}^{T} \varepsilon'_{m,t}
$$

$$
\le 2\nu_T \sqrt{T \sum_{t=1}^{T} \sigma_{t-1}^2(x_t)} + 2T\varepsilon'_{m,T}
$$

$$
\le 2\nu_T \sqrt{T 2c_0 \frac{\lambda}{\kappa_\mu} \widetilde{d_b}} + 2T\varepsilon'_{m,T}.
$$

Again it can be easily verified that as long as the conditions on $m$ specified in Eq. (8) are satisfied (i.e., as long as the NN is wide enough), we have that $2T\varepsilon'_{m,T} \le 1$. Also recall that $\beta_T = \widetilde{\mathcal{O}}(\frac{1}{\kappa_\mu}\sqrt{\widetilde{d_b}})$, and $\nu_T \doteq (\beta_T + B\sqrt{\lambda/\kappa_\mu} + 1)\sqrt{\kappa_\mu/\lambda} = \widetilde{O}(\frac{1}{\sqrt{\kappa_\mu}}\sqrt{\widetilde{d_b}} + B + \sqrt{\kappa_\mu/\lambda})$. This allows us to simplify the regret upper bound to be

$$
\Re_T \le 2\nu_T \sqrt{T 2c_0 \frac{\lambda}{\kappa_\mu} \widetilde{d_b} + 1}
$$

$$
= \widetilde{\mathcal{O}}\left( \left( \frac{\sqrt{\widetilde{d_b}}}{\kappa_\mu} + B\sqrt{\frac{\lambda}{\kappa_\mu}} \right) \sqrt{\widetilde{d_b} T} \right).
$$

The proof for the Thompson sampling algorithm follows a similar spirit, which we omit here.

### D.3 EXPERIMENTS FOR NEURAL GLM BANDITS

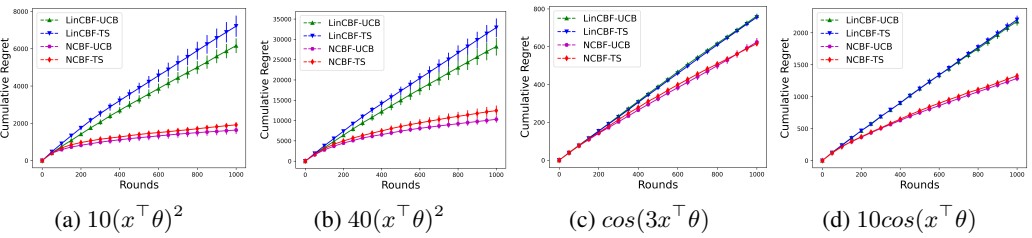

(a) $10(x^\top\theta)^2$  (b) $40(x^\top\theta)^2$  (c) $cos(3x^\top\theta)$  (d) $10cos(x^\top\theta)$

Figure 7: Comparing cumulative regret of GLM bandits algorithms for non-linear reward functions.

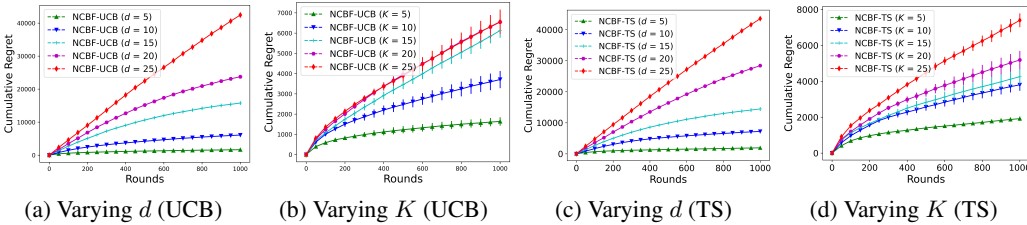

(a) Varying $d$ (UCB)  (b) Varying $K$ (UCB)  (c) Varying $d$ (TS)  (d) Varying $K$ (TS)

Figure 8: Cumulative regret of Algorithm **NCBF-UCB** and **NCBF-TS** vs. different number of arms ($K$) and dimension of the context-arm feature vector ($d$) for Square reward function (i.e., $10(x^\top\theta)^2$).

# E    THEORETICAL INSIGHTS FOR REINFORCEMENT LEARNING WITH HUMAN FEEDBACK

Our algorithms and theoretical results can also provide insights on the celebrated *reinforcement learning with human feedback* (RLHF) algorithm (Chaudhari et al., 2024), which has been the most widely used method for the alignment of large language models (LLMs). In RLHF, we are given a dataset of user preferences, in which every data point consists of a prompt and a pair of responses generated by the LLM, as well as a binary observation indicating which response is preferred by the user. Following our notations in Section 2, the action $x_{t,1}$ (resp. $x_{t,2}$) corresponds to the concatenation of the prompt and the first response (resp. second response). Of note, RLHF is also based on the assumption that the user preference over a pair of responses is governed by the BTL model (Section 2). That is, the binary observation $y_t$ is sampled from a Bernoulli distribution, in which the probability that the first response is preferred over the second response is given by $\mathbb{P}\{x_{t,1} \succ x_{t,2}\} = \mu\left(f(x_{t,1}) - f(x_{t,2})\right)$. Here $f$ is often referred to as the reward function, which is equivalent to the latent utility function $f$ in our setting (Section 2).

Typically, RLHF consists of two steps: *(a)* learning a reward model using the dataset of user preferences and *(b)* fine-tuning the LLM to maximize the learned reward model using reinforcement learning. In step *(a)*, same as our algorithms, *RLHF also uses an NN $h$ (which takes as input the embedding from a pre-trained LLM) to learn the reward model by minimizing the loss function* (1). The accuracy of the learned reward model is crucial for the success of RLHF (Chaudhari et al., 2024). Importantly, *our Theorem 1 provides a theoretical guarantee on the quality of the learned reward model $h$*, i.e., an upper bound on the estimation error of the estimated reward differences between any pair of responses. Therefore, our Theorem 1 provides a theoretically principled measure of the accuracy of the learned reward model in RLHF, which can potentially be used to evaluate the quality of the learned reward model.

In addition, some recent works have proposed the paradigm of online/iterative RLHF (Bai et al., 2022; Menick et al., 2022; Mehta et al., 2023; Ji et al., 2024; Das et al., 2024) to further improve the alignment of LLMs. In online RLHF, the RLHF procedure is repeated multiple times. Specifically, after an LLM is fine-tuned to maximize the learned reward model, it is then used to generate pairs of responses to be used to query the user for preference feedback; then, the newly collected preference data is added to the preference dataset to be used to train a new reward model, which is again used to fine-tune the LLM. In this case, as the alignment of the LLM is improved after every round, the newly generated responses by the improved LLM are expected to achieve progressively higher reward values, which have been shown to lead to better alignment of LLMs (Bai et al., 2022; Menick et al., 2022; Mehta et al., 2023; Ji et al., 2024; Das et al., 2024). In every round, we can let the LLM generate a large number of responses (i.e., the actions in our setting, see Section 2), from which *we can use our algorithms to select two responses $x_{t,1}$ and $x_{t,2}$ to be shown to the user for preference feedback*. In addition, our algorithm can also potentially be used to select the prompts shown to the user, which correspond to the contexts in our problem setting (Section 2). Our theoretical results guarantee that our algorithms can help select responses with high reward values (Theorem 2 and Theorem 3). Therefore, *our algorithms can be used to improve the efficiency of online RLHF*.

# F    RESPONSE OPTIMIZATION IN LARGE LANGUAGE MODELS (LLMS)

As demonstrated in recent works (Bai et al., 2022; Menick et al., 2022; Mehta et al., 2023; Das et al., 2024; Ji et al., 2024), using responses with higher reward values can significantly improve the alignment of LLMs, especially in online/iterative RLHF where the responses generated by the initial LLM tend to have low rewards. Motivated by this, we use our proposed algorithms, **NDB-UCB** and **NDB-TS**, to select the responses with higher estimated reward values for a given prompt. In our experiment, we use the following setting from Lin et al. (2024): In each iteration, a user provides a prompt (context in our setting) to ChatGPT, and then ChatGPT generates 50 responses (arms in our setting). We use Sentence-BER (Reimers & Gurevych, 2019) to get embedding representation for each prompt-response pair. Specifically, for iteration $t$, we use $x_{t,i} = (c_t, a_{t,i})$ to denote the embedding representation generated by Sentence-BERT, where $c_t$ denotes the prompt and $a_{t,i}$ represents the $i$-th response. Of note, we adopt a reward model that is pre-trained using the Anthropic Helpfulness and Harmlessness datasets (Bai et al., 2022). For every prompt-response pair $x_{t,i}$, we use the output from this pre-trained reward model as the value of the unknown reward function $f(x_{t,i})$. This approach

allows us to simulate the preference feedback in this experiment. Note that our approach to simulating preference feedback is common in the literature (Dwaracherla et al., 2024).

The embeddings $x_{t,i}$ are used as input to a trained neural network (NN) that estimates the latent reward model. Our proposed algorithms compute the reward estimate for each prompt-response pair to select the first response (using Eq. (2)) and then use it with the optimistic reward estimates of other responses to select the second response (using either Eq. (3) or TS-based selection criterion for the second response). This way, our algorithms will ensure the selection of responses with consistently high reward estimates. As shown in Fig. 9, we compare the score (calculated as the sum of rewards for selected responses divided by the number of iterations) after $T$ iterations (multiple of 10) of **NDB-UCB** and **NDB-TS** and against

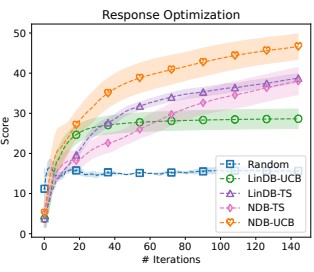

Figure 9: Scores of different algorithms for response optimization.

two baselines: Linear variants (which uses a linear model to estimate latent reward model instead of an NN): Lin-UCB and Lin-TS, and Random (where two responses are selected randomly). The results clearly show that our algorithms achieve the highest score (especially **NDB-UCB**), demonstrating the superior performance of our proposed algorithm in identifying high-reward responses.

The experimental results shown in Fig. 9 corroborate our theoretical results, which guarantee the selection of responses with high reward values, as also discussed in Section 4.2 and Appendix E. This result further shows that our proposed algorithms can be used to improve the quality of the human preference dataset, which could then be used to improve the efficiency of online RLHF/DPO for LLM alignment. We leave it to future works to verify that the preference dataset with high-reward responses collected using our algorithms can indeed lead to better LLM alignment, which is beyond the scope of the current work (since the focus of this work is primarily theoretical) and will require significantly more extensive experiments.

