# OpenReview forum: "Neural Dueling Bandits: Preference-Based Optimization with Human Feedback"
_ICLR.cc/2025/Conference — ICLR 2025 Poster_

### Official Review · Reviewer_kAG9 · 2024-10-31

**Soundness:** 3
**Presentation:** 2
**Contribution:** 3
**Rating:** 6
**Confidence:** 3

**Summary:**

This work studies dueling bandits based on neural networks. UCB and TS algorithms are suggested, and sublinear regret bounds are achieved for both algorithms. The theoretical analysis is robust as well.

**Strengths:**

This work provides theoretical results of two mainstream algorithms, UCB and TS, for neural dueling bandits. These results are significant contributions to bandits.

**Weaknesses:**

Overall, this paper is difficult to follow due to the use of numerous quantities and notations that are not explicitly defined. Given the long-standing history and inherent complexity of neural network analysis, it is understandable that some notations and intermediate reasoning steps are abbreviated. However, the reviewer believes that enhancing clarity would greatly improve readability for uninitiated readers.

L59-63: Could you be more specific?

L85-94: Since many readers may be unfamiliar with dual bandits, the description should be more detailed. For instance, the authors should clarify that the learner selects two arms from the action set with multiple arms, plays both arms, and receives feedback from both, which is captured as the preference y_t. It would be helpful to emphasize how this differs from standard contextual bandits. Additionally, the authors introduce the context-arm set X_t. As the reviewer understands, this is no different from setups where either an action or context set is considered as long as a context and action pair is in R^d. Is there any technical distinction between the two? If not, this setup seems to introduce unnecessary complexity.

L95-111:  The authors stated that the logistic function is considered as the link function, but stated in Assumption 1 that a class of link functions satisfying two conditions is considered in this work. For the reviewer, this description is confusing.

L112-120: As these regrets are not common in other literature, regret should be better explained with the basic ideas of regret.

Could you provide more specific practical applications of dueling bandits?

**Questions:**

The reviewer notes that while the suggested setup differs somewhat from previous works (Zhou et al., 2020; Zhang et al., 2021), the analysis appears to be quite similar, based on the main manuscript. Although certain distinctions are mentioned in the introduction, the reviewer finds it challenging to identify the specific differences. Could you clarify more explicitly the technical novelty of this work compared to the existing literature?

As the authors provide two algorithms, the reviewer would like to know some comparisons between these two. While some comparative simulation results are provided, could you also include a more qualitative analysis, incorporating some mathematical insights, to highlight the differences between the two algorithms?

For regret analysis, what kind of regret is analyzed between average instantaneous regret and weak
instantaneous regret?

For TS, what do we do if the first and second arms are the same?

Could the authors define the matrix H in any way? What are the differences between H and H’?

In Assumption 2, [x]_k = [x]_k+d/2, [] is used for multiple purposes, making some confusion. It should be defined clearly.

L191: Is \phi is g? Could the authors please explain why a different notation is needed if they are the same?

---

> ### Author Response · Authors · 2024-11-25
> **Clarifications about problem setup**
>
> We would like to thank the reviewer for taking the time to review our paper. We also thank the reviewer for acknowledging that our results are significant contributions to bandits. We have addressed each of the weaknesses and answer your questions as follows:
>
> > **L59-63: Could you be more specific?**
>
> In previous work, Saha (2021) and (Li et al., 2024) select the **most informative pair of arms**, while Bengs et al. (2022) select the **first arm optimistically and select the second arm that beats the first arm optimistically**. In contrast, our proposed algorithms select the **first arm greedily to ensure the best-performing arm is always selected and then the second arm optimistically to focus on exploration**. This strategy makes our arm-selection process computationally efficient as our algorithms only need to compare $(K-1)$ pairs instead of all possible pairs (i.e., $O(K^2)$). Additionally, since we use an NN-based reward estimator, our optimistic term calculations for the context-arm pairs differ from the existing algorithms. As a result, our regret analysis uses novel techniques different from existing algorithms for contextual dueling bandits.
> We have added this discussion in Appendix C.2 and a reference to this discussion in the introduction of our revised paper.
>
>
> > **L85-94: Since many readers may be unfamiliar with dual bandits, the description should be more detailed. For instance, the authors should clarify that the learner selects two arms from the action set with multiple arms, plays both arms, and receives feedback from both, which is captured as the preference y_t. It would be helpful to emphasize how this differs from standard contextual bandits.**
>
> We appreciate your suggestion to clarify the difference between contextual dueling bandits and standard contextual bandits. In the **contextual dueling bandits** problem, the learner **selects two arms for a given context and observes comparative preference feedback** over selected arms, indicating which arm was preferred. This problem differs from **standard contextual bandits** in which a learner **selects a single arm and observes an absolute numerical reward** for that arm. To clarify the difference, we added this discussion in the revised paper to highlight the difference between contextual dueling bandits and standard continual bandits.
>
>
> > **Additionally, the authors introduce the context-arm set X_t. As the reviewer understands, this is no different from setups where either an action or context set is considered as long as a context and action pair is in R^d. Is there any technical distinction between the two? If not, this setup seems to introduce unnecessary complexity.**
>
> Your understanding is correct. The set $X_t$ represents all the context-arm feature vectors in round $t$. While it is equivalent to setups where context-arm pairs are considered separately, our motivation for introducing this notation is to simplify and unify our mathematical representation. By grouping context and arm space into a single notation $X_t$, we avoid using separate notations for context and action spaces. **Using $X_t$ helps simplify the presentation and makes the paper's notation cleaner and more concise**. We hope this clarifies the reasoning behind our choice of notation.
>
>
> > **L95-111: The authors stated that the logistic function is considered as the link function, but stated in Assumption 1 that a class of link functions satisfying two conditions is considered in this work. For the reviewer, this description is confusing.**
>
> Our theoretical results are not limited to the BTL (Bradley-Terry-Luce) model but also hold for other preference models, such as the Thurstone-Mosteller model and Exponential Noise, as long as the stochastic transitivity property holds (Bengs et al., 2022). **To generalize our results across different preference models, we need to make standard assumptions on the function $\mu$**. The function $\mu$ is also referred to as a link function in the literature (Li et al., 2017; Bengs et al., 2022). The value of $\kappa_\mu$ and $L_\mu$ for the link function depends on the context space and preference model, e.g., $L_\mu \le 0.25$ for the BTL model.
>
>
> > **L112-120: As these regrets are not common in other literature, regret should be better explained with the basic ideas of regret.**
>
> The average and weak regret definitions used in the paper are **commonly used notions of regret in existing works** on contextual dueling bandits (Saha, 2021; Bengs et al., 2022; Li et al., 2024). Thus, we also consider average and weak cumulative regret and provide regret bounds for them. By definition, average regret is the difference between the maximum latent reward and the average of the latent rewards of the selected arms, while weak regret is the difference between the maximum latent reward and the maximum of the latent rewards of the selected arms. We have revised the paper to highlight these points.

---

> > ### Author Response · Authors · 2024-11-25
> > **Clarifications about practical applications and technical novelty**
> >
> > > **Could you provide more specific practical applications of dueling bandits?**
> >
> > Contextual dueling bandits are well-suited for practical applications where relative feedback (preferences) is easier to observe than absolute feedback. The following are examples of real-world practical applications where contextual dueling bandit algorithms can be used:
> >
> > - **Online Recommendations:** Comparing two items (e.g., products, movies, or songs) to determine user preferences and improve recommendation systems.
> >
> > - **Ranking Web Search Results:** Optimizing search algorithms by comparing search results based on user clicks to refine rankings.
> >
> > - **A/B Testing:** Evaluating two versions of a webpage, feature, or system to determine which performs better.
> >
> > - **Fine-Tuning Large Language Models (LLMs):** Leveraging human preferences to compare two outputs from an LLM to fine-tune LLM performance through preference-based feedback.
> >
> > - **Comparative Ratings:** Learning rankings by comparing preferences for restaurants, movies, or other rated items where numerical feedback may not be available.
> >
> >
> > > **The reviewer notes that while the suggested setup differs somewhat from previous works (Zhou et al., 2020; Zhang et al., 2021), the analysis appears to be quite similar, based on the main manuscript. Although certain distinctions are mentioned in the introduction, the reviewer finds it challenging to identify the specific differences. Could you clarify more explicitly the technical novelty of this work compared to the existing literature?**
> >
> >
> > Our work has the following technical novel contributions compared to existing works in neural bandits (Zhou et al., 2020; Zhang et al., 2021):
> >
> > - **Difference objective function compared to existing neural bandits:** The main difference between our work and the existing neural bandit (Zhou et al., 2020; Zhang et al., 2021) is the use of **cross-entropy loss** as an objective function for training the neural network to estimate the unknown latent reward function due to the preference feedback (0/1 feedback). Existing neural bandit works use root mean square error (RMSE) as an objective function to train neural networks due to the assumption of real-valued reward or feedback. Hence, their key results, especially the confidence ellipsoid, only hold for RMSE and can not be extended to our setting, which uses cross-entropy loss. Therefore, deriving the confidence ellipsoid for the latent reward function is the novel contribution of our paper.
> >
> > - **Novel theoretical contributions:** Our result, presented in Theorem 1, establishes an upper bound on the estimation error (i.e., represented as a confidence ellipsoid) for the difference between the reward values of any pair of arms predicted by the trained neural network. This result provides theoretical assurance regarding the accuracy of our trained neural network that uses preference feedback to minimize cross-entropy loss. Therefore, we can also use Theorem 1 to provide a theoretical guarantee of the quality of the learned reward model during RLHF (more discussion in Appendix E). Furthermore, we also adapt our results with the appropriate changes to the contextual bandit problem with binary feedback having a non-linear reward function to give an upper bound on the estimation error for the reward values of arms predicted by the trained neural network, and then using it to prove sub-linear regret guarantees (more details are given in Section 4.1 and Appendix D).

---

> ### Author Response · Authors · 2024-11-25
> **Clarifications about algorithms and theoretical results**
>
> > **As the authors provide two algorithms, the reviewer would like to know some comparisons between these two. While some comparative simulation results are provided, could you also include a more qualitative analysis, incorporating some mathematical insights, to highlight the differences between the two algorithms?**
>
> The only difference between the UCB-based algorithm (NDB-UCB) and the TS-based algorithm (NDB-TS) is **how these two algorithms select the second arm**.
>
> The second arm for NDB-UCB is selected (as given in Eq. (3)) as follows:
> $$
> x_{t,2} = \arg\max_{x\in\mathcal{X}_t} \left[ h(x;\theta_t) +  \nu_T\sigma _{t-1}(x,x_t)  \right].
> $$
>
> While the second arm for NDB-TS is selected (Line 202-203) as follows:
>
> $$
> x_{t,2} = \arg\max_{x\in\mathcal{X}_t} \mathcal{N}\left( h(x;\theta_t) - h(x_1;\theta_t),  \nu_T^2\sigma^2 _{t-1}(x,x_t)  \right). $$
>
> This difference leads to using different techniques for the regret analysis of NDB-UCB and NDB-TS, as given in Appendix A.3 and Appendix B, respectively.
>
>
> > **For regret analysis, what kind of regret is analyzed between average instantaneous regret and weak instantaneous regret?**
>
> By definition, weak regret is always smaller than average regret; hence, an upper bound for average regret is, by default, an upper bound for weak regret. Therefore, we have **analyzed the average cumulative regret** in our paper. Our results in Theorem 2 and Theorem 3 give upper bounds on the average cumulative regrets; hence, these cumulative regret upper bounds also hold for weak cumulative regret.
>
>
> > **For TS, what do we do if the first and second arms are the same?**
>
> Since our proposed algorithms have a sub-linear regret guarantee, they will eventually select the best arm and recommend it in duels. This implies that for a given context, both the first and second arms chosen may ultimately be the same, representing the best-performing arm. Therefore, the first and second arms can be the same in our proposed algorithms. However, observations in which both arms are the same do not provide any useful preference information for estimating the latent reward function, as no comparison is being made. Including such observations for estimating latent reward function would only contribute to noise and result in a constant loss. Thus, we have to exclude these observations from the reward estimation. We have added this discussion in Appendix C.3 of our revised paper.
>
>
> > **Could the authors define the matrix H in any way? What are the differences between H and H’?**
>
> Let $\left( x_{(n)} \right)_{n=1}^{TK}$  be a set of all possible context-arm feature vectors:
>
> $\left( x_{t,a}\right)_{1\le t \le T, 1\le a \le K}$, where $n=K(t-1) + a$.
>
> Define
>
>
> $$
>     \widetilde{\mathbf{H}}_{p,q}^{(1)} = \mathbf{\Sigma} _{p,q}^{(1)} = \langle x _{(p)}, x _{(q)}  \rangle ,
> $$
>
> $$
> \mathbf{A}_{p,q}^{(l)} =
> \begin{pmatrix}
>     \mathbf{\Sigma} _{p,p}^{(l)} & \mathbf{\Sigma} _{p,q}^{(l)} \\\\
>     \mathbf{\Sigma} _{p,q}^{(l)} & \mathbf{\Sigma} _{q,q}^{(l)}
> \end{pmatrix},
> $$
>
> $$
>     \mathbf{\Sigma} _{p,q}^{(l+1)} = 2\mathbb{E} _{(u,v)\sim\mathcal{N}(0,\mathbf{A} _{p,q}^{(l)} ) } [\max(u,0)\max(v,0)],
> $$
> and
>
> $$
>     \widetilde{\mathbf{H}} _{p,q}^{(l+1)} = 2\widetilde{\mathbf{H}} _{p,q}^{(l)}\mathbb{E} _{(u,v)\sim\mathcal{N}(0,\mathbf{A} _{p,q}^{(l)} )}[\mathbb{1}(u \ge 0)\mathbb{1}(v \ge 0)] + \mathbf{\Sigma} _{p,q}^{(l+1)}.
> $$
>
> Then, $\mathbf{H} = (\widetilde{\mathbf{H}}^{(L)} + \mathbf{\Sigma}^{(L)})/2$ is called the neural tangent kernel (NTK) matrix on the set of context-arm feature vectors $\left( x_{(n)} \right)_{n=1}^{TK}$.
>
> Whereas, $\mathbf{H}' = \sum_{s=1}^T \sum_{(i, j) \in C^K_2} z^i_j(s)z^i_j(s)^\top  \frac{1}{m}$, in which $z^i_j(s) = \phi(x_{s,i}) - \phi(x_{s,j})$ and $C^K_2$ denotes all pairwise combinations of $K$ arms (as defined in Line 223).
> To ensure completeness, we have added the definition of $\mathbf{H}$ at the beginning of Appendix A in the revised version of the paper.
>
>
> > **In Assumption 2, [x]_k = [x]_k+d/2, [] is used for multiple purposes, making some confusion. It should be defined clearly.**
>
> Thank you for pointing this out. We have fixed the notations in the revised paper.
>
>
> > **L191: Is \phi is g? Could the authors please explain why a different notation is needed if they are the same?**
>
> You are correct that $\phi(x)$ is same as $g(x;\theta_0)$. We use $\phi(x)$ as it helps simplify the presentation and makes the notation cleaner and more concise.
>
>
>
> *Thank you again for your time and your careful feedback. We hope our clarifications and answers to your questions will alleviate your concerns and improve your opinion of our work. If you have additional questions, we would be happy to address them.*

---

> ### Author Response · Authors · 2024-12-01
> **We hope our rebuttal has addressed your concerns**
>
> Dear Reviewer kAG9,
>
> Thank you once again for taking the time to review our paper and provide thoughtful and valuable feedback. We hope that our rebuttal has addressed your concerns.
>
> As the discussion period approaches its conclusion, we kindly ask if we could address any additional questions or comments. If you think that our responses have sufficiently addressed your concerns, we would be truly grateful if you could consider improving your evaluation of our paper.
>
> Best,
> Authors

---

### Official Review · Reviewer_aXCy · 2024-11-02

**Soundness:** 3
**Presentation:** 3
**Contribution:** 3
**Rating:** 8
**Confidence:** 2

**Summary:**

The paper titled *Neural Dueling Bandits: Principled Preference-Based Optimization with Non-Linear Reward Function* tackles the challenge of dueling bandits in contexts where the reward function is non-linear—a common scenario in real-world applications like recommendation systems and search ranking. The authors propose leveraging neural networks to model these non-linear reward functions based on preference feedback. They introduce two algorithms, *NDB-UCB* and *NDB-TS*, which are extensions of the Upper Confidence Bound and Thompson Sampling frameworks, respectively. Both algorithms come with sub-linear regret guarantees. The paper also extends the analysis to binary feedback settings and discusses implications for reinforcement learning from human feedback (RLHF). Theoretical results are supported by experiments on synthetic datasets.

**Strengths:**

**Originality**: The paper makes a significant contribution by extending the dueling bandit framework to accommodate non-linear reward functions using neural networks. This advances the field beyond the limitations of linear assumptions prevalent in prior work, allowing for more realistic modeling of user preferences.

**Theoretical Rigor**: The authors provide rigorous theoretical analysis, deriving sub-linear regret bounds for the proposed algorithms under standard assumptions. Extending these results to binary feedback settings enhances the robustness and applicability of their approach.

**Practical Relevance**: By addressing non-linear reward functions, the work is directly applicable to practical domains such as recommendation systems and web search, where user preferences are inherently complex. This increases the algorithms' relevance and potential impact on real-world applications.

**Clarity and Structure**: The paper is well-organized and clearly written. It presents the problem, proposed solutions, and theoretical foundations in a logical and accessible manner, making it easier for readers to comprehend the complex concepts involved.

**Impact on RLHF**: The exploration of reinforcement learning from human feedback broadens the impact of the work. By aligning AI systems more closely with human preferences, the paper contributes to an important area of AI research focused on ethical and user-aligned AI.

**Weaknesses:**

**Experimental Validation**: The empirical evaluation is limited to synthetic datasets. Testing the proposed algorithms on real-world datasets would strengthen the claims regarding their practical utility and effectiveness.

**Computational Complexity**: The paper lacks a detailed discussion on the computational demands of the algorithms, particularly concerning the training of neural networks in an online setting. Insights into scalability and efficiency are essential for practical deployment.

**Baseline Comparisons**: While comparisons are made with linear dueling bandit algorithms, the paper does not benchmark against other non-linear methods (e.g., those using Gaussian processes or kernel methods). Including such comparisons would provide a more comprehensive performance analysis.

**Questions:**

**Real-World Dataset Evaluation**: Have you considered evaluating your algorithms on real-world datasets, such as those from recommendation systems or search logs, to demonstrate their practical applicability and effectiveness?

**Computational Efficiency and Scalability**: Can you provide insights into the computational complexity of your algorithms? How do they scale with the number of contexts and arms, and are there strategies (e.g., model compression, online learning techniques) to mitigate computational overhead?

**Comparison with Other Non-linear Methods**: How does your approach compare to other non-linear bandit algorithms, particularly those leveraging Gaussian processes or deep kernel learning? Including comparisons with these methods could strengthen your empirical evaluation.

---

> ### Author Response · Authors · 2024-11-25
> **Clarifications about experiments and computational complexity**
>
> We would like to thank the reviewer for taking the time to review our paper. We also thank the reviewer for appreciating our contributions and acknowledging our realistic modeling, rigorous theoretical analysis, and practical applications of our results in areas like RLHF. We have answered each of your questions as follows:
>
> > **Real-World Dataset Evaluation: Have you considered evaluating your algorithms on real-world datasets, such as those from recommendation systems or search logs, to demonstrate their practical applicability and effectiveness?**
>
> Our experiments were primarily designed to validate our theoretical results and demonstrate that our proposed algorithms outperform existing contextual dueling bandit approaches, particularly when the underlying reward function is non-linear. To illustrate this, we use two synthetic functions, **Square** and **Cosine**, which are commonly used in earlier work on neural bandits (e.g., Zhou et al., 2020; Zhang et al., 2021; Dai et al., 2023). These controlled experiments provide a rigorous environment for evaluating algorithmic performance and verifying theoretical results.
>
>
> > **Computational Efficiency and Scalability: Can you provide insights into the computational complexity of your algorithms? How do they scale with the number of contexts and arms, and are there strategies (e.g., model compression, online learning techniques) to mitigate computational overhead?**
>
> To discuss the computational efficiency and scalability of our proposed algorithms, we consider the following two key aspects:
>
> **Size of the neural network (NN):** The primary computational complexity of our proposed algorithms is from the NN used to estimate the latent reward function. Let $d$ be the dimension of the context-arm feature vector (input dimension for the NN). Assume the NN has $L$ hidden layers, each with $m$ neurons. Then, the inference cost from the input layer to the first hidden layer is $O(dm)$, the inference cost for hidden layers is $O(Lm^2)$, and the inference cost for the final layer is $O(m)$. Therefore, the overall inference cost for each context-arm pair is $O(dm + Lm^2 + m)$. Note that $p=dm + Lm^2 + m$ is the total number of NN parameters (as discussed in Line 141). The training time for the NN is $O(ECLm^2)$, where $E$ is the number of training epochs and $C$ is the number of context-arm pairs observations. Therefore, choosing the appropriate size of NN for the given problem is very important, as having smaller NNs may not accurately approximate the latent reward function, while larger NNs can incur significant training and inference costs.
>
> **Number of contexts and arms:** Let $K$ be the number of arms and $p$ be the total number of NN parameters. Since we use the gradients with respect to the current estimate of NN as context-arm features, the gradient computation cost for each context is $O(Kdp)$, where $d$ is the dimension of the context-arm feature vector. The computational costs for computing reward estimate and confidence terms are $O(Kp)$ and $O(Kp^2)$, respectively. For selection arms, the computational cost for the first arm is $O(Kp + K)$, which includes estimating the reward for each context-arm pair $O(Kp)$ and then selecting the arm with the highest estimated reward $O(K)$; and the computational cost for the second arm is $O(Kp + (K-1)p^2)$, i.e., computing estimated reward $O(Kp)$ and confidence terms $O((K-1)p^2)$ for all arm with respect to the first selected arm  Thus, the overall computational cost of our proposed algorithms for selecting a pair of arms for each context is $O(Kdp + Kp^2)$. Since each context-arm pair is independent of others, each arm's gradients, reward estimate, and optimistic term can be computed in parallel. Consequently, the computational cost for selecting a pair of arms for each context can reduced to $O(dp + p^2)$.
>
> This discussion has been added in Appendix C.1 of our revised paper.
>
>
> > **Comparison with Other Non-linear Methods: How does your approach compare to other non-linear bandit algorithms, particularly those leveraging Gaussian processes or deep kernel learning?**
>
> Gaussian processes (GPs) and kernel-based methods are indeed popular choices for non-linear function approximation, but they have limited expressive power and fail when optimizing highly complex functions (Dai et al., 2023; Lin et al., 2023). While GPs are a better baseline than linear models, empirical studies have shown that GPs-based baselines perform poorly for the non-linear functions like **Square** and **Cosine** functions than neural network-based counterparts in contextual bandit settings (Zhang et al., 2021; Dai et al., 2023). In our revised paper version, we will also add Gaussian processes-based baselines.
>
>
> *Thank you again for your time and your careful feedback. We hope our answers will alleviate your concerns and improve your opinion of our work. If you have additional questions, we would be happy to address them.*

---

> > ### Comment · Reviewer_aXCy · 2024-11-28
> >
> > Thank you for addressing my questions. I appreciate your work on the theoretical solution, which is quite promising. I believe it would be beneficial to include experiments involving real-world datasets and systems in the final paper. This would effectively test your theoretical formulations in practical settings. All the best for the conference!

---

> > > ### Author Response · Authors · 2024-11-29
> > >
> > > We sincerely thank you for your valuable suggestions and appreciation of our theoretical results. We will include additional experiments with real-world datasets in the final version of our paper, as we agree that these will provide important insights into the practical applicability of our theoretical formulations.
> > >
> > > Thank you once again for your thoughtful feedback and kind wishes!

---

### Official Review · Reviewer_uMSZ · 2024-11-02

**Soundness:** 2
**Presentation:** 2
**Contribution:** 2
**Rating:** 5
**Confidence:** 4

**Summary:**

In this paper, the authors extend the contextual dueling bandit formulation to neural bandit settings, where the reward function is arbitrary and does not assume linearity. To address this task, they propose two methods based on the existing NeuralUCB and NeuralTS algorithms. Both theoretical and empirical analyses on synthetic datasets are provided.

**Strengths:**

The application of neural bandits to solve real-world dueling bandit problems is intuitive, as existing linear contextual dueling bandit algorithms may fail to capture complex reward relationships.

Additionally, the analysis of neural bandits with binary feedback may be of particular interest to the bandit research community.

**Weaknesses:**

Firstly, my concerns are with the problem formulation and theoretical analysis. I will list a few below. For instance:

If my understanding is correct, the problem formulation heavily relies on the link function $\mu$, whose properties significantly influence the regret bound, particularly through terms such as $\kappa\_{\mu}$ and $L\_{\mu}$. Furthermore, unlike conventional stochastic bandit works, this paper assumes the reward feedback associates with the sub-Gaussian noise, which in this case, depends on the chosen link function. Typically in stochastic bandit settings, sub-Gaussian noise is independent of the learning formulation and is determined solely by environmental settings, which makes this formulation unnatural.

Additionally, in Assumption 1, $\mathcal{X}$ is not clearly defined. Does $\mathcal{X}$ represent the entire arm context space with $\mathcal{X}\_t \subset \mathcal{X}$? If so, how would the learner possess prior knowledge of $\mathcal{X}$ to select the optimal $\kappa\_{\mu}$ value for exploration (i.e., the $\beta\_T$ term in Equation 3)? A similar issue also exists for $\tilde{d}$, as it is unrealistic to assume the learner has prior knowledge of this parameter.

The theoretical analysis appears to rely heavily on the proof framework of NeuralUCB and NeuralTS, adapting assumptions like the positive definite NTK matrix for arm context. The authors also mention the $c_0$ term, representing gradient differences between arm contexts in Theorem 2. This term should be clearly shown in the regret bound, as its data dependence could lead it to grow with $T$ and should not be hidden by big-O notation.

All experiments are conducted on synthetic datasets with predefined configurations. While the authors claim the paper is motivated by dueling bandit applications in LLM tuning, no experiments are included to support this claim. Furthermore, the theoretical analysis may not align with real RLHF settings, particularly given the strong over-parameterization assumption, which may not be realistic. In practice, each arm could be a generated response from an LLM, with high-dimensional vector representations and complex reward mappings. While RLHF methods like DPO employ powerful neural architectures, the analysis in this paper assumes an MLP, which may be insufficient. Therefore, I believe real dataset experiments relevant to LLM tuning, as indicated by the authors, are necessary.

**Questions:**

Please refer to "Weaknesses" above.

---

> ### Author Response · Authors · 2024-11-25
> **Clarifications about link function, sub-Gaussian noise,  and Assumption 1**
>
> We would like to thank the reviewer for taking the time to review our paper. We also thank the reviewer for acknowledging that our proposed algorithms address the shortcomings of existing algorithms and can be used to solve real-world problems with complex reward functions. We have addressed each of the weaknesses as follows:
>
> > **If my understanding is correct, the problem formulation heavily relies on the link function $\mu$, whose properties significantly influence the regret bound, particularly through terms such as $\kappa_\mu$ and $L_\mu$**.
>
> You are correct that our problem formulation and regret bound depend on the link function $\mu$ (through $\kappa_\mu$ and $L_\mu$). However, this is in fact consistent with most previous works on dueling bandits (Saha, 2021; Bengs et al., 2022; Li et al., 2024) and bandits with binary feedback (Filippi et al., 2010; Li et al., 2017; Faury et al., 2020). Therefore, the use of the link function $\mu$ is widely accepted and has proven effective in facilitating the development of useful algorithms in both theory and practice.
>
>
> > **Furthermore, unlike conventional stochastic bandit works, this paper assumes the reward feedback associates with the sub-Gaussian noise, which in this case, depends on the chosen link function. Typically in stochastic bandit settings, sub-Gaussian noise is independent of the learning formulation and is determined solely by environmental settings, which makes this formulation unnatural.**
>
> We would like to clarify that **we do not assume that the reward feedback is associated with sub-Gaussian noise**. Instead, **in our problem formulation, the noise $\epsilon_t=y_t-\mu(f(x_{t,1})-f(x_{t,2}))$ is guaranteed to be conditionally 1-sub-Gaussian**, which we have discussed briefly at the end of Section 3.4.1 and shown in more detail in the proof of Lemma 8. We give a brief explanation here: Firstly, note that given the history $F_{t-1}$,
> we have that $\mathbb{E}[\epsilon_t|{F}_{t-1}]=0$; next, since $y_t\in\{0,1\}$ and $\mu(\cdot)\in[0,1]$, we also have that $|\epsilon_t| \leq 1$. Therefore, $\epsilon_t$ is guaranteed to be conditionally 1-sub-Gaussian. For more details, please see the proof of Lemma 8 in Appendix A.2. Therefore, the 1-sub-Gaussian property of the noise $\epsilon_t$ does not depend on the link function, as long as $\mu(\cdot)\in [0,1]$.
>
>
> > **Additionally, in Assumption 1, $\mathcal{X}$ is not clearly defined. Does $\mathcal{X}$ represent the entire arm context space with $X_t \subset \mathcal{X}$? If so, how would the learner possess prior knowledge of $\mathcal{X}$ to select the optimal $\kappa_\mu$ value for exploration (i.e., the $\beta_T$ term in Equation 3)? A similar issue also exists for $\widetilde{d}$, as it is unrealistic to assume the learner has prior knowledge of this parameter.**
>
> Firstly, in Assumption 1, you are correct that $\mathcal{X}$ represents the entire arm context space with $\mathcal{X}_t\subset\mathcal{X}$. We have added the definition of $\mathcal{X}$ at the end of the first paragraph in Section 2 (Lines 95-96 in the revised paper). Thank you for pointing this out.
>
> In addition, the dependence of the exploration parameter $\beta_T$ on $\kappa_\mu$ (and hence on $\mathcal{X}$) is, in fact, very common in dueling bandits and bandits with binary feedback (Saha, 2021; Bengs et al., 2022; Li et al., 2024). Similarly, the dependence of the exploration parameter $\beta_T$ on $\widetilde{d}$ is prevalent in neural bandits (Zhou et al., 2020; Zhang et al., 2021). **In practice, we do not need to have prior knowledge of the values of $\kappa_\mu$ and $\widetilde{d}$** to calculate $\beta_T$. Instead, following the common practice of previous works, we can simply set the exploration parameter $\beta_T$ to be a constant.

---

> > ### Author Response · Authors · 2024-11-25
> > **Clarifications about theoretical analysis and experiments**
> >
> > > **The theoretical analysis appears to rely heavily on the proof framework of NeuralUCB and NeuralTS, adapting assumptions like the positive definite NTK matrix for arm context. The authors also mention the $c_0$ term, representing gradient differences between arm contexts in Theorem 2. This term should be clearly shown in the regret bound, as its data dependence could lead it to grow with and should not be hidden by big-O notation.**
> >
> > You are correct that our theoretical analysis has made use of the proof framework of NeuralUCB and NeuralTS. However, our analysis required significant novel techniques and presented non-trivial technical challenges. For example, our loss function (Eq. (4)) used to train the neural network is different from that of NeuralUCB and NeuralTS, which required novel analysis to obtain theoretical guarantees about the trained neural network; in the proof of the confidence ellipsoid (i.e., the proof of Lemma 6 and Theorem 1, given in Appendix A.2) which is one of the most crucial components of our analysis, we used significantly different proof techniques from those adopted by neural bandits; the subsequent regret analysis also contains major differences from NeuralUCB and NeuralTS, mostly because of our novel arm selection strategies.
> >
> > We have also followed your suggestion to make the dependence of the regret bound on $c_0$ explicit in Theorem 2. Specifically, we have revised the paper to add the detailed regret bound (without big-O notation) to Theorem 2, in which $c_0$ is clearly shown.
> >
> >
> > > **All experiments are conducted on synthetic datasets with predefined configurations. While the authors claim the paper is motivated by dueling bandit applications in LLM tuning, no experiments are included to support this claim. Furthermore, the theoretical analysis may not align with real RLHF settings, particularly given the strong over-parameterization assumption, which may not be realistic. In practice, each arm could be a generated response from an LLM, with high-dimensional vector representations and complex reward mappings. While RLHF methods like DPO employ powerful neural architectures, the analysis in this paper assumes an MLP, which may be insufficient. Therefore, I believe real dataset experiments relevant to LLM tuning, as indicated by the authors, are necessary.**
> >
> > Our primary objective in this work is to design experiments to validate our theoretical results and demonstrate that our proposed algorithms (NDB-UCB and NDB-TS) outperform existing contextual dueling bandit approaches, particularly in scenarios when the underlying reward function is complex and non-linear. To demonstrate the perforamnce of our proposed algorithms, we use two synthetic functions, **Square** and **Cosine** functions, which are commonly used in earlier work on neural bandits (e.g., Zhou et al., 2020; Zhang et al., 2021; Dai et al., 2023). These controlled experiments provide a rigorous environment for evaluating algorithmic performance and verifying theoretical results.
> >
> > Regarding the concern about over-parameterization, prior works on neural bandits (Zhou et al., 2020; Zhang et al., 2021; Dai et al., 2023; Lin et al., 2023) have demonstrated that over-parameterization is not a strict requirement for strong empirical performance.
> > In fact, MLP can perform well when it is combined with pretrained embeddings for generated responses from an LLM that capture high-level semantic information, as evidenced by recent work in prompt optimization for LLMs (Lin et al., 2023). Thus, the MLP assumption in our analysis is both practical and aligned with existing literature. We agree with the reviewer that extending our experiments to real-world problems, particularly in LLM fine-tuning using RLHF (Reinforcement Learning with Human Feedback) or DPO (Direct Preference  Optimization), would strengthen the applicability of our methods. However, our proposed algorithms will need to be adapted to these settings, which is beyond the scope of our current work, and we view this as a promising direction for future research.
> >
> >
> >
> > *Thank you again for your time and your careful feedback. We hope our clarifications will alleviate your concerns and improve your opinion of our work. If you have additional questions, we would be happy to address them.*

---

> ### Author Response · Authors · 2024-12-01
> **We hope our rebuttal has addressed your concerns**
>
> Dear Reviewer uMSZ,
>
> Thank you once again for taking the time to review our paper and provide thoughtful and valuable feedback. We hope that our rebuttal has addressed your concerns.
>
> As the discussion period approaches its conclusion, we kindly ask if we could address any additional questions or comments. If you think that our responses have sufficiently addressed your concerns, we would be truly grateful if you could consider improving your evaluation of our paper.
>
> Best,
> Authors

---

> ### Comment · Reviewer_uMSZ · 2024-12-02
>
> Thank you very much for the rebuttal. Since no additional experiments on real datasets related to LLM modeling have been introduced, my primary concern regarding the experiments remains unresolved. From my personal opinion, the current empirical evaluation can fail to adequately support the main motivation of this paper. I have decided to keep my score.

---

> > ### Author Response · Authors · 2024-12-03
> > **Clarifications about experiment setting and additional experiment results (Part 1)**
> >
> > We thank the reviewer for acknowledging our rebuttal. The following is our response to your concern regarding our experimental results.
> >
> > > **Since no additional experiments on real datasets related to LLM modeling have been introduced, my primary concern regarding the experiments remains unresolved. From my personal opinion, the current empirical evaluation can fail to adequately support the main motivation of this paper. I have decided to keep my score.**
> >
> > We want to highlight that the primary contribution of our work is theoretical: to the best of our knowledge, we are the first to show sub-linear regret upper bounds for the proposed contextual dueling bandit algorithms, NDB-UCB and NDB-TS, under the assumption that the latent reward function is non-linear. Following the common practice of using synthetically generated problem instances to evaluate the performance of contextual dueling bandit algorithms (Saha, 2021; Bengs et al., 2022; Li et al., 2024)  as it allows systematic analysis of the bandit algorithms' behavior across a variety of controlled settings), we also designed our experiments using synthetic problem instances to validate our theoretical results. Our experimental results demonstrated that the proposed algorithms significantly outperform existing contextual dueling bandit algorithms, especially when the reward function is non-linear.
> >
> >
> > ### **Real-world experiment related to LLMs**
> >
> > Despite the theoretical nature of our paper, we have followed your suggestion to add an additional experiment on a real dataset related to LLM modeling. As demonstrated in recent works (Bai et al., 2022; Menick et al., 2022; Mehta et al., 2023; Ji et al., 2024; Das et al., 2024), using responses with higher reward values can significantly improve the alignment of LLMs, especially in online/iterative RLHF where the responses generated by the initial LLM tend to have low rewards. Motivated by this, we use our proposed NDB-UCB algorithm to select the responses with higher estimated reward values for a given prompt. In our experiment, we use the following setting: In each iteration, a user provides a prompt (context in our setting) to LLM, and then LLM generates $50$ responses (arms in our setting). We use Sentence-BERT (Reimers, 2019) to get embedding representation for each prompt-response pair. Specifically, for iteration $t$, we use $x_{t, i} = (c_t, a_{t, i})$ to denote the embedding representation generated by Sentence-BERT, where $c_t$ denotes the prompt and $a_{t, i}$ represents the $i$-th response. Of note, **we adopt a reward model that is pre-trained using the Anthropic Helpfulness and Harmlessness datasets (Bai et al., 2022)**. For every prompt-response pair $x_{t, i}$, we use the output from this pre-trained reward model as the value of the unknown reward function $f(x_{t, i})$. This allows us to simulate the preference feedback in this experiment. Note that our approach to simulating the preference feedback is a common practice in the literature (Dwaracherla et al., 2024).
> >
> > **References:**
> > - N Reimers, Sentence-bert: Sentence embeddings using siamese bert-networks. EMNLP 2019.
> > - Bai et al., Training a helpful and harmless assistant with reinforcement learning from human feedback. arXiv 2022.
> > - Dwaracherla, et al., Efficient exploration for LLMs. arXiv 2024.

---

> > > ### Author Response · Authors · 2024-12-03
> > > **Additional experiment results (Part 2)**
> > >
> > > The embeddings $x_{t, i}$ are used as input to a trained neural network (NN) that estimates the latent reward model. Our proposed algorithm (NDB-UCB) computes the reward estimate for each prompt-response pair to select the first response (using Eq. (2) in the paper) and then uses it with the optimistic reward estimates of other responses to select the second response (using Eq. (3) in the paper). This way, our method will ensure the selection of responses with consistently high reward estimates. The table below compares the average cumulative reward (calculated as the sum of rewards for selected responses divided by the number of iterations) after $N$ iterations (multiple of 10) of NDB-UCB against two baselines: Lin-UCB (which uses a linear model to estimate latent reward model instead of an NN) and Random (where two responses are selected randomly). The results clearly show that NDB-UCB achieves the highest average cumulative reward, demonstrating the superior performance of our proposed algorithm in identifying high-reward responses. We will include these experimental results in the updated version of our paper.
> > >
> > >
> > > | Iteration           |          Random         |        LinDB-UCB           | NDB-UCB (ours) |
> > > |------------------|-----------------------|-------------------------------|------------------------|
> > > | Iteration 10     | 14.42    | 20.35                  | 21.82                  |
> > > | Iteration 20     | 15.52    | 25.20                  | 28.17                  |
> > > | Iteration 30     | 14.48    | 26.65                  | 32.79                  |
> > > | Iteration 40     | 15.07    | 27.27                  | 36.13                  |
> > > | Iteration 50     | 14.93    | 27.64                  | 38.17                  |
> > > | Iteration 60     | 15.03    | 27.89                  | 39.56                  |
> > > | Iteration 70     | 15.33    | 28.07                  | 40.65                  |
> > > | Iteration 80     | 15.20    | 28.21                  | 41.64                  |
> > > | Iteration 90     | 15.52    | 28.32                  | 42.72                  |
> > > | Iteration 100    | 15.78    | 28.40                  | 43.69                  |
> > > | Iteration 110    | 15.71    | 28.47                  | 44.53                  |
> > > | Iteration 120    | 15.74    | 28.53                  | 45.23                  |
> > > | Iteration 130    | 15.58    | 28.58                  | 45.83                  |
> > > | Iteration 140    | 15.55    | 28.62                  | 46.39                  |
> > > | |
> > >
> > >
> > >
> > > The experimental results shown in the above table corroborate our theoretical results, which guarantee the selection of responses with high reward values, as also discussed in our paper (Section 4.2 and Appendix E). This shows that our proposed algorithms can be used to improve the quality of the human preference dataset, which could then be used to improve the efficiency of online RLHF/DPO for LLM alignment. We leave it to future works to verify that the preference dataset with high-reward responses collected using our algorithms can indeed lead to better LLM alignment, which is beyond the scope of the current paper (since the focus of our paper is primarily theoretical) and will require significantly more computation.
> > >
> > >
> > >
> > > ---
> > > We sincerely hope that our new experimental results and clarifications could resolve your concern regarding the experiments. If you have any additional concerns or questions, we would be more than happy to address them.

---

### Official Review · Reviewer_nPZh · 2024-11-04

**Soundness:** 3
**Presentation:** 3
**Contribution:** 3
**Rating:** 6
**Confidence:** 4

**Summary:**

The paper studies the challenge of modeling dueling bandit problems where the reward function is non-linear. Traditional algorithms assume a linear reward function, which can be restrictive in real-life applications like online recommendations or web search rankings. The authors propose using neural networks to estimate the non-linear reward function using noisy preference feedback. They introduce upper confidence bound (UCB) and Thompson sampling (TS) based algorithms with sub-linear regret guarantees for efficient arm selection. Their theoretical results extend to contextual bandit problems with binary feedback, provide meaningful discussion with LLMs and experimental results on synthetic datasets support their theoretical findings.

**Strengths:**

The paper introduces neural network-based methods to estimate the non-linear reward function using preference feedback observed for previously selected arms. The paper proposes two algorithms: one based on UCB and another based on TS. Both algorithms are equipped with sub-linear regret guarantees, ensuring their effectiveness in selecting the best arm over time. Moreover, the authors provide the interesting discussion with contextual bandits with binary feedback and RLHF.

**Weaknesses:**

However, for the regret analysis, the authors still require the NTK gram matrix to be positive definite. This assumption can be a limitation in real-world applications, as it can be easily violated by repeating observed arm contexts, thereby hindering the practicality of the proposed methods.

**Questions:**

See weakness.

---

> ### Author Response · Authors · 2024-11-25
> **Clarifications about NTK matrix**
>
> We would like to thank the reviewer for taking the time to review our paper. We also thank the reviewer for acknowledging that our results are significant contributions to bandits. We have addressed the weakness you mentioned as follows:
>
>
> > **However, for the regret analysis, the authors still require the NTK gram matrix to be positive definite. This assumption can be a limitation in real-world applications, as it can be easily violated by repeating observed arm contexts, thereby hindering the practicality of the proposed methods.**
>
> The assumption of a positive definite NTK gram matrix is, in fact, **a common assumption adopted by previous works on neural bandits** (Zhou et al., 2020; Zhang et al., 2021). In addition, this assumption is only needed in our theoretical analysis and is hence **not required in the practical deployment of our algorithms**. In other words, in real-world applications, even if arm context observations are repeated (i.e. if the NTK gram matrix is not positive definite), neural bandit algorithms (including ours) can still be applied. For example, in the work of Lin et al. (2023), a neural bandit algorithm is used to solve the problem of instruction optimization with a discrete domain (in which the input observations are easily repeated) and still achieves outstanding performance.
> Therefore, this assumption, which is only needed in our theoretical analysis, does not hinder the practicality of our proposed methods.
>
>
> *Thank you again for your comments. We hope our clarification could improve your opinion of our work. If you have additional questions, we would be happy to address them.*

---

> ### Author Response · Authors · 2024-12-01
> **We hope our rebuttal has addressed your concerns**
>
> Dear Reviewer nPZh,
>
> Thank you once again for taking the time to review our paper and provide thoughtful and valuable feedback. We hope that our rebuttal has addressed your concerns.
>
> As the discussion period approaches its conclusion, we kindly ask if we could address any additional questions or comments. If you think that our responses have sufficiently addressed your concerns, we would be truly grateful if you could consider improving your evaluation of our paper.
>
> Best,
> Authors

---

> > ### Comment · Reviewer_nPZh · 2024-12-02
> >
> > Thanks for the authors' response. I would like to keep my score.

---

> > > ### Author Response · Authors · 2024-12-03
> > >
> > > We sincerely appreciate and thank you for acknowledging our rebuttal and maintaining your positive view of our paper.

---

### Meta-Review · Area_Chair_bkN3 · 2024-12-21

**Metareview:**

This paper tackles the challenge of contextual dueling bandits, where the goal is to identify the best arm for a given context using noisy preference feedback. Existing algorithms assume a linear reward function, but this can be too simplistic for real-world applications like online recommendations or search ranking. To address this, the authors propose using a neural network to estimate the reward function based on past feedback, and introduce upper confidence bound- and Thompson sampling-based algorithms with sub-linear regret guarantees. They extend their theoretical results to contextual bandit problems with binary feedback and validate their approach through experiments on synthetic datasets. The authors also adequately addressed the concerns of the reviewers, which led to raising the overall score of the paper. Based on the common consensus, I recommend acceptance of the paper once the authors incorporate all the reviewer's feedback in the final version.

**Additional Comments On Reviewer Discussion:**

See above.

---

### Decision · Program_Chairs · 2025-01-22

Accept (Poster)